# Hardware Acceleration for Neural Networks: A Comprehensive Survey

**Bin Xu**                                                                                      *binxu4@asu.edu*
*School of Electrical, Computer and Energy Engineering*
*Arizona State University, Tempe, USA*

**Ayan Banerjee**                                                                    *Ayan.Banerjee@asu.edu*
*School of Computing and Augmented Intelligence*
*Arizona State University, Tempe, USA*

**Sandeep Gupta**                                                                   *Sandeep.Gupta@asu.edu*
*School of Computing and Augmented Intelligence*
*Arizona State University, Tempe, USA*

**Reviewed on OpenReview:** *https://openreview.net/forum?id=Da8LO5NvDU&referrer=%5BAuthor+Console%5D%28%2Fgroup%3Fid%3DTMLR%2FAuthors%23your-submissions%29*

## Abstract

Neural networks have become a dominant computational workload across cloud and edge platforms, but their rapid growth in model size and deployment diversity has exposed hardware bottlenecks that are increasingly dominated by memory movement, communication, and irregular operators rather than peak arithmetic throughput. This survey reviews the current technology landscape for hardware acceleration of deep learning, spanning Graphics Processing Units (GPUs) and tensor-core architectures, domain-specific accelerators (e.g., Tensor Processing Units (TPUs)/Neural Processing Units (NPUs)), Field-Programmable Gate Array (FPGA)-based designs, Application-Specific Integrated Circuit (ASIC) inference engines, and emerging Large Language Model (LLM)-serving accelerators such as Language Processing Units (LPUs), alongside in-/near-memory computing and neuromorphic/analog approaches. We organize the survey using a unified taxonomy across (i) workloads (Convolutional Neural Networks (CNNs), Recurrent Neural Networks (RNNs), Graph Neural Networks (GNNs), Transformers/Large Language Models (LLMs)), (ii) execution settings (training vs. inference; datacenter vs. edge), and (iii) optimization levers (reduced precision, sparsity and pruning, operator fusion, compilation and scheduling, and memory-system/interconnect design). We synthesize key architectural ideas such as systolic arrays, vector and Single Instruction, Multiple Data (SIMD) engines, specialized attention and softmax kernels, quantization-aware datapaths, and high-bandwidth memory, and we discuss how software stacks and compilers bridge model semantics to hardware. Finally, we highlight open challenges—including efficient long-context LLM inference (Key-Value (KV)-cache management), robust support for dynamic and sparse workloads, energy- and security-aware deployment, and fair benchmarking—pointing to promising directions for the next generation of neural acceleration.

## 1 Introduction

Deep neural networks now underpin a wide range of applications—from perception and control in autonomous and cyber-physical systems Xu et al. (2025d;a) to search, recommendation, and natural language assistants Krizhevsky et al. (2012); Vaswani et al. (2017). At the same time, modern models have expanded rapidly in

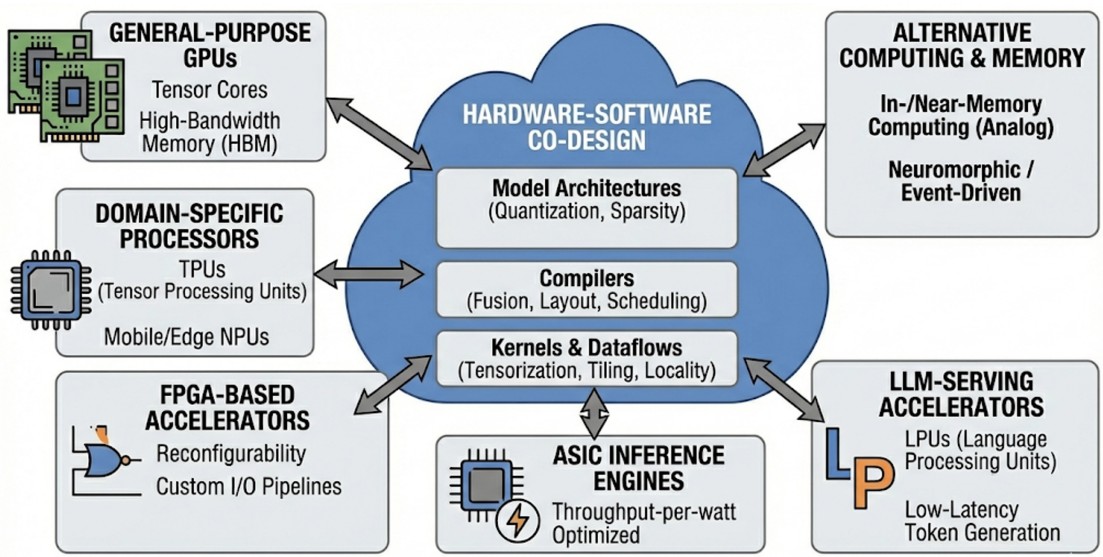

Figure 1: High-level overview of the hardware acceleration landscape, illustrating the spectrum from general-purpose GPUs to domain-specific TPUs, FPGAs, and ASICs, along with the interaction between compute datapaths, memory hierarchies, and the software stack.

scale (parameters, activation sizes, and context lengths), diversity (Convolutional Neural Networks (CNNs), Graph Neural Networks (GNNs), Transformers, diffusion models), and deployment environments (datacenter training clusters, latency-critical edge inference, and battery-powered on-device intelligence) Abadi et al. (2016); Gunter et al. (2024). These trends have elevated hardware acceleration from a performance optimization to an enabling technology: without specialized compute, memory systems, and software co-design, many state-of-the-art models would be impractical due to cost, latency, or energy constraints Hennessy & Patterson (2018); Jouppi et al. (2017).

Why acceleration is harder than "more Floating Point Operations (FLOPs)" is that, although matrix multiplication remains central, end-to-end efficiency is increasingly governed by data movement rather than arithmetic Williams et al. (2009). Training and inference pipelines stress memory capacity and bandwidth (e.g., activation storage and optimizer state during training; Key-Value (KV)-cache and batching dynamics during Large Language Model (LLM) inference) Micikevicius et al. (2018); Kwon et al. (2023), interconnect bandwidth and collectives for distributed execution, and irregular operators arising from sparsity, dynamic control flow, and Mixture-of-Experts (MoE) routing Gale et al. (2019); Shazeer et al. (2017). As a result, accelerator design must address the full system stack: compute datapaths, memory hierarchies, on-chip networks, off-chip Dynamic Random Access Memory (DRAM)/High-Bandwidth Memory (HBM), host-device interfaces, and multi-device interconnects, together with compilers and runtime systems that map high-level models onto hardware efficiently Abadi et al. (2016).

The technology landscape shown in Figure 1 spans general-purpose Graphics Processing Units (GPUs) enhanced with tensor cores and high-bandwidth memory; domain-specific processors such as Tensor Processing Units (TPUs) and mobile/edge Neural Processing Units (NPUs) Jouppi et al. (2017; 2021); Field-Programmable Gate Array (FPGA)-based accelerators offering reconfigurability and tight integration with custom I/O pipelines Nurvitadhi et al. (2017); Umuroglu et al. (2017); Boutros et al. (2024); Application-Specific Integrated Circuit (ASIC) inference engines optimized for throughput-per-watt Chen et al. (2014); and LLM-serving accelerators such as Language Processing Units (LPUs) that target predictable low-latency token generation Groq (2024). Beyond digital Complementary Metal-Oxide-Semiconductor (CMOS) datapaths, in-/near-memory computing and analog approaches attempt to reduce the fundamental cost of memory movement Shafiee et al. (2016), while neuromorphic and event-driven designs explore alternative execution models for sparse, low-power workloads Davies et al. (2018). In practice, the most capable systems

emerge from hardware–software co-design: model architectures and training recipes that are amenable to quantization and sparsity Jacob et al. (2018); Han et al. (2016b); compilers that perform fusion, layout transformations, and scheduling; and kernels and dataflows that exploit tensorization, tiling, and locality Kung (1982); Williams et al. (2009).

In this survey, we focus on current technology and architectural principles that generalize across devices and workloads. We structure the discussion along three axes: (i) *workloads* (CNNs, Recurrent Neural Networks (RNNs), GNNs, and Transformers; and key operators such as attention, convolution, normalization, and sampling) Vaswani et al. (2017), (ii) *execution settings* (training vs. inference; offline vs. online; cloud vs. edge), and (iii) *optimization levers* (reduced precision Micikevicius et al. (2018); Jacob et al. (2018), structured and unstructured sparsity Gale et al. (2019); Fang et al. (2024), compression and pruning Han et al. (2016b); Muralidharan et al. (2024); Liu et al. (2025), operator fusion, compilation and scheduling, and memory/interconnect design Williams et al. (2009); Wang et al. (2024); Li et al. (2024)). Rather than treating hardware in isolation, we highlight how these levers interact—for example, how low precision shifts bottlenecks toward memory, how sparsity challenges utilization and load balance, and how long-context inference amplifies memory-capacity constraints Kwon et al. (2023).

The main contributions of this survey are:

- a unified taxonomy connecting neural workloads to accelerator architectures and system constraints;

- a synthesis of design patterns used in modern accelerators (tensor-core/tensorization, systolic arrays, tiling and dataflow, heterogeneous memory hierarchies, and scalable interconnects);

- a discussion of software stacks (frameworks, compilers, and runtimes) that determine real-world performance and portability; and

- an overview of open challenges and research directions, including efficient LLM serving (KV-cache and memory management), robust support for dynamic and sparse models, energy-aware edge deployment, and reproducible benchmarking.

The remainder of this survey is organized as follows. **Section 2** discusses accelerator challenges spanning power, throughput, area, memory, utilization, and benchmarking, analyzed per platform family. **Section 3** surveys hardware accelerator architectures—GPUs, TPUs/NPUs, ASICs, FPGAs, LPUs, PIM/analog, and neuromorphic—each covering inference and training. **Section 4** presents workload-specific acceleration strategies for ANNs, CNNs, RNNs, and Transformers/LLMs, including compiler and runtime systems. **Section 5** covers evaluation methodology, and **Section 6** concludes with open research directions.

## 2 AI Workloads: Definitions and Hardware Implications

The computational demands of artificial intelligence vary dramatically across the model lifecycle. Understanding the distinct characteristics of each workload stage—pre-training, fine-tuning, and inference—is essential for mapping hardware capabilities to application requirements. Figure 2 illustrates the computational structure of these three workload categories: training and fine-tuning are dominated by dense matrix–matrix multiplications with iterative forward and backward passes (compute-bound), LLM inference is constrained by sequential matrix–vector operations and growing KV-cache access (memory-bandwidth-bound), and batch inference for non-autoregressive models executes a single forward pass through a layer pipeline (compute-bound at scale). This section defines each workload category, characterizes its computational profile, and examines how different accelerator families address its demands.

### 2.1 Pre-Training

Pre-training is the most compute-intensive phase of the AI model lifecycle. A model learns broad statistical representations from massive corpora—often trillions of tokens for large language models (LLMs)—by iteratively performing forward passes, loss computation, backpropagation, and weight updates across billions of parameters. A single training run for a frontier model such as GPT-4 ($\sim$1.76 trillion parameters) has been

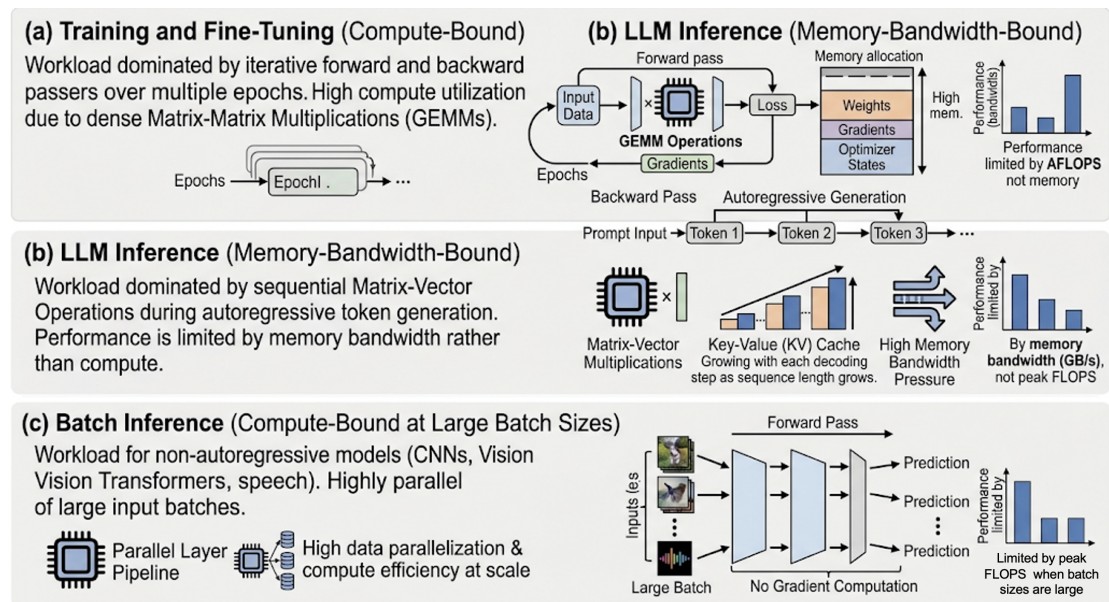

Figure 2: Workload profiles for the three primary AI compute stages. (a) **Training and fine-tuning** are compute-bound, dominated by dense matrix–matrix multiplications (GEMMs) with iterative forward and backward passes over multiple epochs; memory must simultaneously hold weights, gradients, and optimizer states. (b) **LLM inference** is memory-bandwidth-bound due to sequential matrix–vector operations during autoregressive token generation; the KV cache grows with sequence length, increasing bandwidth pressure at each decoding step. (c) **Batch inference** for non-autoregressive models (CNNs, vision transformers, speech) is compute-bound at large batch sizes, executing a single forward pass through a layer pipeline with no gradient computation.

estimated to require ∼25,000 A100 GPUs operating continuously for approximately 90 days Cottier et al. (2024), consuming on the order of $10^{25}$ floating-point operations (FLOPs).

Pre-training is fundamentally *compute-bound*: throughput scales with peak FLOPS and is dominated by dense matrix–matrix multiplications (GEMMs) with high arithmetic intensity. The principal hardware requirements are as follows:

- **High-precision compute:** FP32, BF16, or FP8 arithmetic executed on large tensor cores or systolic arrays.

- **Large memory capacity:** Model parameters, gradients, optimiser states (e.g., Adam momentum and variance), and activations must reside simultaneously in memory. For a model with $N$ parameters trained in mixed precision, the total memory footprint is approximately $16N$–$20N$ bytes when using Adam Rajbhandari et al. (2020).

- **High memory bandwidth:** Sustained data movement between memory and compute units, typically requiring HBM with terabytes-per-second bandwidth.

- **High-speed interconnects:** Multi-GPU and multi-node parallelism (data, tensor, pipeline, and expert parallelism) demands inter-chip bandwidths of hundreds of GB/s to minimise synchronisation overhead.

These requirements favor data-center-class accelerators such as the NVIDIA B200 (2.2 PFLOPS FP8, 8 TB/s HBM3e), Google TPU v5p (459 TFLOPS BF16, 2.76 TB/s HBM2e), and AMD MI300X (1.3 PFLOPS FP16, 5.3 TB/s HBM3), as listed in Table 1. FPGAs and edge accelerators are generally unsuitable for pre-training due to insufficient memory capacity and floating-point throughput.

## 2.2 Fine-Tuning and Adaptation

Fine-tuning adapts a pre-trained foundation model to a specific downstream task or domain by continuing training on a curated, task-specific dataset. Compared to pre-training, fine-tuning operates on substantially smaller data volumes and fewer epochs, reducing total compute by two to four orders of magnitude. However, the *per-iteration* computational profile—forward pass, backpropagation, and weight update—remains structurally identical, preserving the need for gradient storage and high-precision arithmetic.

Parameter-efficient fine-tuning (PEFT) methods such as Low-Rank Adaptation (LoRA) Hu et al. (2022) and adapters Houlsby et al. (2019) further reduce the computational and memory burden by freezing most model weights and training only a small number of additional parameters (typically 0.1–1% of the original). This shifts the workload from being purely compute-bound toward a more balanced compute–memory profile, making fine-tuning feasible on single high-end GPUs, small clusters, or even reconfigurable platforms such as the AMD Alveo V80 for custom low-precision datapaths. Reinforcement learning from human feedback (RLHF), a common post-training alignment technique, introduces additional complexity by requiring simultaneous inference of a reward model alongside policy gradient updates, further increasing memory pressure.

## 2.3 Inference

Inference applies a trained model to new inputs to generate predictions. Unlike training, inference involves only forward passes with no gradient computation, reducing both the memory footprint (parameters and activations only) and the arithmetic requirements per query. However, inference workloads differ qualitatively from training in several important respects:

- **Latency sensitivity:** Real-time applications (chatbots, autonomous driving, speech recognition) require end-to-end response times on the order of milliseconds, making latency—not throughput—the primary optimization target.

- **Memory-boundedness:** Autoregressive decoding in LLMs performs sequential matrix–vector multiplications with low arithmetic intensity ($\ll 1\,\mathrm{FLOP/byte}$), causing inference to bottleneck on memory bandwidth rather than peak compute Advanced Micro Devices, Inc. (2024b). This contrasts sharply with the compute-bound GEMMs that dominate training.

- **Variable batch sizes:** Production serving must handle fluctuating request volumes, from single-user queries (batch size 1) to high-throughput batch processing.

- **Reduced precision:** Models can be quantized to INT8, INT4, or even lower precision with minimal accuracy degradation, favouring hardware with efficient low-precision datapaths.

These characteristics open the design space to a broader range of accelerator architectures. Data-center inference benefits from GPUs (NVIDIA B200), purpose-built inference ASICs (Qualcomm Cloud AI 100 Ultra at 870 TOPS INT8 with 5.8 TOPS/W), deterministic dataflow engines (Groq LPU at 750 TOPS INT8 with sub-millisecond latency), and processing-in-memory solutions (Samsung HBM-PIM) that address the memory wall by performing computation directly within the DRAM banks. Neuromorphic processors such as Intel Loihi 2 exploit event-driven sparsity for ultra-low-power spiking neural network execution within extreme power envelopes ($\sim 2.3\,\mathrm{W}$ per chip), targeting specialised sensing and always-on workloads rather than mainstream dense inference.

## 2.4 Quantitative Hardware–Workload Mapping

Figure 4 presents a log–log scatter plot of peak compute (TOPS-equivalent) versus energy efficiency (TOPS/W) for the accelerator platforms listed in Table 1, color-coded by their primary workload affinity. Dashed iso-power lines indicate constant TDP envelopes (1 W, 10 W, 100 W, 1000 W), providing a visual anchor for the power budget associated with each platform. Several key insights emerge from this cross-platform comparison:

1. **Performance–efficiency trade-off:** Data-center accelerators (NVIDIA B200, AMD MI300X, Google TPU v5p) cluster in the high-TOPS, moderate-TOPS/W region of the plot, reflecting architectures optimized for absolute throughput within thermal envelopes of 250–1000 W. The dedicated inference ASICs (Qualcomm Cloud AI 100 Ultra, Groq LPU) achieve higher TOPS/W at comparable or slightly lower absolute TOPS, demonstrating the efficiency gains available when generality is traded for workload specialisation. At the opposite extreme, Intel Loihi 2 occupies the low-TOPS, high-TOPS/W corner, prioritising energy efficiency within a ∼2.3 W envelope.

2. **The memory-wall gap:** Despite offering high peak TOPS, GPU and TPU platforms frequently underutilise their compute capacity during memory-bound inference workloads. Autoregressive LLM decoding at batch size 1 achieves only 30–40% of the datasheet TOPS on conventional accelerators, whereas architectures that co-locate compute with memory—such as the Samsung HBM-PIM (307.2 GB/s in-DRAM bandwidth) and the Groq LPU (80 TB/s on-chip SRAM)—can sustain higher effective utilisation for these workloads.

3. **Workload-specific niches:** No single architecture dominates across all workloads. Training demands the highest absolute FLOPS and memory capacity, favouring GPUs and TPUs. Latency-critical data-center inference benefits from deterministic architectures (Groq LPU) or purpose-built ASICs (Qualcomm Cloud AI 100 Ultra). Processing-in-memory (Samsung HBM-PIM) addresses the memory wall for bandwidth-limited layers. Reconfigurable workloads benefit from FPGAs (AMD Alveo V80) that can adapt their datapaths to evolving model architectures. Neuromorphic platforms (Intel Loihi 2) serve ultra-low-power, event-driven sensing tasks.

4. **Precision as a design lever:** The transition from FP32 (pre-training) through BF16/FP8 (mixed-precision training) to INT8/INT4 (inference) yields multiplicative gains in both throughput and efficiency at each stage, directly reflected in the TOPS/W spread across platforms. Notably, the NVIDIA B200 reports 4.5 PFLOPS at FP4 versus 2.2 PFLOPS at FP8—a 2× improvement from a single precision step—underscoring precision's role as a first-order architectural knob.

Table 1: Comparison of representative accelerator platforms (approximate per-chip specifications; exact values vary by SKU, configuration, and software stack). TOPS/W is derived from peak INT8/FP8 compute and reported TDP.

| Platform | Peak Compute | TDP | TOPS/W | Mem. BW | Mem. Cap. | Typical Use-Case |
|---|---|---|---|---|---|---|
| NVIDIA B200 (Blackwell) NVIDIA Corporation (2024a) | ∼4.5 PFLOPS (FP4) ∼2.2 PFLOPS (FP8) | 1000 W | ∼4.5 (FP4) | 8 TB/s (HBM3e) | 192 GB (HBM3e) | Training + inference |
| Google TPU v5p Google Cloud (2024) | ∼459 TFLOPS (BF16) | ∼250 W | ∼1.8 (BF16) | 2.76 TB/s (HBM2e) | 95 GB (HBM2e) | Large-scale training |
| Groq LPU (GroqChip 1) Groq (2024) | ∼750 TOPS (INT8) | ∼300 W | ∼2.5 (INT8) | ∼80 TB/s (on-chip SRAM) | 230 MB (SRAM) | Low-latency LLM inference |
| AMD MI300X AMD (2024) | ∼1.3 PFLOPS (FP16) | 750 W | ∼1.7 (FP16) | 5.3 TB/s (HBM3) | 192 GB (HBM3) | Training + inference |
| AMD Alveo V80 (Versal HBM) Advanced Micro Devices, Inc. (2024a) | 10,848 DSP58s (config-dependent) | 300 W | Design-dependent | 820 GB/s (HBM2e) | 32 GB (HBM2e) | HPC + reconfig. inference |
| Qualcomm Cloud AI 100 Ultra (ASIC) Qualcomm Technologies, Inc. (2024) | ∼870 TOPS (INT8) | 150 W | ∼5.8 (INT8) | 548 GB/s (LPDDR4x) | 128 GB (LPDDR4x) | DC AI inference |
| Samsung HBM-PIM (Aquabolt-XL) Lee et al. (2021) | 1.2 TFLOPS (FP16, 32 PCUs) | ∼5 W (per stack) | ∼240 (GFLOPS/W) | 307.2 GB/s (HBM2 ext.) | 4 GB (HBM2) | PIM AI inference |
| Intel Loihi 2 (Hala Point) Davies et al. (2021) | ∼330 GSOPS (synaptic ops) | ∼2.3 W (per chip) | ∼143 (GSOPS/W) | ∼14 TB/s (on-chip SRAM) | 25 MB (SRAM) | Neuromorphic SNN inference |

At a high level, each accelerator family occupies a distinct region of this multi-dimensional design space:

- **GPUs:** High programmability, broad operator coverage, and mature toolchains (CUDA, ROCm); lower energy efficiency than specialised designs.
- **TPUs/NPUs:** Dense tensor throughput with structured datapaths; strong for regular workloads but reduced flexibility for irregular operators or dynamic control flow.
- **FPGAs:** Reconfigurability, custom precision, and streaming pipelines; higher per-operation cost than ASICs but adaptable to evolving model architectures and mixed-precision requirements.

- **Inference ASICs:** Maximum TOPS/W and area efficiency for targeted operators; limited generality, though software–hardware co-design can broaden effective coverage.
- **In-/near-memory (PIM):** Minimal data movement for matmul-dominated layers; constrained by ADC/DAC overhead, precision limits, and non-matmul operator support.
- **Neuromorphic:** Ultra-low power for sparse, event-driven workloads; limited applicability to dense, mainstream DNN models but compelling for always-on sensing and bio-inspired computation.

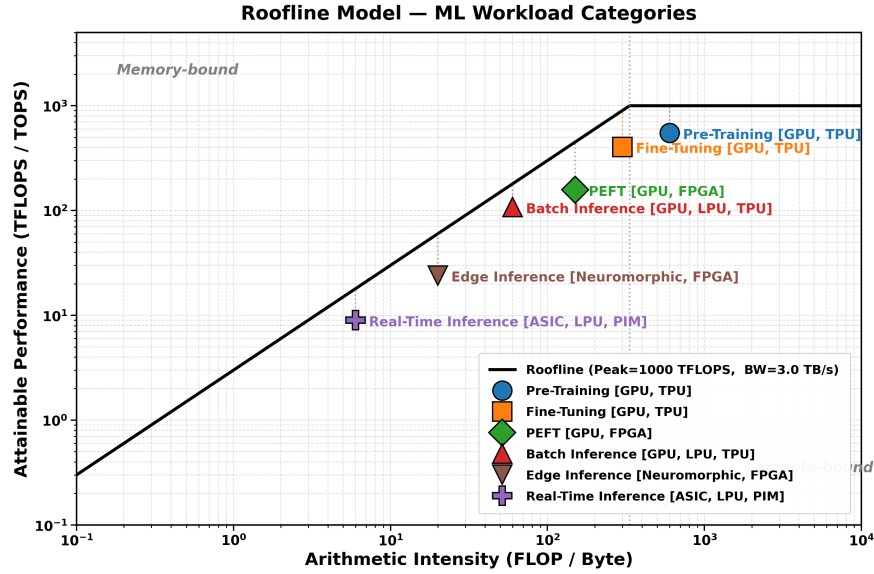

Figure 3: Roofline view of the six workload categories from Table 2 on a reference accelerator (1000 TFLOPS, 3 TB/s, ridge ≈ 333 FLOP/Byte). **Pre-Training** sits on the compute-bound roof, **Fine-Tuning** and **PEFT** cluster near the ridge, while **Batch**, **Edge**, and **Real-Time Inference** fall into the memory-bound region. The placement shift from GPU/TPU at high arithmetic intensity to LPU/ASIC/PIM and FPGA/neuromorphic substrates at low arithmetic intensity reflects the transition from compute-bound to bandwidth- and power-limited regimes.

Table 2: Workload characteristics and corresponding hardware requirements. Arrows indicate relative demand: ↑↑↑ = critical, ↑↑ = important, ↑ = moderate, — = not applicable.

| Characteristic | Pre-Training | Fine-Tuning | PEFT | Batch Inference | Real-Time Inference | Edge Inference |
|---|---|---|---|---|---|---|
| Peak FLOPS / TOPS | ↑↑↑ | ↑↑ | ↑ | ↑↑ | ↑ | ↑ |
| Memory capacity | ↑↑↑ | ↑↑ | ↑ | ↑↑ | ↑ | ↑ |
| Memory bandwidth | ↑↑ | ↑↑ | ↑ | ↑↑↑ | ↑↑↑ | ↑↑ |
| Interconnect BW | ↑↑↑ | ↑↑ | ↑ | ↑ | — | — |
| Latency tolerance | High | High | Moderate | Moderate | Low | Low |
| Precision | FP32/BF16/FP8 | BF16/FP8 | BF16/FP8 | INT8/INT4 | INT8/INT4 | INT8/INT4/binary |
| Power envelope | 300–1000 W | 300–750 W | 75–300 W | 300–1000 W | 75–300 W | 1–15 W |
| **Best-fit platforms** | GPU, TPU | GPU, TPU | GPU, FPGA | GPU, LPU, TPU | ASIC, LPU, PIM | Neurom., FPGA |

Table 2 summarizes the mapping between workload characteristics and hardware requirements, providing a concise reference for the platform-level analysis that follows. Figure 3 recasts this mapping in roofline form, plotting each workload category against a reference accelerator ceiling: **Pre-Training** occupies the compute-bound flat roof, **Fine-Tuning** and **PEFT** cluster near the ridge point, and **Batch**, **Edge**, and **Real-Time Inference** slide progressively into the memory-bound slope as their FLOPS-to-bandwidth ratios decrease. This visual ordering makes explicit why no single substrate dominates the workload spectrum— high-AI training regimes reward peak FLOPS (GPU, TPU), whereas low-AI, latency-critical inference rewards bandwidth proximity and specialization (LPU, ASIC, PIM, neuromorphic).

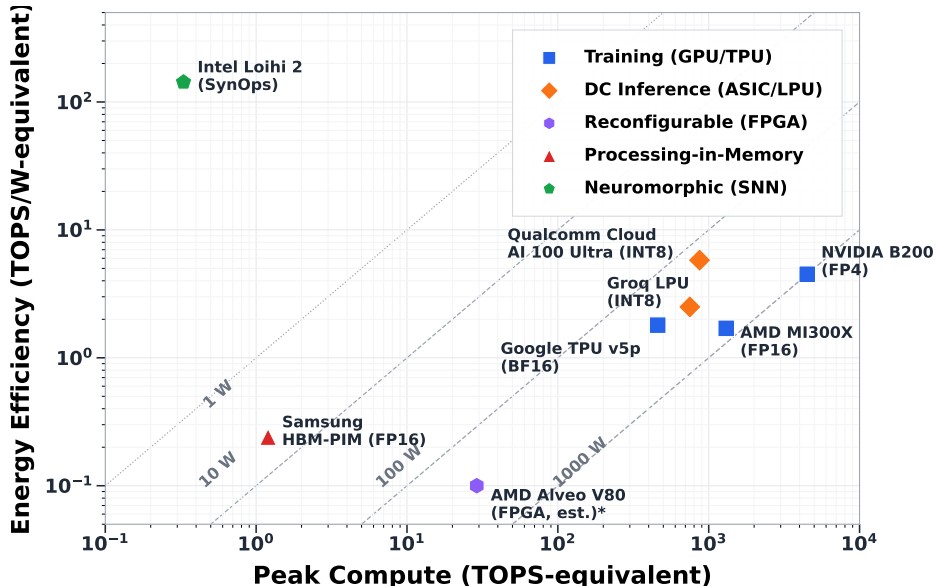

Figure 4: Peak compute (TOPS-equivalent, log scale) versus energy efficiency (TOPS/W-equivalent, log scale) for eight representative accelerator platforms from Table 1. Marker shape and color indicate primary workload affinity: ■ training (GPU/TPU), ♦ data-center inference (ASIC/LPU), ○ reconfigurable (FPGA), ▲ processing-in-memory, and ⬠ neuromorphic (SNN). Dashed lines denote constant-TDP iso-power envelopes. The AMD Alveo V80 FPGA TOPS value is estimated from its 10,848 DSP58 slices; actual throughput is design-dependent.

The data presented in Table 1 and Figure 4 underscore a central thesis of this survey: *workload diversity necessitates hardware diversity.* The eight-order-of-magnitude spread in peak compute—from ∼330 GSOPS for Loihi 2 to ∼4.5 PFLOPS for the B200—and the three-order spread in energy efficiency—from ∼0.1 TOPS/W for the Alveo V80 FPGA to ∼143 GSOPS/W for Loihi 2—reflect fundamentally different design trade-offs: throughput versus efficiency, flexibility versus specialization, and general-purpose programmability versus workload-specific optimization. No single architecture simultaneously optimizes all dimensions. Subsequent sections analyze how each accelerator family navigates these trade-offs in detail.

## 3   Accelerator Challenges

Designing a hardware accelerator is fundamentally a multi-objective optimization problem spanning five interrelated axes: *speed* (throughput and latency), *energy efficiency* (power consumption), *memory and communication* (bandwidth and capacity), *area & cost* (silicon and system-level deployment), and *benchmarking & reproducibility* (fair evaluation across heterogeneous software stacks). Cutting across all five is a sixth, often underappreciated, dimension: *flexibility and programmability*—the degree to which a platform can accommodate new operators, evolving model architectures, and variable-precision datapaths without a hardware respin. Figure 5 visualizes these trade-offs and maps each accelerator family to its characteristic limitations.

Gains along one axis typically come at the expense of another. Highly specialized architectures such as inference ASICs and neuromorphic processors achieve superior TOPS/W and area efficiency but support only a narrow set of operators; when a new activation function or attention variant emerges, these designs may require a costly respin or simply cannot execute the workload. General-purpose GPUs trade some of that efficiency for broad programmability: their SIMT execution model and mature toolchains (CUDA, ROCm) have absorbed successive architectural shifts—from CNNs to Transformers to Mixture-of-Experts—through software updates alone, albeit with higher memory-traffic overhead and reduced operator-fusion efficiency. TPUs and NPUs occupy a structured middle ground: dense tensor throughput via fixed systolic

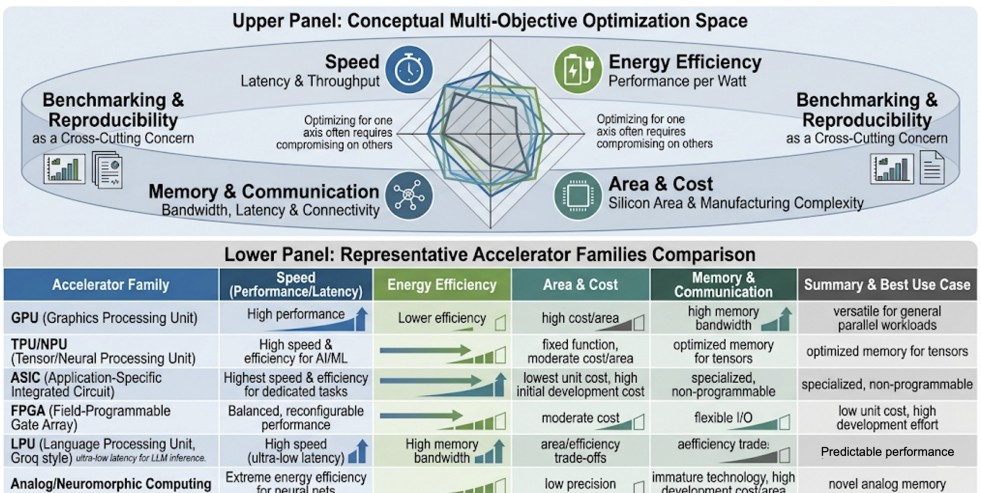

Figure 5: Hardware accelerator design as a multi-objective optimization across **Speed**, **Energy Efficiency**, **Area & Cost**, and **Memory & Communication**, with **Benchmarking & Reproducibility** as a cross-cutting concern. The lower panel summarizes the pros and cons of six representative accelerator families (GPU, TPU/NPU, ASIC, FPGA, LPU, and Analog/Neuromorphic), highlighting how each substrate navigates these axes differently.

datapaths delivers high utilization on regular workloads, yet dynamic control flow, irregular sparsity, and non-standard tensor shapes can degrade utilization substantially. FPGAs offer post-fabrication reconfigurability, enabling custom-precision pipelines and streaming architectures that can be recompiled as models evolve; this flexibility, however, comes at the cost of lower clock frequencies, higher per-operation energy, and significantly more complex toolchains. Emerging LPU designs such as the Groq which targets inference-specific bottlenecks—deterministic scheduling and KV-cache bandwidth—achieving predictable tail latency at the expense of training capability and model-size generality.

This flexibility–efficiency tension is not static. As model architectures evolve at an accelerating pace—from dense Transformers to sparse Mixture-of-Experts, from static computation graphs to dynamic execution with early exit and conditional routing—the programmability axis grows increasingly important, favoring platforms that can adapt without silicon respin. Moreover, "speed" itself is multifaceted: training workloads emphasize time-to-convergence and multi-device scalability, whereas inference workloads prioritize per-request latency, tail-latency guarantees under load, and throughput at a target quality level. Across both regimes, energy and area budgets further constrain what can be deployed, particularly when models must run continuously in power-limited data centers or at the edge Hennessy & Patterson (2018). The accelerator families shown at the base of Figure 5—GPU, TPU/NPU, ASIC, FPGA, LPU, and analog/neuromorphic—each anchor to a different region of this trade-off surface; no single platform simultaneously optimizes all dimensions, motivating the heterogeneous, workload-specific design strategies examined in subsequent sections.

Underpinning these trade-offs is a recurring observation: peak arithmetic throughput alone is rarely the binding constraint. End-to-end performance depends on the balance among compute, memory bandwidth, and communication, and is often dominated by data movement across memory hierarchies and interconnects Williams et al. (2009); Wulf & McKee (1995). This imbalance is amplified in modern workloads such as Transformer-based LLMs, where attention mechanisms and KV-cache management introduce large memory footprints and bandwidth demands that scale with both sequence length and serving concurrency Vaswani et al. (2017); Kwon et al. (2023).

### 3.1 Power and energy consumption

Power consumption is a first-order constraint because it determines thermal limits, battery lifetime, and operating cost. A key observation is that energy is consumed not only by arithmetic, but also by moving data through memory hierarchies and interconnects; for many modern workloads, the energy per byte moved can dominate the energy per operation Horowitz (2014); Williams et al. (2009). This motivates accelerator designs and mappings that reduce off-chip traffic through locality-aware dataflows and larger on-chip buffers Chen et al. (2016).

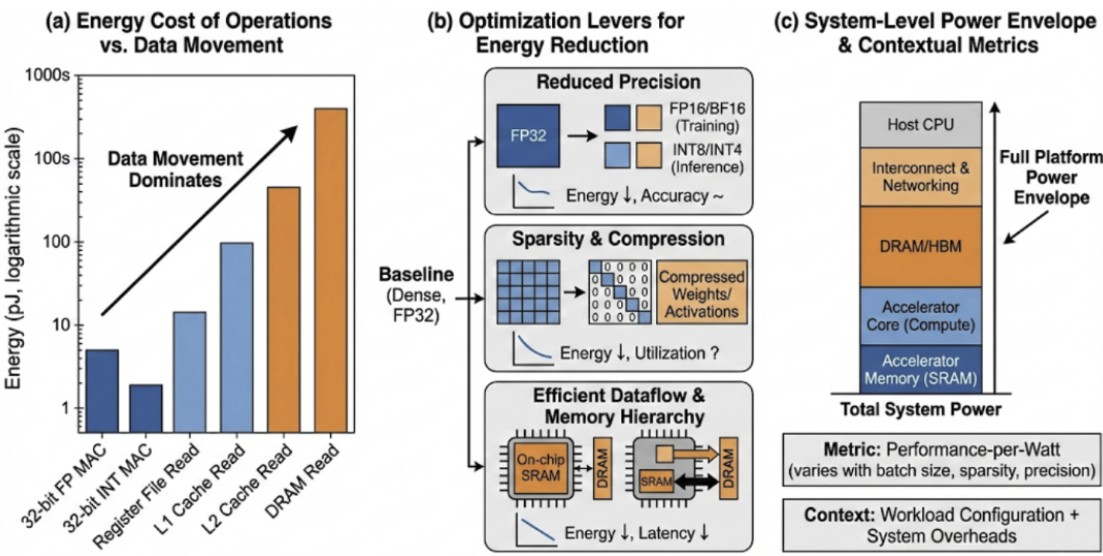

Figure 6: Power consumption analysis highlighting (a) the dominance of data movement energy over arithmetic operations, (b) the impact of reduced precision on energy efficiency, and (c) the trade-offs involved in exploiting sparsity.

Figure 6 highlights the power-consumption perspective that motivates energy-aware mappings and reduced data movement. Precision and sparsity are primary levers for energy reduction. Mixed-precision training improves throughput and reduces memory traffic while maintaining convergence, typically using reduced-precision compute with higher-precision accumulation and stabilization techniques Micikevicius et al. (2018). For inference, quantization to integer arithmetic can significantly reduce energy and improve effective bandwidth, but it introduces calibration and accuracy trade-offs that depend on the model and deployment target Jacob et al. (2018). Compression and sparsity reduce memory footprint and traffic, but unstructured sparsity can add indexing overhead and reduce utilization if the hardware/software stack is not sparsity-aware Han et al. (2016b); Gale et al. (2019).

Meaningful energy comparisons must therefore be contextual. Performance-per-watt numbers are sensitive to batch size, sequence length, precision, sparsity pattern, and system-level overheads (host CPU work, DRAM/HBM power, and network power). As a result, energy evaluation should report both workload configuration and the full platform power envelope rather than only accelerator core power.

#### 3.1.1 GPU-specific energy challenges

On GPUs, energy is frequently dominated by memory traffic rather than arithmetic, especially when arithmetic intensity is low and intermediate tensors are repeatedly materialized Williams et al. (2009); Horowitz (2014). Inference pipelines that lack operator fusion can spend a large fraction of energy moving activations between registers, caches, and off-chip memory, which motivates aggressive fusion in libraries and compiler stacks Chetlur et al. (2014); Chen et al. (2018b); Tillet et al. (2019). Mixed precision improves throughput-per-watt for training, but it can shift the bottleneck toward memory and communication when scaling across devices Micikevicius et al. (2018); Shoeybi et al. (2019). Sparsity can reduce memory traffic, yet unstruc-

tured sparsity may increase metadata and indexing overhead that erodes energy gains without sparsity-aware kernels Gale et al. (2019); Han et al. (2016b).

### 3.1.2 TPU/NPU-specific energy challenges

Tensor processors improve efficiency by specializing dense dataflows and maximizing on-chip reuse, but their energy advantage diminishes when workloads deviate from the assumed shapes or require frequent off-chip traffic Jouppi et al. (2017; 2021); Williams et al. (2009). Attention-heavy workloads and long-context serving stress memory systems via KV-cache and intermediate activations, pushing energy back toward DRAM/HBM movement Vaswani et al. (2017); Kwon et al. (2023); Horowitz (2014). In edge NPUs, power is constrained by System-on-Chip (SoC)-level thermal limits and shared memory bandwidth; as a result, sustained performance-per-watt depends on compiler scheduling, operator coverage, and quantized datapaths Jacob et al. (2018); Mazumder et al. (2021). When sparsity or conditional execution is present (e.g., MoE), energy efficiency can suffer due to reduced reuse and load imbalance unless the stack explicitly supports these patterns Shazeer et al. (2017); Gale et al. (2019). For TinyML deployments, adaptive inference techniques such as early-exit ensembles can improve energy efficiency by trading off compute against model uncertainty, providing opportunities for runtime model monitoring and dynamic resource allocation Ghanathe & Wilton (2025).

### 3.1.3 ASIC-specific energy challenges

ASICs can minimize energy by hardwiring data reuse, choosing an energy-efficient dataflow, and provisioning large on-chip Static Random Access Memory (SRAM) to reduce DRAM access Chen et al. (2016; 2014); Horowitz (2014). However, energy efficiency can degrade when models include unsupported operators (e.g., attention variants or custom normalization) that force host fallback or additional data marshaling Vaswani et al. (2017); Chen et al. (2016). Compression and sparsity reduce weight/activation traffic, but the benefits depend on sparsity structure; irregular sparsity introduces indexing overhead and load imbalance that can negate energy savings Han et al. (2016a;b); Gale et al. (2019). Designs closer to sensors (e.g., vision-centric accelerators) reduce I/O energy, yet they are specialized and may not generalize to broader model families Liu et al. (2015).

### 3.1.4 FPGA-specific energy challenges

FPGAs can be energy-efficient when a streaming pipeline eliminates intermediate DRAM writes and uses custom precision to reduce switching activity and memory traffic Umuroglu et al. (2017); Ma et al. (2018); Horowitz (2014). However, the FPGA landscape spans a wide power range: low-end edge devices (e.g., Xilinx Zynq, Intel Cyclone) operate at 1–5 W and can deliver compelling TOPS/W for quantized CNN inference in always-on or battery-constrained settings, while high-end datacenter FPGAs (e.g., Xilinx Alveo U280, Intel Stratix 10) consume 75–225 W and compete with GPUs on throughput but not necessarily on energy efficiency Nurvitadhi et al. (2017); Boutros et al. (2024). In practice, performance-per-watt is sensitive to external memory bandwidth, on-chip BRAM/DSP utilization, and routing overhead; poor mapping can waste energy through underutilized pipelines and excess off-chip transfers Nurvitadhi et al. (2017); Venieris & Bouganis (2016). Transformer/attention-style workloads further increase bandwidth pressure and can reduce the energy advantage if buffering and fusion are insufficient Zhang et al. (2024a); Vaswani et al. (2017). In cloud FPGA deployments, host-device interfaces and multi-tenant overheads can dominate system energy unless amortized by sustained utilization and carefully engineered serving pipelines Fowers et al. (2018); Nurvitadhi et al. (2017).

### 3.1.5 LPU/LLM-serving energy challenges

For LLM serving, energy is tightly coupled to memory movement from KV-cache and to serving policy (batching, concurrency, and context length), not just to matmul efficiency Kwon et al. (2023); Vaswani et al. (2017); Horowitz (2014). IO-aware attention kernels reduce memory traffic and can improve energy per token, but they must be integrated with runtime memory management to realize system-level gains under concurrency Dao et al. (2022); Kwon et al. (2023). LPU-style designs can reduce control and dispatch

overheads and target predictable execution, yet they still face the fundamental energy cost of sustaining high bandwidth per generated token Groq (2024); Williams et al. (2009). Heterogeneous serving (MoE, tool-calling) can further increase energy variability and complicate power provisioning due to conditional execution and load imbalance Shazeer et al. (2017); Gale et al. (2019).

### 3.1.6 In-/near-memory and analog energy challenges

Analog in-/near-memory approaches reduce data movement for matmul-heavy layers by performing multiply-accumulate in crossbars Shafiee et al. (2016); Chi et al. (2016); Ankit et al. (2019b), but ADC/DAC overheads, calibration, and device non-idealities can offset energy benefits at the system level Horowitz (2014). Energy efficiency is sensitive to mapping: if only a fraction of the model maps to crossbars, the overall system can still be dominated by digital compute and data conversion costs Ankit et al. (2019b). Attention-heavy models introduce softmax and KV-cache access patterns that are not naturally accelerated by crossbar matmul and can erode end-to-end savings Vaswani et al. (2017); Kwon et al. (2023). Consequently, practical designs often require heterogeneous integration and careful partitioning between analog and digital components Hennessy & Patterson (2018).

### 3.1.7 Neuromorphic energy challenges

Neuromorphic systems can achieve extremely low energy when activity is sparse and event-driven computation matches the workload, but benefits are sensitive to encoding/decoding overheads and to whether sparsity is realized in practice Davies et al. (2018); Orchard et al. (2021). When spike rates increase, communication and routing energy rises, and the event-driven advantage diminishes Davies et al. (2018). Mapping conventional dense networks to spiking representations can add conversion overhead and reduce effective sparsity, making comparisons with dense accelerators sensitive to modeling assumptions Gale et al. (2019); Coleman et al. (2017). As a result, neuromorphic energy wins are most robust in always-on sensing and temporal workloads where sparse events are intrinsic to the data Davies et al. (2018). It is important to note, however, that the spike-transport-dominated energy model is specific to *electronics-based* neuromorphic implementations. In alternative physical substrates—such as photonic neural networks, reversible computing, and thermodynamic processors—data can be copied and transported with negligible marginal energy cost after the initial encoding (RAM load, DAC conversion), because the underlying physics permits passive signal fan-out or energy-recycling operations Hamerly et al. (2019); Anderson et al. (2024); Melanson et al. (2025). This leads to fundamentally different big-$\mathcal{O}$ scaling of energy with respect to the number of operations: whereas digital and electronic-neuromorphic systems scale energy at least linearly with operation count, photonic and thermodynamic approaches can in principle achieve sub-linear energy scaling for large matrix-vector products, which is one of their primary theoretical attractions. Whether these advantages survive system-level integration (laser sources, detectors, thermal management) remains an active area of investigation.

## 3.2 Throughput, latency, and speed

Throughput and latency requirements impose different architectural and scheduling pressures. Datacenter inference serving may prioritize tokens-per-second at high utilization while still meeting strict end-to-end latency targets, whereas interactive applications care about single-request latency and worst-case behavior (often measured as p95/p99 latency). Achieving high throughput typically benefits from large batches and deep pipelining, but batching can increase latency and memory footprint, particularly for Transformers where KV-cache grows with sequence length and active requests Vaswani et al. (2017); Kwon et al. (2023).

For LLM inference, the critical path often alternates between compute-heavy projection/Multilayer Perceptron (MLP) General Matrix Multiplications (GEMMs) and bandwidth-heavy attention/KV-cache reads, as illustrated in Figure 7. Input/Output (IO)-aware attention kernels reduce memory traffic by tiling attention and avoiding materialization of large intermediate tensors, improving both throughput and latency under memory pressure Dao et al. (2022). Runtime techniques such as paging KV-cache help maintain throughput at high concurrency, but they introduce their own trade-offs in fragmentation, scheduling overhead, and tail

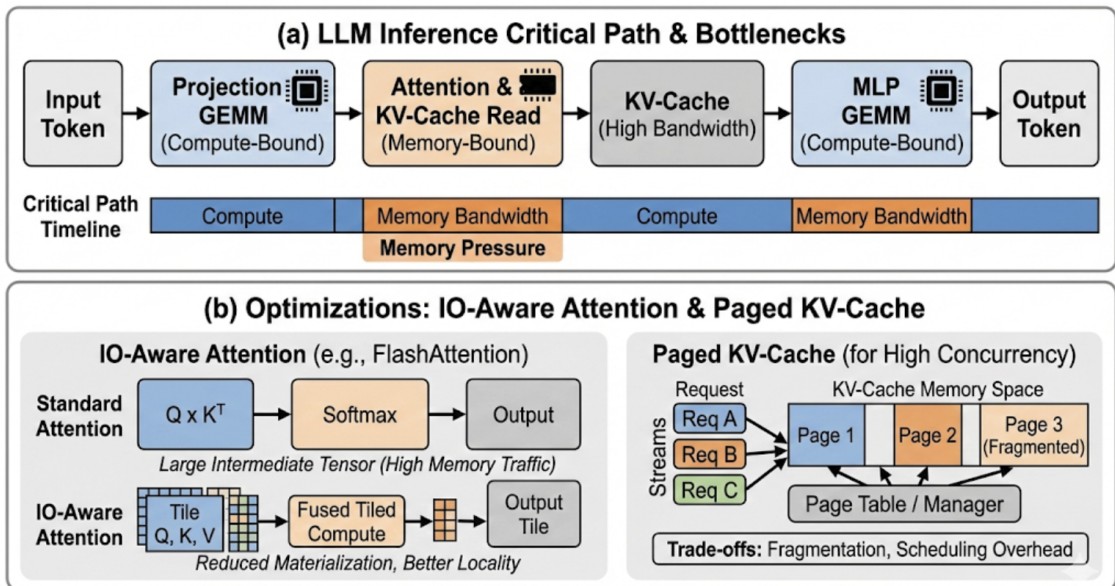

Figure 7: LLM inference bottlenecks and optimization strategies, differentiating between the compute-bound prefill phase and the memory-bandwidth-bound decode phase, and illustrating techniques like KV-cache management, paging, and attention optimization.

latency Kwon et al. (2023). Recent work explores hybrid strategies that balance KV-cache computation and loading to optimize memory bandwidth utilization Jin et al. (2025).

At the software level, kernel fusion and code generation reduce launch overhead and improve locality, which matters especially for small GEMMs and latency-sensitive inference. Libraries and compiler stacks (e.g., vendor primitives, graph compilers, and kernel Domain-Specific Languages (DSLs)) play a central role in realizing these optimizations in practice Chetlur et al. (2014); Chen et al. (2018b); Tillet et al. (2019). Consequently, performance claims should specify not only the hardware but also the kernel/library and compilation path used.

### 3.2.1 GPU-specific throughput/latency challenges

GPUs excel at throughput via batching, but tail latency can suffer when batching increases queueing delay and when small kernels incur launch overhead or synchronization Williams et al. (2009); Chetlur et al. (2014). Latency is also sensitive to kernel fusion and code generation quality; insufficient fusion increases memory traffic and amplifies launch overhead, especially for small GEMMs and normalization/activation chains Chen et al. (2018b); Tillet et al. (2019). For LLMs, efficient attention kernels reduce IO pressure, but predictable latency under concurrency requires careful KV-cache management and paging policies Dao et al. (2022); Kwon et al. (2023). Finally, sparsity and conditional execution can introduce latency variance due to load imbalance and irregular memory access Gale et al. (2019); Shazeer et al. (2017).

Achieving consistent p99 latency on GPUs is further complicated by the "bulk synchronous" programming model: a single long-running kernel can block other streams, making it hard to multiplex small, latency-critical tasks with background throughput-oriented work. Context switching on GPUs is relatively expensive, so preemption strategies for real-time serving are limited compared to CPUs. This forces serving systems to rely on cooperative scheduling (e.g., careful batching windows) or multi-instance GPU (MIG) partitioning to isolate workloads, though MIG partitions compute and memory statically rather than dynamically NVIDIA (2023); Williams et al. (2009).

### 3.2.2 TPU/NPU-specific throughput/latency challenges

Tensor processors often achieve high throughput on static dense kernels, but latency can degrade when shapes are dynamic or when compilation and layout choices are not well matched to the input distribution Jouppi et al. (2017; 2021). Attention-heavy inference introduces bandwidth-heavy phases and intermediate tensors that may not fit on-chip buffers, increasing latency variability unless the compiler and runtime manage tiling and memory explicitly Vaswani et al. (2017); Kwon et al. (2023); Dao et al. (2022). Edge NPUs frequently prioritize deterministic latency, but they are constrained by shared memory bandwidth and quantized operator support; when an operator falls back to the CPU/GPU, latency can spike Jacob et al. (2018); Mazumder et al. (2021). As model architectures evolve, maintaining predictable latency becomes a joint problem of operator coverage, compiler scheduling, and system integration Hennessy & Patterson (2018).

A key trade-off for TPU/NPU architectures is the "compile-time vs. run-time" balance. Relying on static compilation provides excellent determinism and dense packing for fixed graphs, but it can lead to severe latency penalties if dynamic shapes force recompilation or padding. For variable-length sequences (common in LLM serving), the system must either pad to the worst case (wasting throughput) or use specialized dynamic-shape compilers that may generate less optimal code. Managing this trade-off requires sophisticated runtime systems that can bucket requests by size or switch between pre-compiled binaries on the fly, adding complexity to the serving stack Jouppi et al. (2021); Kwon et al. (2023).

### 3.2.3 ASIC-specific throughput/latency challenges

ASICs can deliver low, predictable latency when the operator set is fixed and dataflow is streamed, but they may struggle with emerging operators (e.g., attention variants) and dynamic control, which can force inefficient fallbacks or additional data marshaling Chen et al. (2016); Vaswani et al. (2017). Even for supported kernels, latency depends on whether activations and weights fit on-chip buffers; otherwise off-chip bandwidth dominates and can introduce jitter Williams et al. (2009); Chen et al. (2016). Sparsity-aware ASICs can improve throughput, yet irregular sparsity can increase latency variability due to load imbalance and metadata handling Han et al. (2016a); Gale et al. (2019). In practice, achieving low tail latency often requires constraining the model/operator set or providing sufficient programmability to handle evolving kernels Hennessy & Patterson (2018).

Furthermore, the "throughput at what cost" question is acute for ASICs: maximizing peak Tera Operations Per Second (TOPS) often leads to large systolic arrays that have high startup latency and poor efficiency for small batches (batch-1 inference). To target low-latency serving, designers may split resources into multiple smaller cores or use finer-grained pipelining, but this complicates synchronization and on-chip interconnect design. If the workload is memory-bound (e.g., decode phase of LLMs), the ASIC's peak compute advantage becomes irrelevant, and latency is governed solely by the available HBM bandwidth and the efficiency of the memory controller Groq (2024); Williams et al. (2009).

### 3.2.4 FPGA-specific throughput/latency challenges

FPGAs can provide strong latency determinism through streaming pipelines, but achieving high throughput while maintaining low latency depends on memory bandwidth, pipeline depth, and host I/O overhead Nurvitadhi et al. (2017); Fowers et al. (2018). Designs that require frequent external DRAM access can lose determinism and become bandwidth-limited, especially for large feature maps or long sequences Williams et al. (2009). Transformer inference on FPGAs is particularly sensitive to buffering and fusion decisions because attention amplifies bandwidth pressure and increases intermediate tensor volume Zhang et al. (2024a); Vaswani et al. (2017). In multi-tenant deployments, additional latency variability can come from batching and scheduling at the service layer rather than from the FPGA datapath itself Fowers et al. (2018).

However, achieving this determinism requires resolving the "routing vs. logic" frequency wall. Deeply pipelined designs can run at high clock speeds, but complex control logic or irregular memory access patterns (e.g., from sparse attention) can create routing congestion that lowers the achievable frequency, directly hurting throughput. Moreover, if the model does not fit entirely on-chip, the FPGA must buffer and swap

weights/activations; if this swapping is not perfectly hidden by compute (double buffering), latency spikes occur. Modern FPGAs with HBM stacks help alleviate the bandwidth bottleneck, but maximizing their effective bandwidth requires wide data paths and careful memory controller tuning, which complicates the High-Level Synthesis (HLS)/Register Transfer Level (RTL) design effort Nurvitadhi et al. (2017); **?**.

### 3.2.5 LPU/LLM-serving throughput/latency challenges

LLM-serving accelerators must optimize for end-to-end token latency and throughput simultaneously under variable request patterns; paging and scheduling policies can dominate tail latency even when compute kernels are efficient Kwon et al. (2023). Attention kernels and KV-cache access create a mix of compute- and bandwidth-bound phases that are difficult to keep balanced under fluctuating concurrency Vaswani et al. (2017); Dao et al. (2022). LPU-style designs aim to reduce dispatch overhead and improve predictability, but they remain constrained by KV-cache bandwidth and context-length growth, which set a lower bound on token latency at high concurrency Groq (2024); Kwon et al. (2023). Conditional execution (MoE, tool-calling) can also increase tail latency by introducing load imbalance and unpredictable per-token work Shazeer et al. (2017); Gale et al. (2019).

Ideally, serving systems want *throughput without latency degradation*, but batching inherently trades one for the other. Large batches improve GEMM efficiency and aggregate bandwidth utilization, but they increase the time each token spends waiting in queues or for other sequences to finish their step. This "batching capability gap" is where specialized architectures try to innovate: by managing control flow and dependency tracking in hardware (rather than GPU kernels + CPU driver), LPUs attempt to make fine-grained interleaved execution feasible, allowing high concurrency with lower queuing delays. Even so, the physical reality of moving KV-cache bytes means that scaling context length inevitably hurts latency unless memory bandwidth scales proportionally Groq (2024); Williams et al. (2009).

### 3.2.6 In-/near-memory and analog throughput/latency challenges

Analog accelerators can offer high throughput for large matmuls, but end-to-end latency can be limited by conversion overheads (ADC/DAC), calibration, and handling of non-matmul operators Shafiee et al. (2016); Ankit et al. (2019b). Latency is sensitive to the size and batching of matmuls: small or irregular GEMMs underutilize crossbars and amplify peripheral overheads Ankit et al. (2019b). Workloads with attention and dynamic control reduce the fraction of time spent in the matmul kernels where analog speedups apply and add additional latency for softmax and memory management Vaswani et al. (2017); Kwon et al. (2023). As a result, analog accelerators are often most compelling in pipelines where large dense layers dominate and where conversion can be amortized Chi et al. (2016).

A deeper latency challenge stems from the dataflow between analog and digital domains. If a network requires frequent normalization or activation functions that are best done digitally, data must be repeatedly converted and moved, killing the latency benefits of in-place compute. Pipelining these stages can hide throughput costs but increases pipeline depth (latency). Furthermore, writing weights to analog arrays (e.g., Resistive RAM (ReRAM) or Flash) is often slow and energy-intensive; this makes analog approaches less suitable for workloads requiring rapid model switching or dynamic LoRA adapters, restricting their "speed" advantage to static-weight inference scenarios Shafiee et al. (2016); **?**.

### 3.2.7 Neuromorphic throughput/latency challenges

Neuromorphic devices can achieve low latency for event-driven sensing when spikes are sparse, but throughput and latency advantages diminish as activity increases or when dense preprocessing is needed Davies et al. (2018); Orchard et al. (2021). Latency can be dominated by encoding/decoding between frame-based signals and spikes, and by routing congestion when event rates increase Davies et al. (2018). Mapping conventional models can add conversion overhead and reduce determinism, making end-to-end latency highly workload-dependent Gale et al. (2019). Consequently, neuromorphic advantages are most visible in tasks where event-driven representations are natural and where computation is sparse by design Davies et al. (2018).

Throughput in neuromorphic systems is also non-standard: it is often measured in "events per second" or "inferences per second" under a specific sparsity assumption. If the input data becomes dense (e.g., a complex visual scene), the event traffic can saturate the on-chip interconnect, causing spikes to be dropped or delayed, which degrades both accuracy and latency. This traffic-dependent latency profile makes it hard to guarantee real-time bounds compared to the predictable scan-line processing of a traditional accelerator. Thus, achieving high speed requires not just fast silicon but also algorithms that actively maintain sparsity and locality throughout the network execution Davies et al. (2018); Coleman et al. (2017).

### 3.3 Area and cost

Area and cost constraints shape what can be built and deployed at scale. Silicon area (in mm$^2$) is consumed by compute arrays, on-chip SRAM, Network-on-Chip (NoC) resources, and I/O Physical Layers (PHYs), and each choice impacts achievable frequency and yield. Increasing on-chip memory can reduce off-chip traffic and improve performance-per-watt, but it increases die area and may limit clock rate; conversely, relying on off-chip memory shifts the bottleneck toward bandwidth and increases energy Williams et al. (2009); Horowitz (2014).

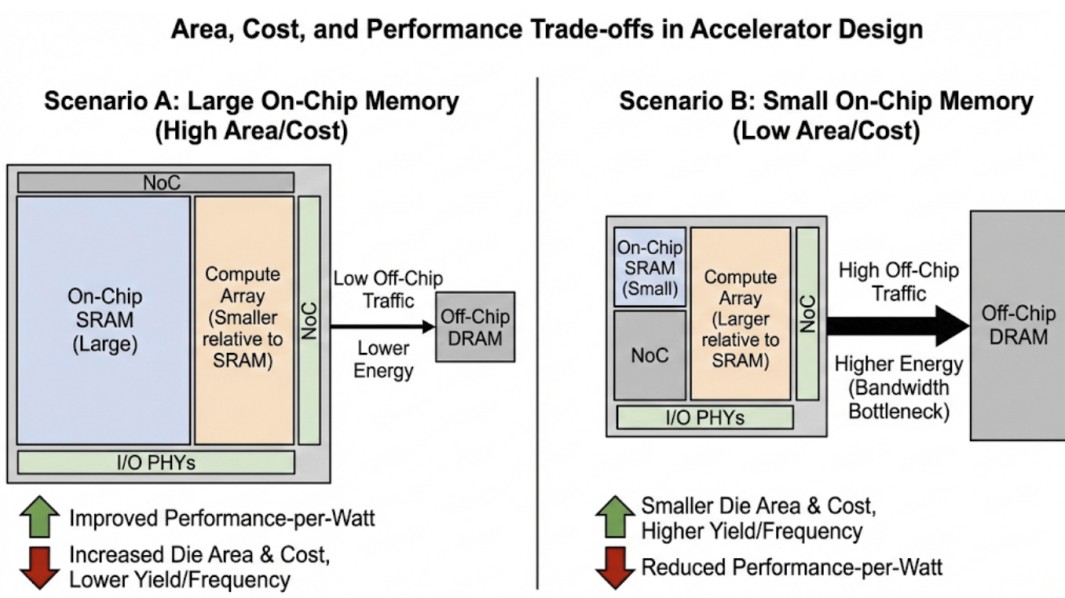

Figure 8: The fundamental trade-off between performance and silicon area/cost, illustrating how increasing parallelism and memory capacity improves throughput but raises manufacturing and packaging costs.

At the system level, packaging technologies and memory stacks (e.g., HBM) can improve bandwidth but raise cost and power, which affects total cost of ownership and practical adoption; Figure 8 illustrates the resulting performance–area trade-off. The "right" design point differs across environments: edge SoCs optimize for unit cost and battery life, while datacenters optimize for throughput-per-dollar and energy efficiency under high utilization. Domain-specific training systems demonstrate that large investments in interconnect and memory can pay off when workloads and utilization are stable at scale Jouppi et al. (2017; 2021).

Cost also includes engineering and opportunity cost. Highly specialized ASICs can be extremely efficient for a stable operator set, but they risk obsolescence as model architectures evolve. More programmable platforms (GPUs, FPGAs, and compiler-driven tensor processors) trade peak efficiency for adaptability, aligning with the broader trend toward hardware/software co-design Hennessy & Patterson (2018).

### 3.3.1   GPU-specific area/cost challenges

GPUs amortize high die area and advanced packaging costs across a broad software ecosystem, but cost-efficiency depends on utilization; under low batch or latency-constrained serving, throughput-per-dollar can drop sharply Williams et al. (2009). In many datacenters, GPU cost is coupled to the cost of power delivery and cooling, so improvements in performance-per-watt translate directly into operational savings Horowitz (2014). For training clusters, scaling also requires expensive interconnect and system integration, making system-level cost a dominant factor beyond the accelerator chip itself Hennessy & Patterson (2018). Software maturity further influences cost: improved kernel libraries and compilers can raise utilization without hardware changes Chetlur et al. (2014); Chen et al. (2018b).

A major driver of GPU cost is the memory subsystem: HBM stacks and CoWoS (Chip-on-Wafer-on-Substrate) packaging are significant fractions of the BOM (Bill of Materials). For inference workloads that are memory-capacity bound (like hosting LLaMA-70B), users must purchase more GPUs just to fit the model weights and KV-cache, even if compute utilization is low. This "capacity tax" has spurred interest in memory-expansion technologies (e.g., CXL) and aggressive quantization (4-bit/3-bit), which effectively reduce the dollar-cost per parameter by fitting larger models onto fewer devices. However, these solutions trade off bandwidth or accuracy, reinforcing the link between algorithmic choices and deployment economics Micikevicius et al. (2018); Kwon et al. (2023).

### 3.3.2   TPU/NPU-specific area/cost challenges

TPU-style systems devote substantial area to systolic arrays and on-chip SRAM, and they rely on specialized interconnects and memory systems to sustain utilization at scale Jouppi et al. (2017; 2021). These design choices can improve throughput-per-watt for stable dense workloads, but they require large capital investment in custom silicon and system infrastructure Hennessy & Patterson (2018). Edge NPUs optimize for SoC integration and unit cost, but they face strict area budgets and must share packaging and DRAM cost with the rest of the device, often limiting on-chip memory and making compiler scheduling critical Mazumder et al. (2021); Jacob et al. (2018). Operator coverage also impacts cost: if important kernels fall back to CPU/GPU, the effective cost per inference rises due to duplicated resources Vaswani et al. (2017).

For datacenter TPUs, the "pod" architecture amortizes the cost of the high-speed torus interconnect across many chips, but this makes the system expensive to deploy in small increments. The fixed topology is highly efficient for large-scale training jobs that map well to the torus, but it can be rigid for fragmented inference workloads or for models that don't naturally shard across the pod's dimensions. This creates a utilization challenge: if a job doesn't need a full pod slice, resources may be stranded. Thus, the total cost of ownership (TCO) depends on the scheduler's ability to bin-pack diverse jobs onto the available topology without fragmentation Jouppi et al. (2021).

### 3.3.3   ASIC-specific area/cost challenges

ASIC inference engines can be very area-efficient for a narrow operator set, but adding programmability, larger SRAM, or support for emerging operators increases area and design complexity Chen et al. (2016; 2014). Attention and LLM-serving workloads tend to demand more memory capacity and bandwidth, which can push ASIC designs toward larger SRAMs and more expensive packaging Vaswani et al. (2017); Kwon et al. (2023). Long design cycles and non-recurring engineering costs create risk when model architectures evolve rapidly, making ASIC viability tightly linked to workload stability and volume Hennessy & Patterson (2018). Compression and sparsity can reduce required memory, but they introduce metadata and control complexity that can increase area and verification cost Han et al. (2016a;b).

Furthermore, the "dark silicon" problem manifests in ASICs when new model features emerge that the hardware was not designed for. If an accelerator devotes 30% of its area to a specific sparsity engine or activation function that later falls out of favor (e.g., a move from ReLU to Swish/GELU), that area becomes wasted cost. This risk forces architects to over-provision programmable elements (vector processors or DSPs) alongside fixed-function units, diluting the area-efficiency advantage. As a result, the most successful

commercial ASICs often pair efficient matrix cores with a fairly capable general-purpose vector Instruction Set Architecture (ISA) to future-proof the investment Hennessy & Patterson (2018); Williams et al. (2009).

### 3.3.4 FPGA-specific area/cost challenges

FPGAs trade silicon efficiency for reconfigurability; Look-Up Table (LUT)/DSP/BRAM resources and routing overhead can limit peak throughput-per-area compared to ASICs Nurvitadhi et al. (2017). Cost-efficiency depends on whether the deployment can keep the FPGA well utilized; otherwise fixed platform cost dominates. However, for deployments that value flexibility or low-volume customization, FPGAs can reduce overall engineering cost and time-to-deployment, particularly in cloud settings where the same fleet can be repurposed Fowers et al. (2018). Toolchain productivity is also a cost factor: design iteration time and the ability to map new operators can dominate total cost of ownership Venieris & Bouganis (2016). Finally, partial reconfiguration and multi-tenant scheduling introduce additional complexity that can affect effective cost-per-inference Fowers et al. (2018).

High-end FPGAs with HBM and hardened NoCs are expensive, often rivaling GPUs in unit cost. To justify this premium, the design must deliver value that GPUs cannot—typically strictly deterministic latency or direct integration with network/storage ("smart Network Interface Card (NIC)" or "smart Solid State Drive (SSD)" use cases). If the FPGA is used merely as a slower GPU for batch processing, the area/cost math rarely works out due to the overhead of programmable fabric. Therefore, cost-effective FPGA deployment often involves moving the compute *to* the data (e.g., inside the network switch) to save system-level energy and bandwidth, rather than competing on raw FLOPs/$ in a rack server Fowers et al. (2018); Nurvitadhi et al. (2017).

### 3.3.5 LPU/LLM-serving area/cost challenges

LPU-style accelerators must justify cost through high utilization in LLM serving, where demand can be bursty and multi-tenant. Because serving is often memory- and bandwidth-limited, cost-efficiency depends on balanced provisioning of compute and memory bandwidth rather than on peak matmul throughput alone Kwon et al. (2023); Groq (2024); Williams et al. (2009). Long-context and high-concurrency workloads increase KV-cache footprint, pushing designs toward more memory capacity per accelerator, which can raise cost Vaswani et al. (2017); Kwon et al. (2023). In addition, cost depends on the software ecosystem: serving runtimes, kernel libraries, and compatibility with popular model formats influence time-to-deployment and operational overhead Kwon et al. (2023); Dao et al. (2022). Finally, heterogeneity (MoE, retrieval) can reduce utilization and increase cost unless supported in scheduling and memory management Shazeer et al. (2017).

A specific cost driver for LPU serving is the need for SRAM-heavy architectures to hide latency. SRAM is orders of magnitude more expensive per bit than DRAM. While an SRAM-centric design (like Groq's) provides unmatched deterministic latency and throughput for small batches, scaling it to store the weights of a 70B+ parameter model requires daisy-chaining many chips, exploding the system cost. This creates a bifurcation in the market: SRAM-based LPUs for latency-critical, lower-parameter (or sharded) serving, and HBM-based GPUs/ASICs for cost-optimized, high-capacity serving. The "right" cost choice depends entirely on the user's willingness to pay for milliseconds of latency reduction Groq (2024); Williams et al. (2009).

### 3.3.6 In-/near-memory and analog area/cost challenges

In-/near-memory designs shift area into memory arrays and mixed-signal peripherals. While crossbars can be dense, ADC/DAC and peripheral circuitry can dominate area and cost, and device variability can increase calibration and testing complexity Shafiee et al. (2016); Ankit et al. (2019b). Cost must also account for yield and reliability: analog non-idealities and endurance constraints can increase test time and reduce manufacturability Chi et al. (2016). Moreover, because not all operators map naturally to crossbars, practical systems may require additional digital logic, increasing area and integration complexity Vaswani et al. (2017). These factors mean that cost-per-inference depends on end-to-end mapping ratio and on the overheads of heterogeneous integration Ankit et al. (2019b).

Fabrication cost is another hurdle: many dense non-volatile memory technologies (Resistive RAM (RRAM), Phase-Change Memory (PCM)) require specialized process steps that may not be available in standard logic finFET nodes, or they require integration via expensive 2.5D/3D stacking. This splits the manufacturing ecosystem and can delay access to the latest lithography nodes that benefit the digital control logic. Consequently, analog accelerators often lag behind digital counterparts in frequency and logic density, forcing them to compete purely on energy efficiency in niche markets (e.g., edge, IoT) rather than replacing general-purpose datacenter silicon Hennessy & Patterson (2018); Chi et al. (2016).

### 3.3.7 Neuromorphic area/cost challenges

Neuromorphic chips allocate area to distributed memory and communication fabrics to support event-driven execution. They can be cost-effective for always-on low-power sensing, but their value depends on workload fit and toolchain maturity rather than on conventional throughput-per-area metrics Davies et al. (2018); Orchard et al. (2021). If the application requires significant preprocessing or dense computation, the effective cost rises because the neuromorphic chip must be paired with conventional processors Gale et al. (2019). In addition, software ecosystem and deployment tooling influence engineering cost, which is often the limiting factor for adopting non-standard execution models Hennessy & Patterson (2018). As a result, neuromorphic cost-efficiency is best evaluated in complete systems and application pipelines, not in isolation Coleman et al. (2017).

The area trade-off in neuromorphic designs is unique: they sacrifice dense arithmetic logic (large MAC arrays) to prioritize state storage (synapses) and fine-grained routing. For sparse workloads, this is area-efficient because inactive circuits consume little static power and no dynamic power. However, for dense workloads, the lack of time-multiplexed arithmetic units means the chip must physically instantiate more neurons/synapses to represent the model, potentially leading to a larger die size than a compact, reused systolic array. Thus, the "cost" of neuromorphic hardware is only justified when the sparsity factor is high enough to offset the lower area-density of computation Davies et al. (2018).

## 3.4 Performance limits from memory and communication

End-to-end performance is often limited by memory capacity, bandwidth, and communication rather than peak compute. Training requires storing activations, gradients, and optimizer state; for very large models, optimizer state and activations can exceed device memory, motivating memory-reduction techniques and sharding strategies Rajbhandari et al. (2020); Micikevicius et al. (2018). Inference for large language models requires managing KV-cache and intermediate tensors whose sizes depend on sequence length and batching, shifting the bottleneck toward memory systems even on compute-rich accelerators Kwon et al. (2023).

Figure 9 summarizes common memory and communication bottlenecks that dominate large-model training and serving at scale. Distributed execution introduces collective communication and synchronization that can dominate iteration time. Data parallel training stresses all-reduce bandwidth, while tensor and pipeline parallelism introduce more frequent communication of activations and gradients; the optimal strategy depends on model size, sequence length, and interconnect characteristics Shoeybi et al. (2019). Large-scale tensor-processor systems explicitly provision interconnect bandwidth and topology for these patterns, illustrating how system design shapes achievable scaling efficiency Jouppi et al. (2021).

The practical implication is that reporting peak FLOPs or TOPS is insufficient. Accurate performance modeling and diagnosis require accounting for arithmetic intensity, memory traffic, and communication volume, along with the ability to overlap communication with compute Williams et al. (2009); Wulf & McKee (1995).

### 3.4.1 GPU-specific memory/communication challenges

On GPUs, memory bandwidth and cache behavior frequently limit end-to-end performance, especially for attention and sparse workloads Williams et al. (2009); Dao et al. (2022); Gale et al. (2019). Long-context inference increases KV-cache footprint and can turn otherwise compute-heavy pipelines into bandwidth-bound workloads Kwon et al. (2023); Vaswani et al. (2017). Multi-GPU training adds all-reduce and synchroniza-

Figure 9: Memory and communication bottlenecks in large-scale training and inference, emphasizing the impact of limited HBM bandwidth, interconnect latency, and the overhead of collective communication primitives.

tion costs; scaling efficiency depends on overlap and on the chosen parallelism strategy (data vs. tensor vs. pipeline) Shoeybi et al. (2019); Rajbhandari et al. (2020). Compiler and runtime choices influence effective communication by determining fusion, scheduling, and overlap behavior Chen et al. (2018b); Abadi et al. (2016).

Specifically, the "memory wall" on GPUs is exacerbated by the growing disparity between arithmetic throughput (scaling rapidly with tensor cores) and memory bandwidth (scaling more slowly). This puts pressure on software to maximize cache reuse through techniques like activation checkpointing (trading compute for memory) and aggressive kernel fusion (keeping data in registers). In distributed settings, communication bandwidth (NVLink/Infiniband) often determines the feasibility of splitting a single model across devices. If the interconnect is slow, tensor parallelism becomes prohibitively expensive due to frequent all-reduce operations, forcing users into less efficient pipeline parallelism or pure data parallelism with lower effective batch sizes Shoeybi et al. (2019).

### 3.4.2 TPU/NPU-specific memory/communication challenges

TPU pods explicitly co-design interconnect and memory to match collective communication patterns of large-scale training, enabling higher scaling efficiency on stable workloads Jouppi et al. (2021; 2017). Nevertheless, workloads with dynamic shapes, irregular sparsity, or long-context attention can stress memory capacity and reduce utilization when intermediate tensors exceed on-chip buffers Kwon et al. (2023); Vaswani et al. (2017); Gale et al. (2019). Quantized inference can reduce bandwidth demand, but only when the operator set is fully supported and conversions are minimized Jacob et al. (2018). In edge NPUs, shared DRAM bandwidth and cache contention with other SoC components amplify memory bottlenecks, making placement and scheduling critical Mazumder et al. (2021).

A distinct challenge for TPUs is the management of "HBM fragmentation" and scratchpad allocation. The compiler must orchestrate data movement between HBM and on-chip memory with cycle-level precision. If a model's activation working set slightly exceeds the SRAM capacity, the compiler may spill data to HBM, causing a sharp "performance cliff." This binary behavior (fit vs. spill) makes performance less predictable than on cache-based architectures (GPUs), where performance degrades more gracefully. Consequently,

developers often spend significant effort manual tuning batch sizes and partition strategies to stay on the "fast path" of the memory hierarchy Jouppi et al. (2017); Williams et al. (2009).

### 3.4.3 ASIC-specific memory/communication challenges

ASIC accelerators often rely on large on-chip SRAM and carefully chosen dataflows to reduce DRAM traffic, but when model layers do not fit the assumed reuse patterns, off-chip bandwidth becomes the bottleneck Chen et al. (2016); Williams et al. (2009). Attention and KV-cache behavior can further increase streaming bandwidth requirements and introduce irregular access patterns that are difficult to support with fixed dataflows Vaswani et al. (2017); Kwon et al. (2023). Compression and sparsity can reduce memory footprint, yet they introduce metadata and irregular accesses that must be handled efficiently to avoid shifting the bottleneck to indexing and control Han et al. (2016a;b); Gale et al. (2019). In multi-accelerator ASIC systems, interconnect and synchronization must be provisioned for the chosen parallelism strategy, otherwise scaling stalls on communication Hennessy & Patterson (2018).

The "bandwidth tax" of programmability also affects ASICs. If an architecture relies on a host CPU to issue commands for every tile or layer, the PCIe bus or command-queue latency can become a bottleneck, especially for small-batch inference. To avoid this, advanced ASICs implement autonomous command processors or graph executors on-chip, allowing the device to run entire subgraphs without host intervention. However, this increases design complexity and requires a robust compiler to generate the command streams. Without this autonomy, the accelerator may sit idle waiting for the host to pointer-chase through the next set of descriptors, wasting the available DRAM bandwidth Hennessy & Patterson (2018).

### 3.4.4 FPGA-specific memory/communication challenges

FPGA designs can minimize external traffic with streaming pipelines, but bandwidth to external DRAM and host-device I/O can dominate performance for large models or attention-heavy workloads Nurvitadhi et al. (2017); Zhang et al. (2024a); Vaswani et al. (2017). For Transformers, buffering Query-Key-Value (QKV) projections and intermediate activations is challenging under limited on-chip BRAM, which can force frequent off-chip transfers and reduce throughput Zhang et al. (2024a); Williams et al. (2009). In cloud deployments, PCIe/host interfaces and multi-tenant scheduling can become communication bottlenecks, and the end-to-end system often determines observed performance Fowers et al. (2018). Toolchain decisions (HLS vs. RTL, memory partitioning) affect achievable bandwidth and hence the effective communication bottleneck Venieris & Bouganis (2016).

Another constraint is the "port limitation" of on-chip memory. BRAMs typically have only two ports. If a compute kernel needs to read 16 operands per cycle to keep the pipeline full, the designer must bank and duplicate memory extensively or run the BRAMs at a higher clock multiple (if possible). This memory partitioning problem is NP-hard and is a frequent cause of routing congestion or lower-than-peak performance. HLS tools try to automate this, but for complex access patterns (like sliding windows with non-unit strides or sparse lookups), manual pragma insertion or RTL rewriting is often needed to saturate the external memory interface Zhang et al. (2015); Canis et al. (2013).

### 3.4.5 LPU/LLM-serving memory/communication challenges

LLM serving is often limited by KV-cache capacity and bandwidth rather than compute, particularly at long context lengths and high concurrency Kwon et al. (2023); Vaswani et al. (2017). Paging and memory management policies are essential to scale concurrency, but they add overheads and can impact tail latency via fragmentation and scheduling decisions Kwon et al. (2023). IO-aware attention reduces intermediate traffic but does not eliminate KV-cache movement, so bandwidth-per-token remains a central system constraint Dao et al. (2022); Horowitz (2014). LPU-style designs must therefore provision balanced memory bandwidth per token and minimize control overheads, while maintaining compatibility with serving runtimes and model formats Groq (2024); Kwon et al. (2023).

Communication between chips is critical for LPUs because a single chip rarely holds a full large model. The interconnect must support extremely low-latency, fine-grained tensor slicing to allow "tensor-parallel"

execution across 16–64 chips without stall bubbles. Unlike GPU clusters that might tolerate microseconds of all-reduce latency, an LPU pipeline executing layer-by-layer requires nanosecond-scale synchronization to maintain its deterministic throughput guarantees. This drives LPU interconnects to be proprietary, statically routed, and integrated directly into the ISA, rather than relying on standard Ethernet or PCIe switches Groq (2024); Shazeer et al. (2017).

### 3.4.6 In-/near-memory and analog memory/communication challenges

In-memory compute reduces the cost of moving weights and activations for matmul, but system-level communication remains for non-matmul operators and for moving results between analog and digital domains Shafiee et al. (2016); Chi et al. (2016); Ankit et al. (2019b). The communication bottleneck can reappear in the peripherals (ADC/DAC) and in interconnect between crossbar tiles and digital control, particularly when workloads are small or irregular Ankit et al. (2019b). Attention workloads can remain bandwidth-bound due to KV-cache and softmax operations, which are not naturally accelerated by crossbar matmul Vaswani et al. (2017); Kwon et al. (2023). As a result, end-to-end speedups depend on mapping ratio and on how efficiently the heterogeneous system orchestrates data movement Hennessy & Patterson (2018).

Additionally, the "fan-out/fan-in" communication within a crossbar array poses signal integrity challenges. As arrays grow larger to hold more weights, the analog signal path lengthens (increasing IR drop and capacitance), which limits the readout speed or precision. Digital communication between tiles (e.g., aggregating partial sums from multiple arrays) then becomes the new bottleneck. If the NoC connecting these tiles is not provisioned with enough bandwidth, the fast analog cores will stall waiting to retire their results, negating the throughput benefit. Thus, PIM architectures effectively trade off-chip memory bandwidth problems for on-chip network design challenges Shafiee et al. (2016); **?**.

### 3.4.7 Neuromorphic memory/communication challenges

Neuromorphic systems distribute memory and communication across cores; performance depends on sparse event traffic and locality Davies et al. (2018); Orchard et al. (2021). When spiking activity is dense, communication fabric contention can dominate, and the energy/latency benefits of event-driven execution diminish Davies et al. (2018). End-to-end communication overhead also includes encoding/decoding between conventional sensors and spiking representations, which can outweigh benefits if sparsity is not intrinsic Gale et al. (2019). These effects mean that neuromorphic communication challenges are best evaluated at the application level rather than by peak metrics alone Coleman et al. (2017).

Scaling neuromorphic systems to large models introduces a "synaptic storage" challenge. While distributed SRAM is fast, it is not as dense as DRAM. Storing a billion-parameter model entirely in on-chip SRAM is cost-prohibitive. Some designs use multi-chip meshes, but this reintroduces inter-chip communication latency, which can disrupt the precise timing required for spike-based plasticity (Spike-Timing-Dependent Plasticity (STDP)). Therefore, efficient memory virtualization or hierarchical storage (local SRAM + backing DRAM) with smart prefetching is needed to support scale-out without killing the event-driven efficiency Davies et al. (2018).

### 3.5 Resource utilization and contention

Resource consumption is broader than power and area: it includes compute utilization, memory footprint, bandwidth demand, and contention for shared resources. Real systems are constrained by multiple shared bottlenecks (Streaming Multiprocessor (SM) occupancy, cache capacity, memory controllers, and interconnect), so maximizing one resource in isolation can degrade overall throughput by increasing contention elsewhere Williams et al. (2009).

As Figure 10 shows, irregular workloads—arising from unstructured sparsity, dynamic shapes, mixture-of-experts routing, and conditional execution—can lead to load imbalance and poor utilization even on highly capable hardware Gale et al. (2019); Shazeer et al. (2017). Sparse accelerators and sparse kernels can mitigate this, but they require careful data structures and scheduling to avoid turning arithmetic savings into indexing overhead and random memory access Parashar et al. (2017); Han et al. (2016a).

**Resource Utilization and Contention**

*Maximizing one resource (e.g., compute) can create contention for shared resources, ultimately reducing overall system performance.*

Figure 10: Resource utilization challenges arising from irregular workloads, showing how sparsity, dynamic shapes, and conditional execution (e.g., MoE) can lead to load imbalance and underutilized compute units.

Support for these workloads often depends on compiler/runtime capabilities. Graph compilers and kernel DSLs can generate fused kernels and choose layouts that improve locality and reduce contention, but dynamic behavior (variable sequence length, adaptive computation, MoE routing) remains challenging for ahead-of-time optimization Chen et al. (2018b); Tillet et al. (2019). As models become more heterogeneous, maintaining high utilization becomes increasingly a systems problem rather than a kernel-level problem.

### 3.5.1  GPU-specific utilization challenges

GPU utilization can collapse on small-batch or latency-constrained serving due to insufficient parallelism and kernel launch overheads, and it can degrade under unstructured sparsity due to load imbalance Gale et al. (2019); Parashar et al. (2017). Utilization is also sensitive to memory behavior: cache misses and bandwidth saturation can stall SMs even when compute resources are plentiful Williams et al. (2009). Compiler-driven fusion and kernel generation help mitigate overheads by reducing intermediate writes and launch count, but dynamic shapes and control flow remain challenging for ahead-of-time optimization Chen et al. (2018b); Tillet et al. (2019). For LLMs, paging and KV-cache layout decisions in the runtime can determine whether hardware utilization is sustained under concurrency Kwon et al. (2023).

Another utilization pitfall is the "tail effect" in distributed training. If one GPU in a 1000-GPU cluster is slow (due to thermal throttling, Error Correction Code (ECC) errors, or straggler tasks), the entire synchronous training step waits, effectively reducing the cluster-wide utilization to zero during the wait time. Fault tolerance and straggler mitigation strategies are thus essential for utilization at scale. Similarly, in pipeline parallelism, "pipeline bubbles" (idle time during fill/drain phases) reduce effective FLOPs. Minimizing these bubbles requires complex interleaved schedules (e.g., 1F1B) that increase memory pressure, creating a tension between utilization and memory capacity Shoeybi et al. (2019); Rajbhandari et al. (2020).

### 3.5.2  TPU/NPU-specific utilization challenges

TPUs/NPUs achieve high utilization on static dense kernels, but utilization can suffer when models include irregular operators, dynamic control flow, or sparse routing that breaks the accelerator's preferred dataflow Jouppi et al. (2017); Shazeer et al. (2017). Attention kernels with long contexts can also reduce utilization if intermediate tensors do not fit on-chip buffers and the execution becomes bandwidth-bound Vaswani et al.

(2017); Kwon et al. (2023). Compiler scheduling and operator coverage are therefore critical determinants of realized performance, both for datacenter TPUs and for edge NPUs Jouppi et al. (2021); Mazumder et al. (2021). Quantization improves throughput when supported end-to-end, but mixed precision and conversion overhead can reduce utilization if not carefully managed Jacob et al. (2018).

The rigidity of systolic arrays poses a utilization risk for non-GEMM operations. While matmuls run at near-peak efficiency, element-wise ops (activations, normalization) or reductions often run on separate vector units with much lower throughput. If a model is heavy on LayerNorm or Softmax (relative to matmul), these vector units become the bottleneck, leaving the massive systolic arrays idle. This "Amdahl's Law" effect means that as matrix engines get faster, the utilization bottleneck shifts aggressively to the vector/transcendental units and data-reshaping logic Jouppi et al. (2017); Williams et al. (2009).

### 3.5.3 ASIC-specific utilization challenges

ASICs can be highly utilized when the workload matches the designed dataflow, but utilization suffers when operator mix shifts (e.g., attention variants) or when data-dependent sparsity introduces irregularity Chen et al. (2016); Vaswani et al. (2017). Off-chip bandwidth limits can also reduce utilization when the intended reuse pattern breaks down, forcing MAC arrays to idle while waiting for data Williams et al. (2009); Chen et al. (2016). Sparsity-aware engines can help, but they require careful scheduling and metadata handling to avoid idle compute and to prevent indexing overhead from dominating Han et al. (2016a); Gale et al. (2019). Maintaining utilization over time often requires either a sufficiently expressive programming model or frequent redesign, reflecting the ASIC flexibility trade-off Hennessy & Patterson (2018).

A subtle utilization killer for ASICs is the "batching mismatch." If an ASIC is designed with a very wide vector width (e.g., 512-byte SIMD) to maximize peak TOPS, it requires a minimum batch size to fill those vectors. In real-time serving where batch size might be 1, the hardware effectively runs at a fraction of its capacity (e.g., 1/32th utilization). Techniques like "batch-1 optimization" or systolic arrays that stream weights (weight-stationary) can help, but they often require different on-chip buffering strategies than training-optimized designs. Thus, an ASIC built for training may show abysmal utilization in inference, and vice-versa Hennessy & Patterson (2018).

### 3.5.4 FPGA-specific utilization challenges

FPGA utilization depends on balancing pipelines, memory ports, and limited DSP/BRAM resources. Designs can become bandwidth-bound or routing-limited, and achieving high utilization across diverse models is difficult without reconfiguration or overprovisioning Nurvitadhi et al. (2017); Venieris & Bouganis (2016). For attention/Transformer workloads, limited on-chip buffering can reduce utilization by forcing frequent external memory access Zhang et al. (2024a); Vaswani et al. (2017). In cloud inference services, utilization is also shaped by batching and multi-tenant scheduling, which may be dominated by service-level constraints rather than by the FPGA datapath Fowers et al. (2018). Toolchain maturity affects achievable utilization because memory partitioning and pipelining decisions are often compiler/HLS-limited Venieris & Bouganis (2016).

Furthermore, "logic utilization" on FPGAs has a practical ceiling (often around 70-80%) before routing congestion makes timing closure impossible. A design that theoretically uses 95% of the DSPs might fail to route or run at a very low clock frequency, effectively reducing throughput. This routing overhead means that the "usable peak" is significantly lower than the "datasheet peak." Moreover, the time spent reconfiguring the FPGA (programming the bitstream) is dead time; in multi-tenant clouds, this reconfiguration latency discourages fine-grained time-sharing, leading to static partitioning that may leave regions underutilized Fowers et al. (2018); Williams et al. (2009).

### 3.5.5 LPU/LLM-serving utilization challenges

Serving workloads are bursty and heterogeneous (prompt lengths, concurrency), so utilization depends on dynamic batching and memory management rather than on steady-state kernel throughput Kwon et al. (2023). KV-cache placement and paging can determine whether the accelerator sustains high utilization or

becomes bandwidth-bound due to scattered memory access Kwon et al. (2023); Dao et al. (2022). Conditional execution (MoE, tool-calling) introduces additional load-balance challenges that can reduce utilization even on specialized serving hardware Shazeer et al. (2017); Gale et al. (2019). Achieving consistent utilization therefore requires co-design of the serving runtime, model architecture, and hardware scheduling policy Hennessy & Patterson (2018).

Specifically, the "prefill-decode imbalance" creates utilization gaps. During prefill, compute units are saturated; during decode, they are starved by memory bandwidth. In a naive system, the compute logic sits 90% idle during the long decode phase. Mixed-phase scheduling (running prefill for request A while request B is in decode) can recover some utilization, but it requires sophisticated resource isolation to prevent the prefill burst from destroying the decode latency Service Level Objective (SLO). LPU architectures often include specialized scheduling hardware to manage this mixing without Operating System (OS) overhead, aiming to keep the matrix units fed even when request phases diverge Groq (2024); Kwon et al. (2023).

### 3.5.6 In-/near-memory and analog utilization challenges

Analog accelerators can be underutilized when the workload includes many small matmuls, frequent conversions, or substantial non-matmul compute. Achieving high utilization often requires batching and mapping that maximizes crossbar occupancy, which can conflict with low-latency serving Shafiee et al. (2016); Ankit et al. (2019b). Utilization also depends on whether weights remain resident in the arrays; if frequent remapping is needed, setup and calibration overhead can dominate Chi et al. (2016). Attention-heavy models reduce the fraction of time spent in large dense matmuls and therefore reduce analog utilization unless the system is redesigned around these operators Vaswani et al. (2017); Kwon et al. (2023). As a result, utilization is tightly coupled to model structure and to the partitioning between analog and digital components Hennessy & Patterson (2018).

The "mapping fragmentation" problem is severe for crossbars. If a layer's weight matrix does not perfectly align with the physical crossbar size (e.g., a 100x100 matrix on a 128x128 array), the unused cells (rows/columns) are wasted area and power. Unlike digital memories that can be packed, analog weights are spatially fixed. This leads to low effective utilization for layers with odd shapes or for the tail ends of large matrices. Solutions involve virtualizing the arrays or using complex interconnects to route inputs to sub-tiles, but these add overhead. Consequently, utilization is highest for uniform, large-model layers and drops for irregular, optimized architectures (like MobileNets) Shafiee et al. (2016); Chi et al. (2016).

### 3.5.7 Neuromorphic utilization challenges

Neuromorphic utilization depends on spike sparsity and event locality. When activity is dense, communication contention increases and the advantage of event-driven computation diminishes Davies et al. (2018); Orchard et al. (2021). Utilization can also be limited by the mapping of the network to the hardware fabric; poor mapping can concentrate spikes and create hotspots even when global activity is low Davies et al. (2018). Converting conventional models to spiking forms can reduce effective sparsity and therefore reduce utilization benefits Gale et al. (2019). Consequently, utilization must be evaluated jointly with the encoding, network design, and application workload Coleman et al. (2017).

Moreover, "temporal utilization" in neuromorphic chips can be low if the input events are bursty. The hardware must be provisioned for the peak event rate to avoid dropping spikes, but during quiet intervals, the dedicated routing and update logic sits idle. Unlike clock-gated digital logic where idle means zero dynamic power, neuromorphic circuits often have leakage or bias currents that persist. Thus, the effective utilization (useful work per unit time / provisioned capacity) can be quite low for sporadic signals, challenging the TCO justification unless the standby power is exceptionally well managed Davies et al. (2018).

### 3.6 Benchmarking and reproducibility

Fair evaluation remains challenging (Figure 11). Accelerator results are sensitive to software stack maturity (kernels, compilers, graph optimizers), numerical choices (precision and quantization), model variants, and serving policies. Even small changes in batch size, sequence length, or kernel fusion can change whether a

workload is compute-bound or memory-bound, leading to large swings in measured performance Williams et al. (2009); Kwon et al. (2023).

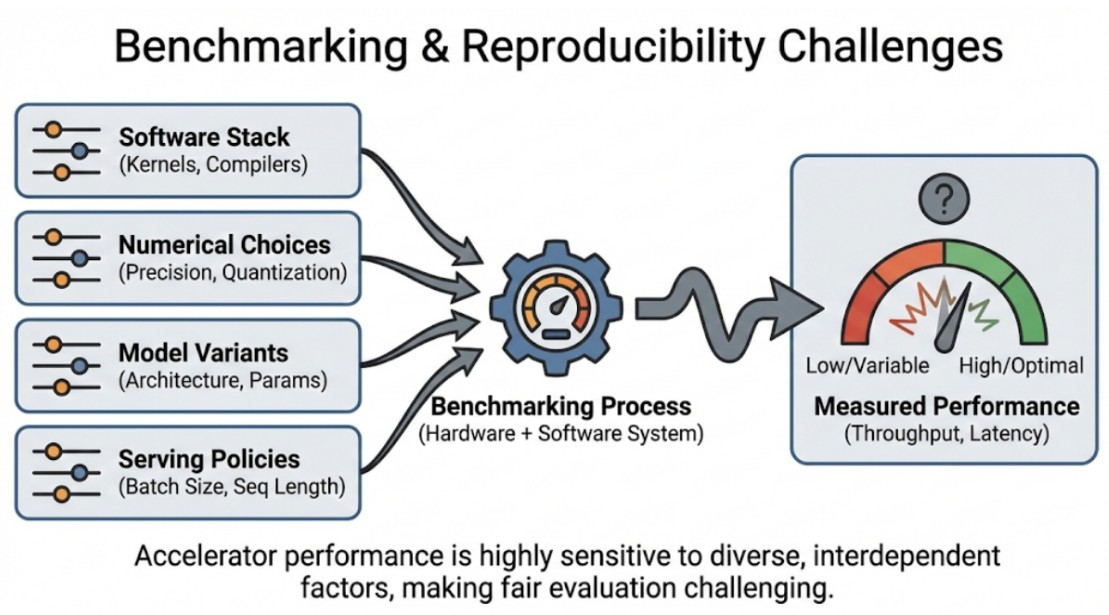

Figure 11: Benchmarking and reproducibility challenges in hardware acceleration, underscoring the sensitivity of results to software versions, compiler settings, model variants, and measurement methodology.

Benchmark suites such as MLPerf aim to standardize tasks, accuracy targets, and reporting rules for both training and inference, improving comparability across vendors and platforms Mattson et al. (2020). However, representative evaluation still requires careful choice of workload mix: production systems often run diverse models, experience non-stationary request distributions, and care about tail latency and cost, not only peak throughput. End-to-end benchmarks and competitions (e.g., time-to-train style evaluations) complement kernel microbenchmarks by capturing the interaction between the model, the input pipeline, the optimizer, and the distributed system Coleman et al. (2017).

Reproducible comparisons therefore require clear reporting of hardware configuration, software versions, workloads, accuracy targets, compilation settings, and measurement methodology. For inference, reporting should include batch/concurrency, sequence lengths, and latency percentiles; for training, reporting should include scaling strategy, communication overlap, and convergence criteria. Without this context, results can be misleading even when individual measurements are correct.

### 3.6.1 GPU benchmarking challenges

GPU results are highly dependent on kernel/library versions and compiler paths (e.g., vendor libraries vs. compiler-generated kernels) Chetlur et al. (2014); Chen et al. (2018b); Tillet et al. (2019), and on serving policies such as batching and KV-cache management Kwon et al. (2023). IO-aware attention kernels can dramatically change the memory/computation balance, so benchmarks should explicitly report attention implementations and sequence-length distributions Dao et al. (2022); Vaswani et al. (2017). MLPerf provides standardized reporting, but reproducing results still requires careful control of software versions, compilation flags, and measurement methodology Mattson et al. (2020). For training, reporting should include scaling strategy and memory optimizations (e.g., ZeRO) that change communication and memory footprints Rajbhandari et al. (2020); Shoeybi et al. (2019).

Furthermore, benchmarking often ignores the "warm-up" phase, Just-In-Time (JIT) compilation time, and memory fragmentation effects, which are critical in production. A benchmark that measures steady-state throughput after 1000 iterations hides the initial 30-second compilation delay or the gradual performance

degradation from heap fragmentation. For dynamic serving, measuring *sustainable* throughput (throughput at a fixed latency SLO) is more honest than peak throughput, but harder to standardize. Benchmarks should therefore report the "latency-throughput curve" rather than a single point, capturing the behavior under load Mattson et al. (2020); Kwon et al. (2023).

### 3.6.2 TPU/NPU benchmarking challenges

For TPUs/NPUs, performance depends on compilation, layout, and sharding decisions that may be opaque or highly tuned for specific workloads Jouppi et al. (2017; 2021). Benchmarking should report compiler settings and input distributions (shapes, sequence lengths), especially for attention-heavy models where KV-cache and intermediate tensor sizes drive memory behavior Vaswani et al. (2017); Kwon et al. (2023). In edge NPUs, benchmarks should also report operator coverage and fallback behavior, since CPU/GPU fallback can dominate latency and energy Mazumder et al. (2021); Jacob et al. (2018). MLPerf-style rules help, but reproducibility still depends on consistent compiler versions and system configuration Mattson et al. (2020).

The closed nature of some NPU software stacks adds a "black box" variability. An OS update or driver patch might change the NPU frequency governor or memory allocation policy, altering results significantly. Unlike open GPU kernels (e.g., in Triton or CUDA), NPU behavior is often hidden behind firmware blobs. This makes attribution difficult: did performance improve because of the hardware or because the compiler recognized a specific subgraph pattern? Fair benchmarking requires freezing not just the model but the entire firmware/driver image Mazumder et al. (2021).

### 3.6.3 ASIC benchmarking challenges

ASIC benchmarks must clearly specify operator coverage, precision, and how unsupported operations are handled (host fallback vs. accelerator implementation) Chen et al. (2016; 2014). Because ASIC efficiency is tightly tied to the chosen dataflow and memory hierarchy, reporting should include on-chip SRAM size, off-chip bandwidth, and any assumed reuse or tiling strategy Chen et al. (2016); Williams et al. (2009). Sparsity and compression results should report sparsity pattern, metadata overhead, and utilization to avoid misleading comparisons Han et al. (2016a;b); Gale et al. (2019). For modern attention-based workloads, benchmarks should clarify how softmax/KV-cache operations are handled, since these can dominate end-to-end behavior Vaswani et al. (2017); Kwon et al. (2023).

Additionally, ASIC power measurements are often reported at the "chip" level, excluding DRAM or host power. This can be misleading for memory-bound workloads where DRAM power is 30-50% of the total. A "system-level" benchmark (wall power) is necessary to compare an ASIC against a GPU fairly. Also, many research ASICs report simulation results rather than silicon measurements; simulation assumptions (e.g., zero DRAM latency, perfect clock gating) should be scrutinized. Benchmarks should explicitly state whether numbers are measured on silicon, emulated on FPGA, or estimated via architectural simulators Hennessy & Patterson (2018); Williams et al. (2009). To facilitate principled design-space exploration before silicon fabrication, several open-source frameworks have become widely adopted. Timeloop models the interaction between dataflow mapping and memory hierarchy to estimate throughput and energy for spatial accelerators Parashar et al. (2019); its companion tool Accelergy provides a modular energy-estimation framework that composes technology-level energy tables for arithmetic units, buffers, and interconnects Wu et al. (2019). SCALE-Sim focuses on systolic-array architectures and can rapidly sweep tile sizes, array dimensions, and dataflow choices to identify bottlenecks before RTL implementation Samajdar et al. (2018). Together, these tools enable researchers to report estimated performance with transparent assumptions, and reviewers to reproduce and challenge those estimates.

### 3.6.4 FPGA benchmarking challenges

FPGA results are sensitive to toolchain choices (HLS vs. RTL), frequency targets, and memory configuration, and they often depend on end-to-end pipeline integration (host I/O, streaming, reconfiguration) Nurvitadhi et al. (2017); Fowers et al. (2018). Reporting should include resource utilization (DSP/BRAM/LUT), achieved clock, and I/O constraints, because these determine whether the design is compute- or bandwidth-limited Venieris & Bouganis (2016); Williams et al. (2009). Transformer benchmarking should include se-

quence lengths and attention implementation details, since buffering and external memory access dominate performance Zhang et al. (2024a); Vaswani et al. (2017). In cloud settings, multi-tenant effects and batching policies can change both throughput and tail latency and should be reported explicitly Fowers et al. (2018).

A common pitfall is benchmarking only the "kernel execution time" on the FPGA, ignoring the PCIe transfer time to move data to/from the device. For small batches, this transfer dominates. Benchmarks must report end-to-end latency including all data movement. Also, precision matters: comparing an 4-bit Integer (INT4) FPGA implementation against an 16-bit Floating Point (FP16) GPU implementation is an apples-to-oranges comparison unless accuracy metrics are also provided. The "Pareto frontier" of Accuracy vs. Throughput/Watt is the only robust way to compare flexible precision FPGAs against fixed-precision logic Nurvitadhi et al. (2017); Umuroglu et al. (2017).

### 3.6.5  LPU/LLM-serving benchmarking challenges

LLM serving benchmarks must report latency percentiles, concurrency, prompt/response lengths, and cache policies because these dominate performance and cost in practice Kwon et al. (2023). "Tokens per second" without context can hide tail-latency regressions, memory-capacity limits, and instability under bursty load Mattson et al. (2020). Benchmarks should also specify attention kernels and paging strategy, since IO-aware attention and KV-cache management can fundamentally change memory behavior Dao et al. (2022); Kwon et al. (2023). When MoE or conditional execution is present, benchmarks should report load balance and routing behavior because these affect utilization and tail latency Shazeer et al. (2017); Gale et al. (2019).

Furthermore, the choice of "random" vs. "real" text inputs affects benchmarking for LLMs due to tokenization and sparsity. If a benchmark uses random noise, the tokenizer might produce a different number of tokens than for English text, skewing throughput numbers. If the model uses content-dependent sparsity (MoE or early exit), random data might trigger worst-case paths. Therefore, standard datasets (like ShareGPT or Alpaca) are preferred over synthetic tensors to capture realistic execution divergence and cache hit rates Mattson et al. (2020); Kwon et al. (2023).

### 3.6.6  In-/near-memory and analog benchmarking challenges

Analog and in-memory compute results must report accuracy under non-idealities, calibration procedure, ADC/DAC overheads, and the fraction of the model mapped to the accelerator Shafiee et al. (2016); Chi et al. (2016); Ankit et al. (2019b). End-to-end comparisons should include the cost of non-matmul operators and data conversions, because these can dominate when only part of the graph is mapped to crossbars Vaswani et al. (2017); Williams et al. (2009). Results should also clarify whether weights are assumed stationary and how often remapping occurs, as this affects both throughput and energy Ankit et al. (2019b). Finally, for attention-heavy workloads, benchmarks should state how softmax and KV-cache are handled, since these are not naturally accelerated by crossbar matmul Kwon et al. (2023).

The definition of "operation" in analog is tricky. Is a noisy, low-precision MAC equivalent to a deterministic digital MAC? Benchmarks should use "effective" throughput at a target accuracy (e.g., ImageNet Top-1) rather than raw TOPS. If the analog chip requires a larger model or retraining to regain accuracy lost to noise, that overhead must be penalized. A metric like "ISO-accuracy energy efficiency" is more meaningful than "peak efficiency at unknown accuracy" Chi et al. (2016); Jacob et al. (2018). To systematize these evaluations, circuit-/architecture-level simulators such as NeuroSim Chen et al. (2018a) and MNSIM Zhu et al. (2023) model crossbar array non-idealities (device variation, IR drop, ADC resolution), peripheral circuit overhead, and inter-array data movement, enabling researchers to project area, latency, and energy for a given network mapping under realistic analog assumptions. These tools are essential for closing the gap between idealized algorithm-level claims and silicon-realistic performance estimates.

### 3.6.7  Neuromorphic benchmarking challenges

Neuromorphic benchmarks must report spike encoding, activity sparsity, and application-level accuracy, since performance and energy benefits depend strongly on event sparsity and representation Davies et al. (2018); Orchard et al. (2021). Benchmarks should also report event rates and communication patterns, because

routing congestion can dominate latency and energy when activity increases Davies et al. (2018). Comparing against dense accelerators requires careful definition of equivalent workloads and latency/accuracy targets, and should account for preprocessing and encoding overheads Coleman et al. (2017); Gale et al. (2019). As with analog accelerators, end-to-end evaluation is essential: reporting only chip-level energy without the full pipeline can be misleading Mattson et al. (2020).

Finally, the "baseline" problem is pervasive: neuromorphic papers often compare against unoptimized CPU/GPU code. A fair comparison requires comparing against a highly optimized sparse-dense matrix engine or a dedicated low-power edge NPU running a quantized version of the same model. Benchmarking suites like "NeuroBench" attempt to standardize this by defining tasks where temporal dynamics are essential, ensuring that the event-driven nature is actually exercising a relevant capability rather than just running MNIST inefficiently Davies et al. (2018); Coleman et al. (2017).

# 4 Hardware Accelerator Architectures for Neural Networks

Neural-network acceleration has evolved into a heterogeneous ecosystem where different platforms optimize different points in the design space. At a high level, accelerators implement dense linear algebra efficiently (GEMM/conv), reduce data movement through carefully designed memory hierarchies and dataflows, and increasingly incorporate support for sparsity, mixed precision, and attention-centric kernels. This section surveys major architectural families and discusses how they are used for inference and training across common model classes (CNNs, RNNs, GNNs, and Transformers/LLMs) Williams et al. (2009); Vaswani et al. (2017); Jouppi et al. (2017). Figure 12 provides an overview of the neural-network acceleration pipeline and where architectural choices impact performance.

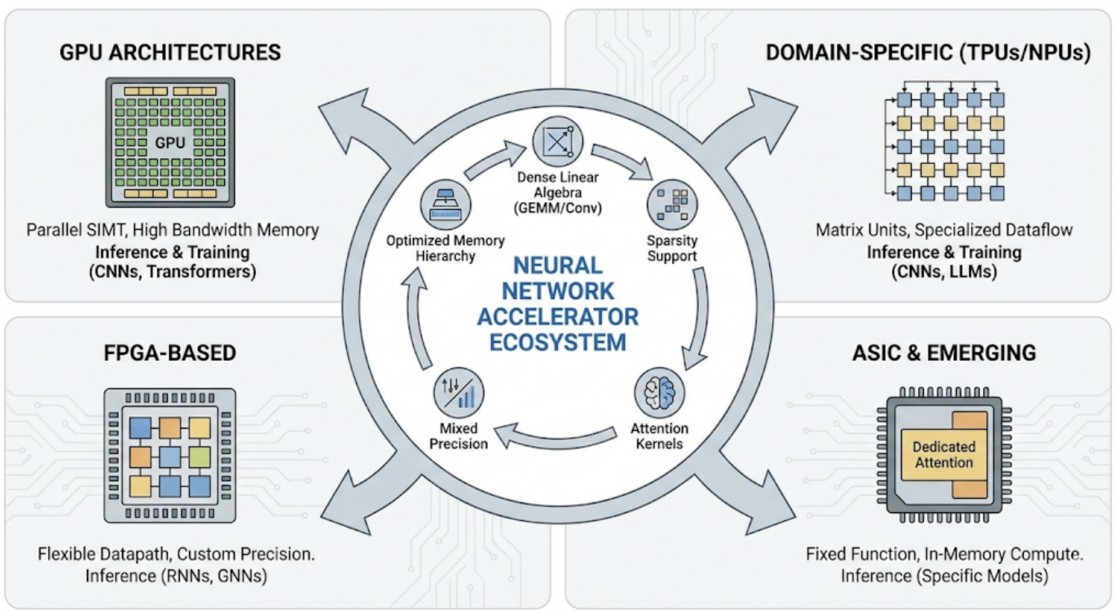

Figure 12: Overview of the neural-network acceleration pipeline, mapping the flow from high-level model definitions through graph compilation and optimization to execution on specialized hardware architectures.

## 4.1 GPU architectures

GPUs remain the dominant platform in many production and research settings due to their programmability, mature software ecosystem, and strong support for both training and inference. Architecturally, modern GPUs combine wide Single Instruction, Multiple Threads (SIMT) execution, deep multithreading to hide latency, and high-throughput memory systems (large on-chip caches and register files, plus high-bandwidth off-chip memory), together with specialized matrix-multiply units (tensor cores) that accelerate dense linear

algebra at reduced precision Williams et al. (2009); Micikevicius et al. (2018). In practice, GPU performance is inseparable from the software stack: operator libraries (e.g., highly tuned convolution and GEMM kernels), graph compilers, and kernel DSLs determine whether a model saturates compute or becomes bottlenecked by memory bandwidth, kernel launch overheads, or synchronization Chetlur et al. (2014); Chen et al. (2018b); Tillet et al. (2019).

GPUs are used across model families. CNNs map well to convolution/GEMM kernels with high reuse, while RNNs and sequence models expose different bottlenecks such as limited parallelism and recurrent dependencies. Transformers and LLMs place sustained pressure on memory bandwidth and capacity through attention and KV-cache, and they are sensitive to kernel fusion and memory-aware scheduling Vaswani et al. (2017); Kwon et al. (2023). As models adopt structured sparsity and mixture-of-experts routing, GPUs increasingly rely on sparse-friendly primitives and compiler support to preserve utilization under irregular access patterns Gale et al. (2019); Shazeer et al. (2017); Parashar et al. (2017).

### 4.1.1 Inference on GPUs

For inference, GPUs deliver high throughput by batching and by mapping convolutions and GEMMs onto highly optimized kernels. CNN inference benefits from implicit GEMM lowering, data layout transformations, and fusion of activation and normalization operations to reduce memory traffic. For Transformers and LLMs, inference efficiency hinges on fast attention and softmax kernels, fused MLP blocks, and careful KV-cache management; at high concurrency, KV-cache can dominate memory footprint and memory bandwidth Vaswani et al. (2017); Kwon et al. (2023); Dao et al. (2022). Kernel DSLs and compiler stacks are increasingly used to generate fused kernels and to adapt to rapidly changing operator mixes (e.g., different attention variants), which helps avoid fragmentation across hand-written kernels Tillet et al. (2019); Chen et al. (2018b).

Quantization is widely used to reduce bandwidth and improve throughput, particularly for edge-like deployment constraints and cost-efficient serving. Integer arithmetic and mixed-precision inference can be effective, but end-to-end gains depend on calibration, accuracy targets, and whether kernels can exploit the reduced precision without falling back to inefficient conversion paths Jacob et al. (2018). Sparsity and compression can further reduce memory traffic, yet unstructured sparsity can harm utilization without dedicated sparse kernels and scheduling that preserves coalesced memory access and load balance Gale et al. (2019); Han et al. (2016b).

### 4.1.2 Training on GPUs

Training emphasizes throughput, numerical stability, and scalability. Mixed-precision training has become a standard approach to improve performance while maintaining convergence, typically using 16-bit Floating Point (FP16)/Brain Floating Point (BF16) compute with 32-bit Floating Point (FP32) accumulation and loss-scaling or similar techniques Micikevicius et al. (2018). Memory footprint is a dominant constraint: activations, gradients, and optimizer states can exceed device memory for large models, motivating activation checkpointing and optimizer-state partitioning Rajbhandari et al. (2020), as well as distributed parallelism strategies (data, tensor, and pipeline parallelism Shoeybi et al. (2019); Jhoo et al. (2025); Gusak et al. (2025), and expert parallelism Shazeer et al. (2017)). These strategies trade off communication volume, synchronization, and pipeline bubbles against memory savings and compute efficiency.

At scale, training performance is often limited by communication and synchronization overheads (collectives for data parallelism; all-reduce and all-gather for tensor parallelism), making overlap strategies and interconnect bandwidth critical to end-to-end time-to-train Williams et al. (2009). As workloads evolve, compiler and runtime support becomes increasingly important for selecting numerically stable kernels, fusing memory-bound operations, and scheduling communication to hide latency behind compute Chen et al. (2018b); Abadi et al. (2016).

## 4.2 Domain-specific tensor processors (TPUs and NPUs)

Domain-specific accelerators pursue higher performance-per-watt by specializing the datapath and dataflow for dense linear algebra. A common pattern is a systolic-array-like matrix engine (or multiple such engines) coupled with on-chip SRAM and a compiler stack that performs graph compilation, operator fusion, and layout planning to match hardware constraints Jouppi et al. (2017); Kung (1982). Compared to GPUs, these processors often make stronger assumptions about operand shapes and memory access patterns, which enables higher utilization and better energy efficiency on the targeted kernels. In edge settings, NPUs are frequently integrated into SoCs, where they must operate under strict power, thermal, and latency envelopes and share memory bandwidth with CPUs/GPUs and other accelerators Mazumder et al. (2021).

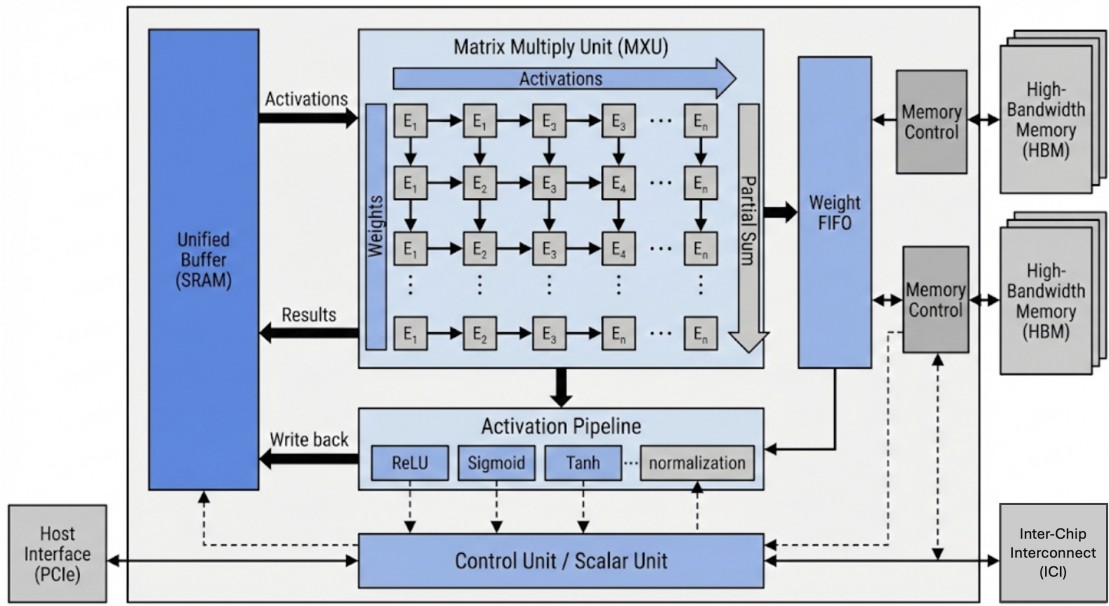

Figure 13: Representative TPU architecture featuring a large systolic array for dense matrix multiplication, coupled with unified on-chip buffers and a specialized control unit to maximize data reuse.

### 4.2.1 Inference on TPUs/NPUs

Figure 13 illustrates a TPU-style tensor-processor organization used to motivate compilation, tiling, and on-chip reuse. The "tensor processor" category includes both data center-class training systems (e.g., TPU pods with specialized interconnect) and embedded inference engines. Across both, the core architectural challenge is balancing compute with on-chip reuse: increasing MAC throughput is only beneficial if activations and weights can be delivered at sufficient bandwidth without thrashing the memory hierarchy. This is especially visible for attention-heavy models, where KV-cache and intermediate activations introduce large, streaming memory demands that may not match the accelerator's preferred dense reuse patterns Vaswani et al. (2017); Kwon et al. (2023). Inference on tensor processors benefits from predictable dense kernels and static shapes, enabling aggressive compilation, operator fusion, and on-chip reuse. Many deployments target integer arithmetic (8-bit Integer (INT8) and below) to reduce energy and improve throughput, and hardware often includes dedicated datapaths for quantized GEMM/conv and activation functions to avoid expensive conversions Jacob et al. (2018). For CNNs, static-shape inference allows compilers to pre-plan tiling and buffering to maximize weight/activation reuse; for RNNs, throughput is more sensitive to sequence length and batch size due to limited parallelism and recurrent dependencies.

Transformer inference introduces additional constraints because attention and KV-cache behavior can pressure memory. Even with efficient matmul engines, end-to-end latency can be dominated by memory movement and softmax-like operators, motivating compiler/runtime techniques that manage cache growth, mem-

ory fragmentation, and kernel fusion for attention blocks Vaswani et al. (2017); Kwon et al. (2023); Dao et al. (2022). When workloads include structured sparsity or MoE routing, utilization depends on whether the platform supports sparse-friendly data layouts and load-balanced scheduling Gale et al. (2019); Shazeer et al. (2017).

### 4.2.2 Training on TPUs/NPUs

Training on TPUs has been demonstrated at datacenter scale using large systolic arrays, high-bandwidth memory systems, and scalable interconnects designed for collective communication Jouppi et al. (2017; 2021). These systems rely on compilation to map graphs onto hardware efficiently, including partitioning, operator fusion, and layout transforms that maximize systolic-array utilization while minimizing off-chip traffic. Mixed precision is central for achieving high utilization and managing memory footprint, often in conjunction with numerically stable accumulation and loss-scaling techniques Micikevicius et al. (2018).

Distributed training on tensor processors introduces system-level trade-offs between parallelism strategy, communication volume, and memory footprint. Data parallelism stresses all-reduce bandwidth, while tensor/pipeline parallelism introduces more frequent synchronization and activation transfers; MoE adds routing and load-balance issues that can reduce utilization if not well matched to the interconnect and compiler Shazeer et al. (2017); Shoeybi et al. (2019). Overall, achieving high performance-per-watt requires co-design between the model (operator mix and shapes), the compiler (fusion, sharding, scheduling), and the hardware (on-chip memory and interconnect).

## 4.3 ASIC inference accelerators

ASIC inference accelerators focus on maximizing throughput-per-area and throughput-per-watt for fixed or semi-fixed operator sets. Many designs exploit dataflow specialization (e.g., weight-stationary, output-stationary, or row-stationary mappings), large on-chip SRAM to reduce DRAM traffic, and support for reduced precision, pruning, and compression Chen et al. (2016; 2014); Han et al. (2016b). Unlike GPUs, ASICs can hardwire control and buffering patterns to reduce overhead, but this comes at the cost of reduced flexibility when operators and data layouts evolve. Consequently, ASIC designs often target stable kernels (conv/GEMM) while providing limited programmability for control and scheduling Hennessy & Patterson (2018).

Architecturally, ASIC accelerators typically include arrays of MAC units, multi-bank on-chip SRAM scratch-pads, and an on-chip network designed to sustain the chosen dataflow. Some families emphasize bringing computation closer to sensors or memory to reduce I/O energy, while others target datacenter inference with high throughput at modest precision. As models evolve toward attention-heavy and mixture-based architectures, ASICs increasingly require either richer programmability or carefully chosen operator coverage to avoid falling back to a host CPU/GPU for unsupported kernels Vaswani et al. (2017); Shazeer et al. (2017). Importantly, limited operator coverage is less of a liability when accompanied by deliberate software/hardware co-design: if the compiler can decompose novel operators into sequences of supported primitives (e.g., tiled GEMM plus element-wise transforms), and the architecture provides a minimal programmable control path alongside its hardwired dataflow, the effective coverage can be much broader than the native instruction set suggests. Recent ASIC designs therefore tend to adopt a "wide dataflow core + thin programmable shell" strategy that preserves efficiency for the dominant kernels while accommodating operator evolution through compiler support Chen et al. (2016); Hennessy & Patterson (2018).

### 4.3.1 Inference on ASICs

CNN inference has been a primary driver of ASIC accelerators due to its structured computation and high reuse. Eyeriss introduced a spatial architecture and dataflow taxonomy aimed at minimizing energy by reducing data movement through careful mapping and local reuse Chen et al. (2016). Complementary designs such as the DianNao family explore microarchitectural specialization for neural operators, including support for common activation functions and efficient handling of weight storage and reuse Chen et al. (2014); Liu et al. (2015). Beyond dense compute, engines such as EIE demonstrate how exploiting compression and

sparsity can reduce memory bandwidth demand and improve energy efficiency by skipping zeros and storing compressed weights Han et al. (2016a;b).

Sparsity-aware ASICs can achieve large gains when sparsity is structured or when the architecture can avoid the overhead of irregular indexing. However, unstructured sparsity introduces load imbalance, irregular memory access, and reduced reuse, which must be handled by sparse data structures, scheduling, and often dedicated hardware support Gale et al. (2019); Parashar et al. (2017). For attention-based models, inference ASICs must also address memory-intensive KV-cache behavior and the growing prominence of non-matmul components (softmax, normalization, sampling), which complicates a purely matmul-centric datapath Vaswani et al. (2017); Kwon et al. (2023).

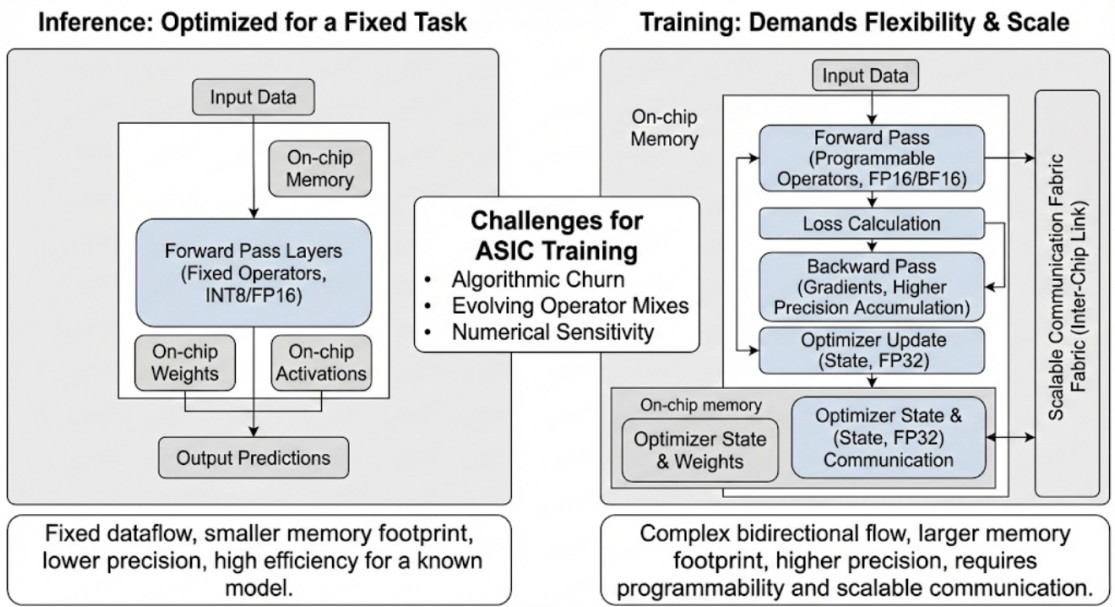

Figure 14: Challenges in designing ASICs for training, highlighting the need for higher precision, backward-pass support, and flexibility to handle evolving optimization algorithms compared to inference-only designs.

### 4.3.2 Training on ASICs

As illustrated in fig:section19, training on fixed-function ASICs is harder due to frequent algorithmic changes, evolving operator mixes, and the need for flexibility in numerical formats and optimizers. Compared to inference, training requires backward operators, optimizer updates, and larger memory footprints for activations and optimizer state; it is also more sensitive to numerical accuracy and stability, which limits the most aggressive quantization options Micikevicius et al. (2018). As a result, training-capable ASICs typically provide a broader set of supported operations, higher precision accumulation, and a scalable communication substrate.

Nevertheless, domain-specific training systems show that carefully chosen primitives (matmul, collective communication, and a sufficiently expressive compiler/runtime) can support a wide range of training workloads while delivering strong performance-per-watt Jouppi et al. (2021); Hennessy & Patterson (2018). The key is to co-design hardware capabilities with compiler abstractions so that new models can be expressed as compositions of supported primitives, and to ensure the memory system and interconnect are provisioned for the communication patterns of large-scale training.

### 4.4 FPGA-based accelerators

FPGAs offer reconfigurability and the ability to tailor datapaths, memory access, and I/O interfaces to a workload, making them attractive for edge inference, low-latency pipelines, and rapid architectural exper-

imentation. FPGA accelerators commonly use deeply pipelined streaming architectures, custom precision, and specialized memory layouts, and can integrate tightly with sensors and networking to reduce end-to-end latency Nurvitadhi et al. (2017); Umuroglu et al. (2017); Venieris & Bouganis (2016). Unlike GPUs, FPGAs allow designers to specialize the datapath for a fixed model (or model family) and to exploit determinism in the workload, which can yield strong energy efficiency when memory bandwidth is sufficient and the operator set is well covered.

The FPGA design space also includes cloud-scale deployments where reconfigurable logic is used as a shared inference service. In these settings, throughput and tail latency depend not only on compute pipelines but also on host-device interfaces, batching policies, and the ability to switch between models with minimal reconfiguration overhead Fowers et al. (2018); Nurvitadhi et al. (2017). Toolchains and abstraction layers (HLS, OpenCL, compiler-based mapping) play an outsized role in productivity and performance portability, influencing how quickly designs can adapt to new model operators and sparsity patterns Venieris & Bouganis (2016).

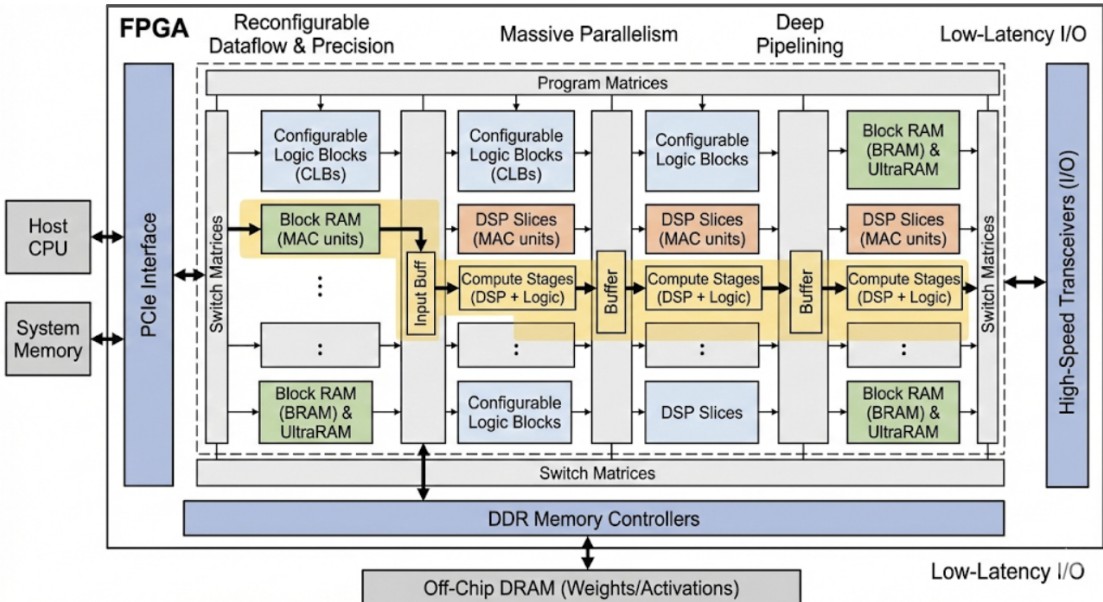

Figure 15: FPGA-based acceleration architecture illustrating a streaming datapath approach with custom buffering and I/O integration, designed to minimize off-chip memory access for deterministic inference.

### 4.4.1 Inference on FPGAs

Figure 15 sketches a representative FPGA-based acceleration setup used to discuss streaming datapaths, buffering, and I/O constraints. Inference on FPGAs has a long history of efficient CNN and Binarized Neural Network (BNN) implementations using quantization, folding, and pipeline parallelism Umuroglu et al. (2017); Ma et al. (2018). Designs typically trade flexibility for efficiency by fixing layer parameters and building a streaming pipeline that reuses on-chip buffers and minimizes off-chip traffic. When the model changes frequently or includes operators outside the accelerator's coverage, the benefits can diminish due to host fallback and data marshaling overhead Nurvitadhi et al. (2017); Venieris & Bouganis (2016).

Recent work also targets Transformer inference by accelerating attention, softmax, and feed-forward layers with structured sparsity patterns and custom compute engines Zhang et al. (2024a). Attention-heavy workloads stress memory bandwidth and require efficient buffering strategies for QKV projections and intermediate activations; they also benefit from kernel fusion to reduce intermediate writes. Compared to GPUs, FPGAs can deliver strong latency determinism and energy efficiency, but performance depends heavily on external memory bandwidth, operator coverage, and toolchain quality (HLS scheduling, memory partitioning,

and DSP utilization) Nurvitadhi et al. (2017). Recent work has also demonstrated the efficacy of FPGAs for accelerating Digital Twin learning and model recovery in mission-critical edge applications Xu et al. (2025e;c;b).

### 4.4.2 Training on FPGAs

Full training on FPGAs is less common due to limited on-chip memory, lower peak throughput than high-end GPUs, and challenges in supporting rapidly evolving training pipelines. Training requires storing intermediate activations and gradients, supporting backpropagation for a wide operator set, and performing optimizer updates, all of which can stress both FPGA resources and memory bandwidth. Mixed precision can help, but numerical stability and the complexity of implementing training kernels in reconfigurable logic remain barriers Micikevicius et al. (2018).

Hybrid approaches that offload selected kernels (e.g., matmul) or specialize for specific models and fixed training regimes can still be effective, particularly when the end-to-end pipeline benefits from tight I/O integration or when energy constraints dominate Nurvitadhi et al. (2017). Another pragmatic approach is to use FPGAs for inference and for hardware-in-the-loop workloads, while reserving training for GPUs/TPUs, which allows the FPGA design to focus on deterministic low-latency execution.

## 4.5 LLM-serving accelerators (LPUs and related designs)

Recently, specialized accelerators have emerged targeting the inference-serving regime of large language models. These systems often emphasize predictable execution, low-latency token generation, and high throughput under real-time constraints, with architecture and software co-designed around attention-centric workloads and memory management Groq (2024); Kwon et al. (2023). Recent surveys highlight that achieving "faster and lighter" LLMs requires a combination of algorithmic compression (quantization, pruning) and system-level optimization, reinforcing that hardware cannot be designed in isolation from the serving stack Chavan et al. (2024). This deployment regime differs from classical batch inference: it combines strict latency targets with fluctuating concurrency, diverse prompt lengths, and complex scheduling policies for multi-tenant serving. As a result, accelerator design must address not only peak GEMM throughput but also memory capacity, memory bandwidth, and the overheads of dynamic batching and KV-cache allocation Vaswani et al. (2017); Kwon et al. (2023).

From an architectural perspective, LLM serving is dominated by repeated execution of relatively small GEMMs (per token) and bandwidth-heavy memory accesses for KV-cache. This encourages designs that reduce dispatch overhead, favor predictable pipelines, and provide sufficient memory bandwidth per unit of compute. Software remains central: runtime systems that manage KV-cache paging, kernel fusion, and scheduling are often as important as the hardware datapath Kwon et al. (2023); Dao et al. (2022).

### 4.5.1 Inference on LPUs

Inference serving is frequently bottlenecked by memory capacity and bandwidth due to KV-cache growth, especially for long contexts and high concurrency Kwon et al. (2023), as contextualized in Figure 16. LPU-style designs aim to reduce scheduling overheads and improve predictability, but they still must manage fundamental trade-offs between batching (throughput) and latency, as well as between compute provisioning and memory-system design Groq (2024). Systems-level factors such as tokenization overhead, host-device transfer, and kernel launch granularity can materially affect tail latency, motivating end-to-end pipeline design rather than isolated kernel optimization Williams et al. (2009).

Another emerging factor is model heterogeneity: mixtures of experts, retrieval-augmented generation, and tool-calling introduce conditional execution and variable compute per token, which can reduce predictability and complicate scheduling. Serving accelerators must therefore either constrain the model interface (to preserve determinism) or provide sufficient flexibility in control and memory management to handle dynamic behaviors without collapsing utilization Shazeer et al. (2017); Gale et al. (2019).

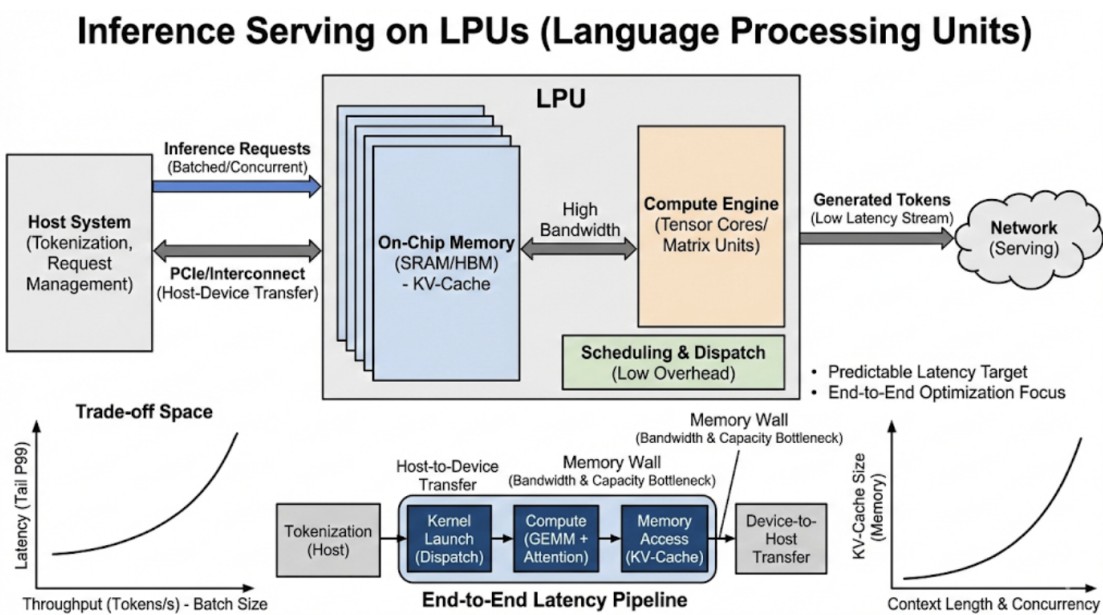

Figure 16: Inference on LPU-style accelerators, focusing on deterministic low-latency token generation for LLMs by managing control flow in hardware and optimizing memory access patterns for the decode phase.

### 4.5.2 Training implications

Most LPU-style systems primarily target inference rather than training. However, their emphasis on deterministic execution, low-overhead dispatch, and efficient attention kernels can inform training-system design (e.g., operator implementations and memory management), especially as training increasingly incorporates long-context sequences and MoE-style routing Shazeer et al. (2017). In particular, techniques that reduce attention memory traffic or that improve cache locality can benefit both inference and training, though training places additional demands on precision and backward-pass support Micikevicius et al. (2018).

More broadly, LLM serving highlights that memory is a first-class resource alongside FLOPs. Training systems already face similar pressures through activation storage and optimizer state, so ideas developed for inference memory management (paging, fragmentation control, layout-aware scheduling) may translate into improved training efficiency when combined with standard memory-reduction techniques Rajbhandari et al. (2020).

### 4.6 In-/near-memory and analog accelerators

To address the energy cost of data movement, in-/near-memory computing places computation closer to storage, often using crossbar arrays for analog matrix-vector multiplication (fig:section21). Such designs can offer high energy efficiency for dense linear algebra by exploiting physical laws for multiply-accumulate, but they face challenges in precision, device variability, ADC/DAC overheads, endurance, and system integration Shafiee et al. (2016); Chi et al. (2016); Ankit et al. (2019b). In addition, mapping modern networks is not only about matmul: normalization, attention/softmax, activation functions, and data-dependent control can erode the end-to-end benefit if they require frequent conversion between analog and digital domains Vaswani et al. (2017).

A key architectural question is where computation should occur relative to memory. Near-memory approaches may attach compute engines to DRAM/HBM stacks or to ReRAM banks, while in-memory approaches place compute directly inside the memory arrays. Both approaches attempt to reduce the energy per byte moved, but they trade this against increased complexity in data placement, programming models, and accuracy management Chi et al. (2016); Ankit et al. (2019b).

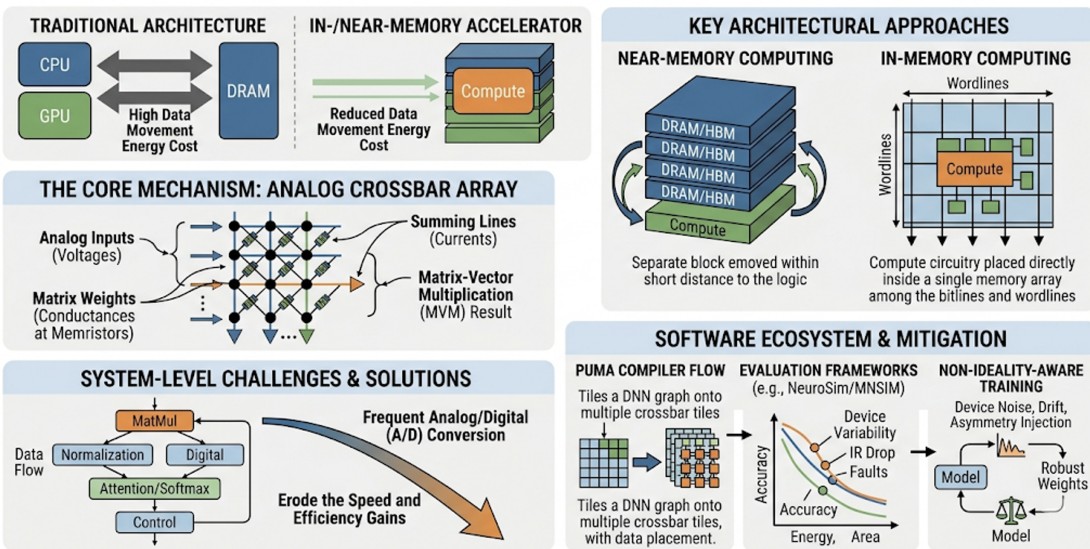

Figure 17: In-memory and near-memory computing for DNN acceleration. Analog crossbar arrays encode weights as memristive conductances to perform matrix-vector multiplication in a single step, reducing data movement overhead. System-level challenges—including frequent analog-to-digital conversion and device non-idealities—are addressed through compiler frameworks (PUMA), device-aware simulators (NeuroSim/MNSIM), and non-ideality-aware training.

Beyond hardware, the software ecosystem for PIM/analog accelerators is maturing. The PUMA compiler provides an ISA and compiler flow that maps DNN graphs onto pipelined crossbar-array tiles, handling tiling, data layout, and inter-tile communication automatically Ankit et al. (2019a). Circuit-level evaluation frameworks such as NeuroSim Chen et al. (2018a) and MNSIM Zhu et al. (2023) enable end-to-end accuracy–energy–area projections under realistic device models, including IR drop, stuck-at faults, and cycle-to-cycle variation. To mitigate analog non-idealities at the algorithm level, non-ideality-aware training injects device noise, conductance drift, and programming asymmetry into the forward and backward passes during software training, so that the learned weights are robust when deployed on physical crossbars Joshi et al. (2020); Rasch et al. (2023). These software advances are essential for closing the gap between idealized crossbar throughput and deployable system performance.

### 4.6.1 Inference

Analog in-memory accelerators are well matched to inference settings where weights are fixed and calibration can be amortized. Architectures such as ISAAC, PRIME, and PUMA demonstrate high throughput for convolution and fully connected layers when mapping is feasible and when the analog compute can be kept busy with sufficiently large matmuls Shafiee et al. (2016); Chi et al. (2016); Ankit et al. (2019b). Inference pipelines can also tolerate certain approximations if they preserve task-level accuracy, motivating quantization and algorithm-hardware co-design that accounts for analog noise and non-idealities Jacob et al. (2018).

However, end-to-end inference increasingly includes components that are not straightforward dense matmuls. Attention introduces softmax and KV-cache-like access patterns, and modern LLM serving often becomes memory bound even on highly optimized digital accelerators Vaswani et al. (2017); Kwon et al. (2023). These trends suggest that in-memory/analog accelerators must either broaden operator support or be integrated into heterogeneous systems where they accelerate the dominant matmul portions while other processors handle control-heavy components.

### 4.6.2 Training

Training with in-memory/analog hardware is more challenging because weight updates, gradient noise, and optimizer dynamics increase sensitivity to numerical error and device drift. The backward pass introduces additional operators and accumulation patterns that are less tolerant to analog noise, and frequent weight updates stress device endurance and calibration overhead. Research explores mixed-signal training strategies and algorithm-hardware co-design, but broad applicability across model classes and training regimes remains an open problem Hennessy & Patterson (2018).

Concretely, backpropagation on crossbar arrays maps onto three distinct phases: (1) the *forward pass* performs matrix-vector multiplication by applying input voltages to rows and reading column currents; (2) the *backward pass* computes error gradients via transposed reads on the same crossbar (or a duplicate array storing the transpose); and (3) the *weight update* programs conductance changes in situ using voltage pulses whose magnitude encodes the outer product of activations and errors Rasch et al. (2023). Because device programming is inherently noisy and asymmetric (SET and RESET responses differ), algorithms such as the Manhattan update rule replace continuous gradients with sign-based, fixed-magnitude pulses, improving robustness to programming variability at the cost of slower convergence Mackin et al. (2022). Mixed-signal dataflows interleave analog VMM with digital accumulation for partial sums, batch normalization, and non-linear activations, keeping precision-sensitive operations in the digital domain while exploiting crossbar parallelism for the dominant GEMM workload Joshi et al. (2020); Rasch et al. (2023).

One promising direction is hybrid training where analog crossbars accelerate the dominant GEMM/conv kernels while digital logic performs sensitive reductions and optimizer updates. Another is to use analog compute primarily for inference or fine-tuning regimes with constrained updates, while using conventional accelerators for full pretraining Micikevicius et al. (2018).

### 4.6.3 In-sensor computing

A related paradigm is *in-sensor computing*, which pushes computation even closer to the data source by embedding neural-network operations directly within or immediately adjacent to the sensing element itself (Zhou & Chai, 2020; Mennel et al., 2020). In vision applications, for example, processing-in-pixel (PIP) architectures integrate convolutional kernels within CMOS image sensor arrays, performing feature extraction before pixel data are ever read out to a separate processor (Hsu et al., 2023). This eliminates the analog-to-digital conversion and data-transfer bottleneck for the vast majority of redundant pixel data—often exceeding 90% in typical scenes—yielding substantial energy and latency savings for always-on edge inference. However, in-sensor designs face distinct constraints that separate them from the in-memory/near-memory accelerators surveyed here: pixel-area budgets severely limit per-pixel compute and storage, CMOS image sensor process nodes (typically 65,nm and older) lag behind logic nodes, and the tight coupling between sensing modality and compute architecture restricts generality across tasks. We consider in-sensor computing an important complementary direction but one whose design constraints—driven primarily by the sensor–processor interface rather than the memory–compute interface—place it outside the scope of this survey's accelerator taxonomy.

### 4.7 Neuromorphic accelerators

Neuromorphic processors explore event-driven computation and local learning rules, targeting sparse spiking workloads and ultra-low-power operation. Architecturally, they emphasize distributed local memory, spike-based communication, and asynchronous execution, which can be advantageous when activity is sparse and when always-on sensing requires strict energy budgets Davies et al. (2018); Orchard et al. (2021). However, these systems depart significantly from the dense linear algebra abstraction that dominates mainstream deep learning, so mapping conventional models typically requires conversion to spiking representations or training with spiking dynamics.

Intel's Loihi 2 Davies et al. (2018); Orchard et al. (2021) illustrates the design space. Fabricated in Intel 4, each Loihi 2 chip integrates up to 128 neuromorphic cores supporting up to 1 million neurons with programmable neuron models, graded (multi-bit) spikes, and a three-factor on-chip learning engine that

can implement spike-timing-dependent plasticity (STDP), eligibility traces, and reward-modulated learning. This programmability enables researchers to explore a wide range of spiking algorithms—including surrogate-gradient training—without off-chip weight updates, making Loihi 2 both a research platform and a prototype for low-power deployment. Efficient neuromorphic signal processing demonstrations on Loihi 2 have shown, for example, 47× lower output bandwidth for Short-Time Fourier Transform computation and over 90× fewer operations for optical flow estimation compared to conventional deep learning baselines Orchard et al. (2021). The architecture's emphasis on programmable learning engines over fixed datapaths mirrors the broader flexibility-versus-efficiency tension discussed in Section 4.3.

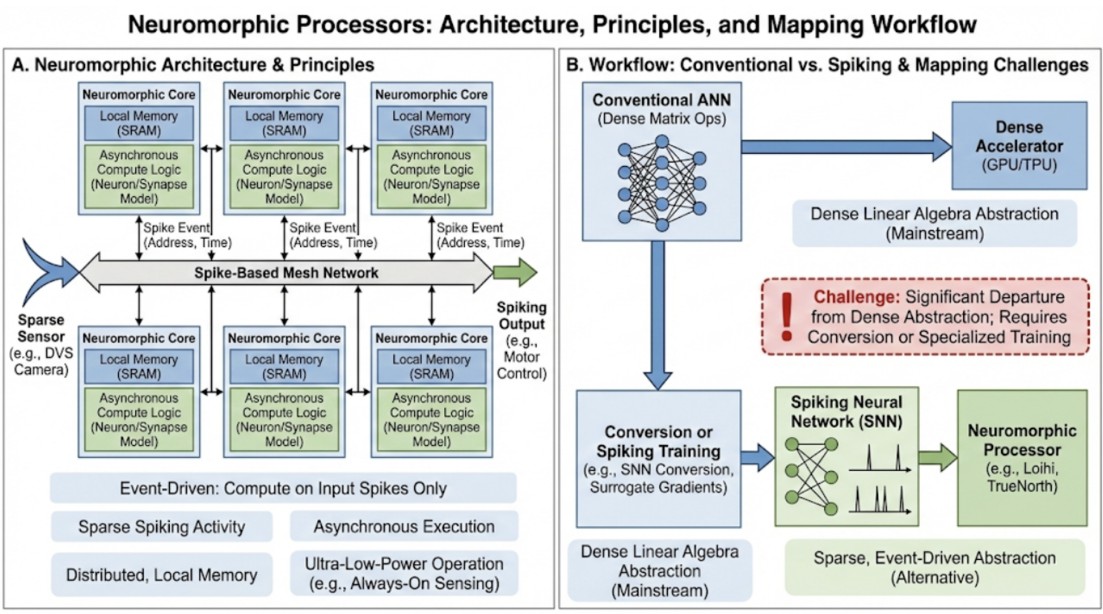

Figure 18: Neuromorphic accelerator architecture emphasizing event-driven processing, distributed local memory, and asynchronous communication to achieve high energy efficiency for sparse workloads.

IBM's TrueNorth (Akopyan et al., 2015; Merolla et al., 2014) occupies a complementary design point. Fabricated in 28 nm CMOS, it integrates 4,096 neurosynaptic cores—each containing 256 leaky integrate-and-fire neurons and a 256 × 256 binary synapse crossbar yielding 1 million neurons and 256 million synapses on a 5.4 billion transistor die consuming ~65 mW. The architecture is Globally Asynchronous Locally Synchronous (GALS): cores communicate via an asynchronous packet-switched mesh, so energy scales with spike activity rather than a global clock, achieving a power density roughly $10^4\times$ below conventional processors. Unlike Loihi 2, TrueNorth uses a fixed neuron model with no on-chip learning and no graded spikes, and requires a proprietary Compass/Corelet toolchain with no backward compatibility to standard compilers. This makes TrueNorth highly efficient for fixed inference tasks—IBM demonstrated $50 - 100\times$ speedups on real-time video recognition—but less adaptable to evolving algorithms, illustrating the flexibility, efficiency tension discussed in Section 4.3.

IBM's subsequent NorthPole chip (Modha et al., 2023) retains the brain-inspired principle of compute–memory co-location but abandons spiking entirely: 256 cores with 224 MB on-chip SRAM execute low-precision (2/4/8-bit) DNN inference, achieving 25× higher energy efficiency and 22× lower latency than comparable 12 nm GPUs on ResNet-50. Together, TrueNorth, Loihi 2, and NorthPole span a neuromorphic design spectrum from fixed-function ultra-low-power spiking (TrueNorth) to programmable on-chip learning (Loihi 2) to brain-inspired non-spiking inference competing directly with conventional accelerators (NorthPole).

Neuromorphic accelerators are therefore best viewed as a complementary platform rather than a drop-in replacement for GPUs/TPUs. Figure 18 depicts a representative neuromorphic organization used to motivate event-driven execution, local memory, and spike-based communication. They can excel on certain

temporal or sensory workloads, but their utility for large-scale dense models is limited by representation mismatch, toolchain maturity, and the cost of interfacing between event-driven and frame-based computation Davies et al. (2018). Beyond electronic spiking processors, the broader neuromorphic landscape includes photonic and thermodynamic substrates where data transport can be nearly free after initial encoding, yielding qualitatively different energy-scaling properties (see Section 3.1.7 for discussion) Hamerly et al. (2019); Anderson et al. (2024); Melanson et al. (2025).

### 4.7.1 Inference

Event-driven inference can be efficient when activity is sparse and latency constraints are tight, as computation and communication occur only on spikes. This can make neuromorphic systems attractive for always-on perception, keyword spotting, or certain control workloads where sparse events capture the essential dynamics. Mapping conventional deep networks to spiking representations typically requires conversion or training with spiking dynamics, and performance depends strongly on sparsity and the cost of encoding/decoding between representations Davies et al. (2018); Orchard et al. (2021).

Compared to dense accelerators, the primary performance metric is often energy per inference rather than peak throughput. As with other sparse approaches, benefits depend on whether sparsity is structured and predictable enough for the hardware to exploit without introducing large overheads for routing and buffering Gale et al. (2019).

### 4.7.2 Training

On-chip learning mechanisms exist for some neuromorphic systems, but training large-scale spiking models with competitive accuracy and efficiency remains an active research area. Learning rules may be local and biologically inspired, or they may approximate gradient-based updates, but bridging these approaches to mainstream training pipelines remains challenging. In many pipelines, neuromorphic devices are used primarily for inference, while training is performed offline on conventional accelerators Davies et al. (2018).

A practical implication is that neuromorphic systems often participate in heterogeneous workflows: GPUs/TPUs perform training and model development, and neuromorphic devices execute specialized low-power inference when the application and representation are a good match. Improving toolchains for conversion, calibration, and deployment is therefore central to broader adoption Davies et al. (2018).

## 5 Accelerator Approaches

Several machine learning approaches, such as artificial neural networks (ANNs), convolutional neural networks (CNNs), and recurrent neural networks (RNNs), are implemented on hardware. Figure 19 provides an organizing view of accelerator architectures across these workload families. This section discusses common accelerator approaches for each category, emphasizing how workload structure drives compute mapping, dataflow, and memory-system design Williams et al. (2009); Hennessy & Patterson (2018).

### 5.1 ANN acceleration (MLP and fully connected networks)

Classical feed-forward ANNs (e.g., multilayer perceptrons) are dominated by dense matrix multiplications (GEMMs) and elementwise activations. As a result, the primary accelerator approach is to maximize throughput on dense linear algebra while minimizing memory traffic:

- **Matrix-engine mapping**: map each layer to GEMM on tensor cores / systolic arrays / MAC arrays, typically using mixed precision (FP16/BF16 for training; INT8 and below for inference when accuracy permits) Micikevicius et al. (2018); Jacob et al. (2018); Jouppi et al. (2017).

- **Tiling and reuse**: block weights and activations so they can be reused from on-chip storage (registers/SRAM) rather than repeatedly fetched from DRAM/HBM Williams et al. (2009).

- **Fusion**: fuse activation and normalization chains with GEMM outputs to avoid materializing intermediate tensors in off-chip memory Chen et al. (2018b); Tillet et al. (2019).

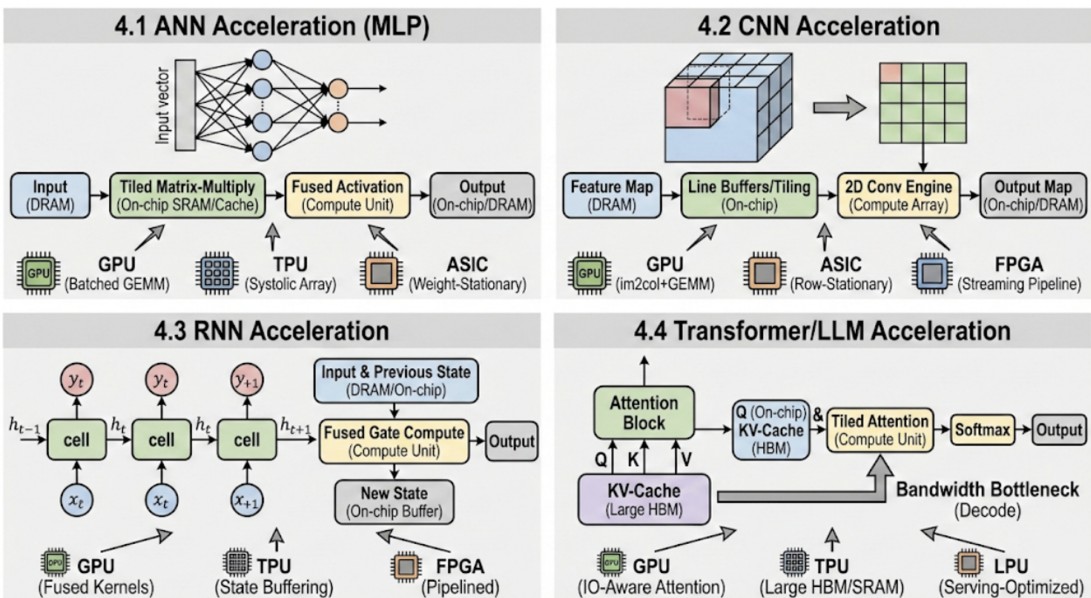

Figure 19: Taxonomy of accelerator architectures tailored for different neural network types (ANNs, CNNs, RNNs, Transformers), showing how workload characteristics drive specific hardware design choices.

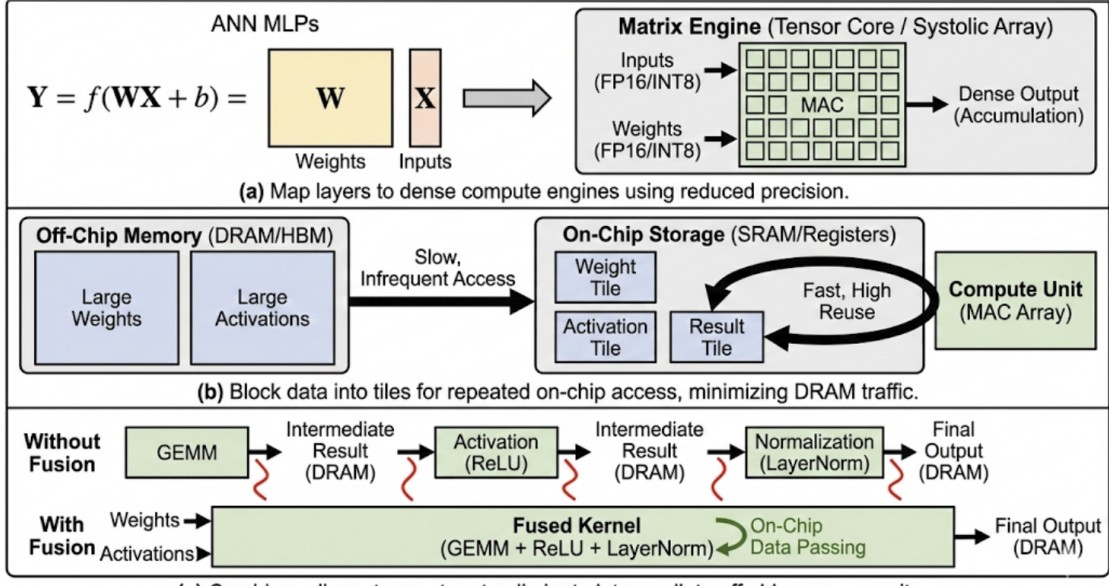

Figure 20: Standard Artificial Neural Network (ANN) structure used as a reference for acceleration discussions, featuring dense fully connected layers that benefit from matrix-multiplication optimization.

When models are small or batch sizes are low, launch/dispatch overhead and memory latency can dominate; Figure 20 provides a reference ANN structure for the optimization discussion. Thus, practical ANN accelerators rely heavily on compiler/runtime fusion and layout selection to sustain utilization Chen et al. (2018b); Tillet et al. (2019).

### 5.1.1 ANNs on GPUs

On GPUs, MLP layers are typically executed as a sequence of GEMMs where the software stack selects kernels, layouts, and epilogues (bias, activation, dropout) to reduce global-memory traffic and kernel launch overhead Chetlur et al. (2014); Tillet et al. (2019). A common approach is to fuse as much of the per-layer "tail" as possible into the GEMM epilogue so that intermediate tensors remain in registers/shared memory rather than being written out and reloaded Tillet et al. (2019); Chen et al. (2018b). For end-to-end networks, graph compilers further fuse adjacent pointwise operators and choose tensor layouts that improve locality and coalesced access, which can matter as much as peak GEMM throughput Chen et al. (2018b); Williams et al. (2009).

Training commonly uses mixed precision (FP16/BF16 compute with higher-precision accumulation) to increase effective throughput while maintaining numerical stability Micikevicius et al. (2018). Inference acceleration often focuses on quantized GEMMs (e.g., INT8) to improve throughput-per-watt and reduce memory bandwidth demand, provided the accuracy target allows Jacob et al. (2018). Quantization is most effective when it is applied end-to-end so that repeated conversions do not reintroduce memory traffic and overhead Jacob et al. (2018).

The dominant bottlenecks depend on regime. At large batch sizes and large hidden dimensions, the workload can approach compute-bound behavior; at small batch sizes, MLP inference becomes launch- and latency-dominated, making fusion and kernel granularity critical Williams et al. (2009); Tillet et al. (2019). Even when compute is ample, many MLP blocks are effectively bandwidth-bound because weights and activations must stream through the memory hierarchy every layer; reducing materialization (fusion), improving reuse (tiling), and selecting favorable layouts are therefore the primary "accelerator approach" on GPUs Williams et al. (2009); Chen et al. (2018b).

### 5.1.2 ANNs on TPUs/NPUs

TPU/NPU-style tensor processors accelerate ANNs by mapping GEMMs onto systolic arrays (or similar dense matrix engines) with compiler-managed tiling and on-chip SRAM reuse Jouppi et al. (2017); Kung (1982). The key approach is to treat the matrix engine as the "throughput core" and use the compiler to stage operands through on-chip buffers so the array receives a steady stream of tiles. This requires selecting blocking factors that match both SRAM capacity and the array geometry, and choosing layouts that minimize data reshaping between layers Jouppi et al. (2017); Williams et al. (2009).

Because many MLP layers are structurally similar, these platforms benefit from static-shape compilation: the compiler can pre-plan data movement, fuse epilogues, and schedule transfers to overlap memory with compute. When shapes are dynamic or when layers are too small, utilization drops because the array cannot be fully occupied, and overheads (padding, layout transforms) become visible in end-to-end latency Williams et al. (2009). In training settings, mixed precision plays a similar role as on GPUs: reduced-precision compute with higher-precision accumulation increases throughput while keeping convergence stable Micikevicius et al. (2018).

In edge NPUs integrated in SoCs, quantized inference (INT8 and below) is common because it reduces both compute energy and memory bandwidth. In practice, realized speedups depend heavily on *operator coverage* and avoiding costly format conversions or CPU fallbacks in the middle of the graph Jacob et al. (2018); Mazumder et al. (2021). The "accelerator approach" here is therefore as much about compilation and operator availability as it is about raw TOPS: keep the computation inside the NPU with a consistent layout/precision so that data movement does not dominate Williams et al. (2009).

### 5.1.3 ANNs on ASIC accelerators

ASIC inference engines accelerate fully connected layers using fixed or semi-programmable MAC arrays coupled with multi-bank on-chip SRAM scratchpads to keep weights and activations close to compute Chen et al. (2014; 2016). The "accelerator approach" is to hardwire a small set of dense primitives (GEMM and simple elementwise ops) and then organize the memory system so that most accesses hit local SRAM rather than DRAM. Compared to GPUs, ASICs can reduce control overhead (scheduling, instruction fetch,

dispatch) by constraining programmability and by specializing the datapath for the dominant operations Hennessy & Patterson (2018).

Dataflow choice is central. Weight-stationary mappings are natural for MLP layers because weights are reused across many activations within a batch; output-stationary mappings can reduce partial-sum movement when accumulation dominates. The architecture provisions buffering and an on-chip interconnect to sustain the chosen reuse pattern and to stream tiles through the MAC array efficiently Chen et al. (2016); Williams et al. (2009). When batch sizes are small, the ability to keep weights resident and to minimize DRAM traffic is still valuable, but the design must also avoid excessive underutilization due to limited parallelism.

Quantization and compression reduce footprint and bandwidth, but practical gains depend on whether the design supports efficient quantized datapaths and avoids repeated conversion or unpacking overhead Jacob et al. (2018); Han et al. (2016b). Unstructured sparsity can further reduce arithmetic, but it introduces indexing and load-balance overhead; benefits are largest when sparsity structure matches the hardware's scheduling and dataflow Gale et al. (2019). Overall, ASICs are most compelling when the operator set and deployment workload are stable enough that specialization remains useful over the product lifetime Hennessy & Patterson (2018).

### 5.1.4 ANNs on FPGAs

On FPGAs, ANN acceleration often uses spatial pipelines: a GEMM is implemented as a tiled matrix-multiply engine with customized precision, and data are streamed through compute stages to avoid intermediate DRAM writes Nurvitadhi et al. (2017); Venieris & Bouganis (2016). Rather than launching kernels, the design builds a fixed datapath where each cycle advances partial sums through a pipeline, enabling deterministic latency when the workload shape is fixed. Tiling controls how much parallelism is unrolled in hardware and how operands are buffered on-chip Williams et al. (2009).

The approach emphasizes (i) matching the compute pipeline to available DSP resources, (ii) partitioning on-chip BRAM to supply sufficient parallel memory ports, and (iii) selecting quantized formats that reduce bandwidth while meeting accuracy constraints Umuroglu et al. (2017); Jacob et al. (2018). Quantized and binary arithmetic can map efficiently onto FPGA resources, but end-to-end efficiency depends on keeping data in a consistent format and avoiding expensive packing/unpacking steps at stage boundaries Umuroglu et al. (2017); Jacob et al. (2018).

Performance is frequently bounded by external memory bandwidth and by the overhead of moving tensors across the host–FPGA interface in system deployments Nurvitadhi et al. (2017). Consequently, many effective FPGA ANN designs try to keep weights resident on-chip (when feasible), stream activations, and fuse adjacent elementwise operations into the datapath so that the pipeline remains compute-fed rather than I/O-limited Williams et al. (2009); Venieris & Bouganis (2016).

### 5.2 CNN acceleration

CNNs are characterized by structured spatial reuse: the same filters are applied across many positions, and adjacent outputs share overlapping input windows. Hardware acceleration approaches therefore focus on exploiting this regularity:

- **Convolution lowering and optimized kernels**: map convolution to direct conv kernels or GEMM-like forms, using carefully chosen data layouts and fusion of pointwise operators to reduce bandwidth Chetlur et al. (2014); Chen et al. (2018b).

- **Specialized dataflows**: designs often adopt weight-/output-/row-stationary dataflows to maximize reuse and reduce DRAM energy, supported by multi-bank on-chip buffers Chen et al. (2016).

- **Quantization and compression**: CNN inference commonly adopts INT8 (and lower) arithmetic; sparsity/compression reduce memory footprint but require kernels that preserve regular access and load balance Jacob et al. (2018); Han et al. (2016b); Gale et al. (2019).

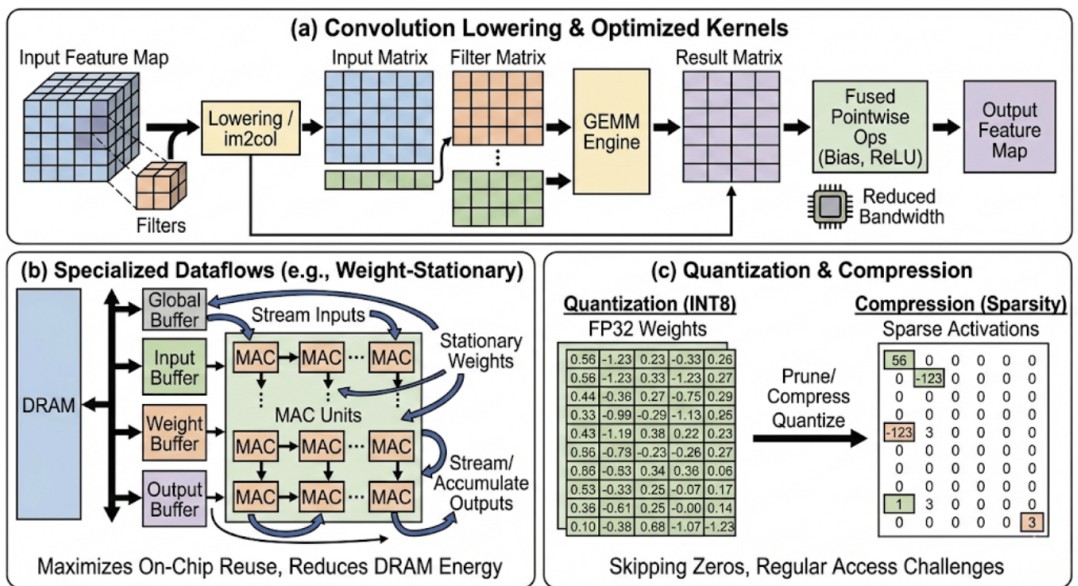

Figure 21: Convolutional Neural Network (CNN) structure highlighting spatial locality and weight reuse, which are exploited by hardware through specialized dataflows and buffering strategies.

Compared to generic dense layers, CNN accelerators can achieve high energy efficiency because reuse is predictable and amenable to spatial architectures and streaming pipelines (e.g., ASIC and FPGA designs); Figure 21 summarizes a representative CNN structure Chen et al. (2016); Umuroglu et al. (2017); Venieris & Bouganis (2016).

### 5.2.1  CNNs on GPUs

GPUs accelerate CNNs primarily through highly tuned convolution kernels and implicit GEMM mappings, where the software stack selects layouts (e.g., NCHW variants) and fuses pointwise ops to reduce memory traffic Chetlur et al. (2014); Chen et al. (2018b). In practice, convolution performance is shaped by data layout, tiling strategy, and how well the kernel uses on-chip memory (registers/shared memory/L2) to reuse input patches and filters. Vendor libraries provide many specialized kernels (for different strides, filter sizes, depthwise vs. standard conv), and compilers help select and fuse them into efficient end-to-end graphs Chetlur et al. (2014); Chen et al. (2018b).

Mixed precision is common for training to increase throughput and reduce activation bandwidth, while inference frequently uses quantized convolution/GEMM for efficiency Micikevicius et al. (2018); Jacob et al. (2018). Quantized inference is most effective when adjacent layers can stay in the same quantized format and when activation/weight scales are handled without repeated conversion, otherwise the conversion overhead and extra memory traffic erode gains Jacob et al. (2018). Sparsity and pruning can reduce arithmetic, but they require sparse-aware kernels; unstructured sparsity can introduce irregular access patterns and load imbalance that reduce utilization Gale et al. (2019).

CNN performance is often limited by memory movement when kernels are not fused or when feature maps are small (reducing arithmetic intensity), motivating graph-level optimization, operator fusion, and careful kernel selection Williams et al. (2009); Tillet et al. (2019). For latency-sensitive inference, another limiting factor is kernel launch and synchronization overhead across many small layers; fusing conv+activation+normalization chains and using compiler-generated kernels helps amortize this overhead and reduce intermediate materialization Chen et al. (2018b); Tillet et al. (2019).

### 5.2.2 CNNs on TPUs/NPUs

On TPUs/NPUs, convolutions are mapped to systolic arrays or dedicated convolution engines with compiler-selected tiling to maximize on-chip reuse Jouppi et al. (2017). The compiler chooses how to block input feature maps, filters, and output tiles so that partial sums and reused data remain on-chip across many MAC operations. This is particularly effective for standard CNN layers with regular shapes, where blocking and layout decisions can be planned ahead of time and reused across many inferences Williams et al. (2009).

Because CNN shapes are typically static in inference, compilers can pre-plan blocking and buffering to keep the array utilized and minimize off-chip traffic. For training, reduced precision compute (FP16/BF16) increases throughput in a similar manner as GPUs, but large activation tensors and gradient/optimizer state still stress memory capacity and bandwidth, making tiling and reuse central to efficiency Micikevicius et al. (2018); Williams et al. (2009). When models include less common operators or dynamic behavior, compilation may require additional layout transforms, which can increase memory traffic and reduce end-to-end performance.

Edge NPUs often target INT8 pipelines end-to-end; the key practical requirement is *operator coverage* (conv, depthwise conv, pooling, activations) so that the graph does not fall back to CPU/GPU and incur extra copies and latency Mazumder et al. (2021); Jacob et al. (2018). Realized performance therefore depends not only on the conv engine but also on memory-system constraints (shared DRAM bandwidth on an SoC) and on whether quantization is applied consistently without repeated conversions Jacob et al. (2018). When shapes deviate from expected regimes (very small feature maps, unusual channel counts) or when layers are too small, utilization can drop even if peak TOPS are high Williams et al. (2009).

### 5.2.3 CNNs on ASIC accelerators

CNN ASICs exploit convolution's regular reuse with spatial architectures and carefully chosen dataflows. Eyeriss demonstrated how row-stationary mapping and hierarchical buffering reduce energy by minimizing data movement, and similar ideas appear in many CNN inference engines Chen et al. (2016). The accelerator approach is to bind the convolution loop nest to a fixed dataflow that maximizes local reuse: keep either weights, partial sums, or input tiles stationary in local buffers while streaming the remaining operands through the MAC array Chen et al. (2016); Williams et al. (2009).

The hardware typically includes a MAC array, local buffers for weights/activations/partials, and an on-chip interconnect to sustain the chosen reuse pattern. DMA engines and buffering orchestrate streaming from DRAM, with the goal of turning expensive off-chip access into long, predictable bursts that can be amortized across many MACs. Because CNN layers vary in shape, practical ASICs may support multiple dataflow modes or configurable tiling so that both early (large feature map) and late (small feature map) layers run efficiently Hennessy & Patterson (2018).

Quantization (INT8 and below) is widely used to reduce memory footprint and energy, while pruning/compression can help when sparsity is structured enough to be exploited without large indexing overhead Jacob et al. (2018); Han et al. (2016b); Gale et al. (2019). Unstructured sparsity can reduce arithmetic but can also introduce irregular memory access and load imbalance; the energy/performance benefit therefore depends on having sparse-aware scheduling and data formats that keep accesses regular enough for the memory system Gale et al. (2019). Overall, CNN ASICs aim to trade flexibility for efficiency by specializing for the stable, regular operators that dominate CNN inference Hennessy & Patterson (2018).

### 5.2.4 CNNs on FPGAs

FPGA CNN accelerators often use streaming dataflows where feature maps flow through a pipeline of convolution/activation stages, avoiding intermediate off-chip traffic and providing deterministic latency Venieris & Bouganis (2016); Yan et al. (2024). A common pattern is to use line buffers and sliding-window datapaths that reuse input pixels across neighboring convolution windows, matching convolution's spatial locality. The pipeline can be "folded" to trade throughput for area by time-multiplexing a smaller compute engine across channels or layers, while still maintaining a streaming schedule Venieris & Bouganis (2016); Williams et al. (2009).

Quantized and even binary networks can map efficiently to FPGA LUT/DSP resources, enabling high throughput-per-watt when bandwidth is sufficient Umuroglu et al. (2017). In these designs, precision is a first-class architectural knob: fixed-point widths are chosen to reduce BRAM and external bandwidth while still meeting accuracy constraints, and keeping the entire pipeline in a consistent quantized format avoids conversion overhead Umuroglu et al. (2017); Jacob et al. (2018).

Practically, the design must balance DSP utilization, BRAM partitioning, and external memory bandwidth; otherwise the pipeline stalls on memory rather than compute Nurvitadhi et al. (2017); Williams et al. (2009). System-level bottlenecks such as host I/O (PCIe), buffering, and reconfiguration overhead also affect end-to-end performance in deployment settings, motivating designs that stream directly from input sources and fuse as many layers as feasible into a single datapath Nurvitadhi et al. (2017); Venieris & Bouganis (2016).

### 5.3 RNN acceleration

RNNs introduce temporal dependencies: hidden-state updates create a sequential critical path across timesteps. Acceleration approaches therefore combine dense-math specialization with techniques to reduce the overheads of recurrent execution:

- **Gate fusion**: fuse multiple gate computations (e.g., in Long Short-Term Memory (LSTM)/Gated Recurrent Unit (GRU)-style cells) into fewer kernels to reduce memory traffic and kernel-launch overhead, improving latency at small batch sizes Chetlur et al. (2014); Tillet et al. (2019).

- **Batching and sequence packing**: increase parallelism by batching sequences and packing variable-length inputs when latency constraints allow, trading off throughput against responsiveness Williams et al. (2009).

- **On-chip state buffering**: keep recurrent state and frequently reused parameters in on-chip SRAM/registers where possible to reduce round-trips to DRAM Williams et al. (2009).

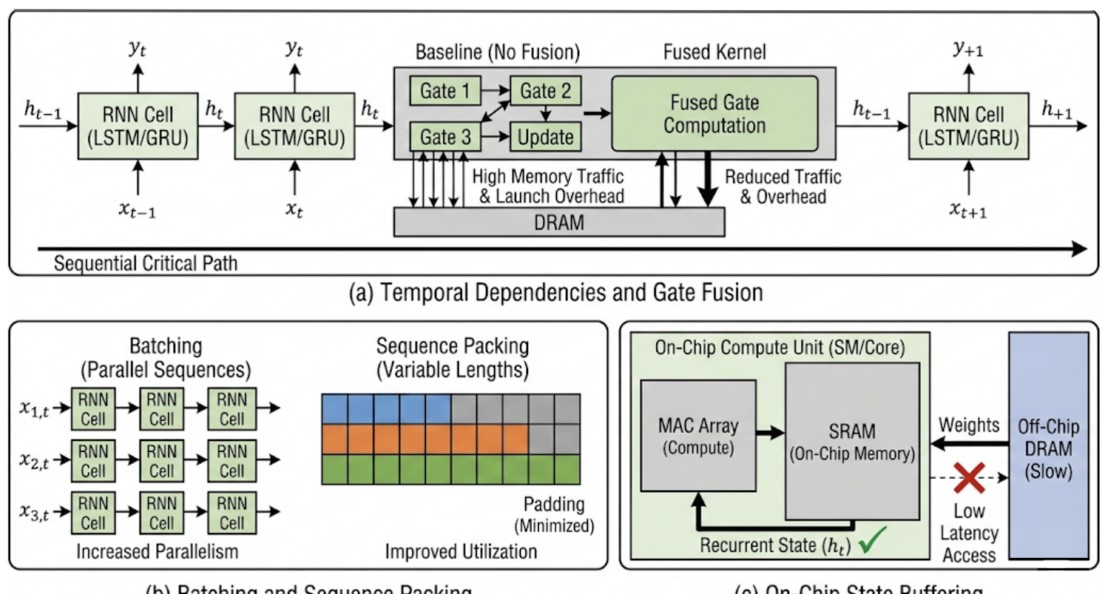

Figure 22: Recurrent Neural Network (RNN) structure showing temporal dependencies and feedback loops, which present challenges for parallelization and require specialized state-buffering techniques.

In practice, RNN acceleration is often limited less by peak MAC throughput than by the ability of the software stack to schedule recurrent kernels efficiently and to minimize overhead between timesteps; Figure 22 provides

the reference RNN structure for this discussion Chen et al. (2018b). Recent advances in linear recurrent models demonstrate that unstructured sparsity can be effectively leveraged for edge deployment, offering improved efficiency for resource-constrained settings Pierro et al. (2025).

### 5.3.1 RNNs on GPUs

GPU RNN acceleration relies on fusing the per-timestep gate computations into a small number of kernels and using persistent/fused implementations that reduce kernel launches and global-memory traffic Chetlur et al. (2014); Tillet et al. (2019). The main idea is to treat each timestep as a compact block (input projection + recurrent projection + nonlinearities) and to execute that block with minimal intermediate materialization, keeping gate activations and partial sums in registers/shared memory when possible Tillet et al. (2019); Chen et al. (2018b). This is especially important for latency-sensitive inference, where each extra kernel launch or synchronization step adds overhead that cannot be amortized across many parallel tokens Williams et al. (2009).

Mixed precision improves throughput for training, but sequential dependencies limit parallelism across timesteps, so utilization can be sensitive to batch size, hidden dimension, and sequence length Micikevicius et al. (2018). For high throughput, systems often rely on batching and sequence packing to increase parallel work per timestep, while for low latency they prioritize persistent kernels and fusion to minimize overhead between timesteps Williams et al. (2009). Compiler stacks and kernel DSLs are increasingly used to generate fused recurrent kernels and to tailor layouts to specific hidden sizes, which helps reduce launch overhead and improve locality Chen et al. (2018b); Tillet et al. (2019).

Practical bottlenecks include launch overhead (small batches), memory bandwidth (materialized intermediates), and synchronization between timesteps. When the hidden state cannot remain in fast memory, repeated reads/writes of state and gate tensors can become bandwidth-bound, making fusion and on-chip buffering more important than peak FLOPs Williams et al. (2009); Chen et al. (2018b). As a result, GPU "acceleration" for RNNs often looks like a compilation and kernel-engineering problem rather than a new hardware primitive: reduce the number of kernels, reduce memory traffic, and keep the recurrent loop tight Tillet et al. (2019).

### 5.3.2 RNNs on TPUs/NPUs

On tensor processors, RNNs are accelerated by mapping each timestep's dense transforms onto systolic arrays, while the compiler/runtime manages loop structure, layout, and buffering to reuse weights and state Jouppi et al. (2017). The primary approach is to keep the per-timestep GEMMs large enough (via batching or packing) to efficiently use the array, and to stage recurrent state through on-chip SRAM so that the loop does not become dominated by off-chip traffic Williams et al. (2009). Because the computation repeats over many timesteps, keeping weights resident and minimizing layout changes across iterations can be critical for latency and energy.

Compared to CNNs, recurrent control flow can be harder to optimize ahead-of-time, and utilization depends on keeping the matrix engine busy despite timestep-level dependencies. When the compiler can unroll or partially unroll loops with static sequence lengths, it can fuse parts of the recurrent body and schedule data movement to overlap with compute; when sequence lengths are dynamic, additional bookkeeping and padding can reduce efficiency Williams et al. (2009). Mixed precision provides throughput benefits for training, but the sequential structure still limits parallelism across time Micikevicius et al. (2018).

In edge NPUs, support depends on whether the device/compiler provides fused recurrent primitives; otherwise, repeated small operators may fall back and lose efficiency Mazumder et al. (2021). In these settings, end-to-end performance depends less on peak array throughput and more on staying within the NPU execution envelope (supported ops, supported layouts) so that the recurrent loop does not repeatedly cross the CPU/NPU boundary Mazumder et al. (2021); Williams et al. (2009).

### 5.3.3 RNNs on ASIC accelerators

Dedicated RNN ASIC acceleration is less common than CNN-focused designs, but the same principles apply: map gate GEMMs to a MAC array with on-chip SRAM buffers to keep recurrent state local and reduce DRAM traffic Chen et al. (2014). The accelerator approach is to hardwire efficient dense transforms and to treat the recurrent state as a first-class on-chip resident tensor, minimizing off-chip reads/writes each timestep. With sufficient buffering, weights for the recurrent matrices can remain on-chip and be reused across timesteps, while activations stream through the datapath in a predictable order Williams et al. (2009).

The main challenge is balancing efficiency with flexibility, since RNN variants and sequence lengths can vary and the sequential nature reduces opportunities for large-batch amortization Hennessy & Patterson (2018). If the ASIC is specialized too narrowly (specific hidden sizes, specific cell types), it risks poor coverage as models evolve; if it is made more programmable, it loses some of the efficiency advantage that motivates ASIC design in the first place Hennessy & Patterson (2018). Additionally, supporting training requires backward operators and higher precision accumulation, which further increases complexity compared to inference-oriented designs Micikevicius et al. (2018).

When such accelerators are deployed, they are most effective in stable inference workloads (fixed cell type, fixed dimensions) where deterministic execution and low DRAM traffic dominate performance and energy. In these regimes, specialized buffering, streaming schedules, and reduced-precision datapaths can yield strong efficiency, provided the system integration does not reintroduce overhead through host-device transfers Williams et al. (2009); Hennessy & Patterson (2018).

### 5.3.4 RNNs on FPGAs

FPGAs can accelerate RNNs using deeply pipelined datapaths that stream inputs through fused gate computations, offering deterministic latency when dimensions are fixed Nurvitadhi et al. (2017). Similar to FPGA CNNs, the design emphasizes a spatial schedule: compute for one timestep advances through a pipeline, while recurrent state is buffered and fed back with minimal control overhead. When hidden sizes are fixed, designers can unroll inner products and gates into a balanced pipeline that uses DSP blocks efficiently and sustains a steady initiation interval Williams et al. (2009).

The approach benefits from keeping weights/state on-chip and using custom precision to reduce bandwidth and DSP usage. Fixed-point formats are often chosen to fit BRAM and reduce external bandwidth, and keeping the entire recurrent loop in a consistent format avoids conversion overhead Jacob et al. (2018). Inference is generally a better fit than training because weight updates and backward-pass accumulation are harder to implement efficiently in a fixed pipeline Micikevicius et al. (2018).

However, long sequences can stress on-chip buffering, and performance is often bounded by external memory bandwidth or host I/O if the pipeline cannot keep data resident Williams et al. (2009); Venieris & Bouganis (2016). In system deployments, additional overhead can come from host-device transfers and from how sequences are batched/packed; therefore, practical FPGA RNN acceleration often co-designs the data ingress/egress pipeline with the compute datapath so that the recurrent loop remains streaming rather than I/O-bound Nurvitadhi et al. (2017); Williams et al. (2009).

## 5.4 Transformer and LLM acceleration

Transformers are now a dominant neural architecture for language, vision, and multimodal workloads, with large language models (LLMs) typically implemented as decoder-only Transformers Vaswani et al. (2017). While the compute core still consists of dense GEMMs (QKV projections and MLP blocks), end-to-end efficiency is increasingly dominated by *attention* and by memory-system behavior (especially KV-cache during autoregressive inference) Vaswani et al. (2017); Kwon et al. (2023); Williams et al. (2009).

### 5.4.1 Different Transformer and LLM model variants

From an accelerator perspective, Transformer variants differ mainly in their attention pattern, sequence lengths, and whether execution is dense or conditional. These choices directly determine arithmetic intensity,

memory footprint, and the amount of irregular control (routing, masking) that the hardware/software stack must handle efficiently Vaswani et al. (2017); Williams et al. (2009).

- **Encoder-only Transformers**: used for representation learning (e.g., classification/embedding). They process the full sequence in parallel, so throughput is driven by batched matmuls and attention over the full context (e.g., Bidirectional Encoder Representations from Transformers (BERT)-style encoders) Vaswani et al. (2017); Devlin et al. (2019).

- **Encoder–decoder Transformers**: used for sequence-to-sequence tasks. They combine self-attention with cross-attention, increasing memory traffic and operator count due to additional attention blocks and intermediate activations (e.g., T5-like models) Vaswani et al. (2017); Raffel et al. (2020).

- **Decoder-only LLMs**: used for autoregressive generation (e.g., Generative Pre-trained Transformer (GPT)-style and LLaMA-style). Inference alternates between a *prefill* phase (processing the prompt) and a *decode* phase (generating tokens). Decode is often bandwidth-limited because each token requires reading/writing KV-cache and executing relatively small GEMMs Brown et al. (2020); Touvron et al. (2023); Kwon et al. (2023).

- **Long-context models**: increase sequence length, amplifying KV-cache footprint and attention IO. This shifts bottlenecks from compute toward memory capacity/bandwidth and makes tiling and paging policies critical Kwon et al. (2023). Architecturally, long-context variants may use sparse attention (e.g., Longformer/BigBird Beltagy et al. (2020); Zaheer et al. (2020)), recurrence (Transformer-XL Dai et al. (2019)), or linearized attention (Performer Choromanski et al. (2021)) to reduce the quadratic cost of attention, changing the balance between compute and memory access patterns.

- **Mixture-of-Experts (MoE) Transformers**: route tokens to a subset of experts, reducing arithmetic but introducing conditional execution, routing overhead, and load imbalance; benefits depend on runtime scheduling and interconnect (e.g., Switch Transformer) Shazeer et al. (2017); Fedus et al. (2022); Gale et al. (2019).

These variants also differ in their *serving behavior*. Encoder-only and encoder–decoder models are often used for batched inference with full-sequence computation, whereas decoder-only LLMs are typically deployed in interactive settings where decode speed (tokens/s) and tail latency matter most Kwon et al. (2023). As a result, accelerator strategies that work well for training or offline inference (large GEMMs, large batches) can underperform for real-time decode unless kernels and runtimes are explicitly optimized for small, repeated per-token work Williams et al. (2009).

### 5.4.2 Key kernels and bottlenecks: attention, KV-cache, and fusion

Transformer blocks are a mix of compute-heavy GEMMs (projections and MLP) and bandwidth-heavy attention, as summarized in Figure 23. A useful lens is to separate *prefill* (full-sequence computation) from *decode* (token-by-token generation) for decoder-only LLM inference. Prefill can be GEMM-heavy and benefits from batching, while decode often becomes memory- and latency-bound due to KV-cache access and small GEMM shapes Kwon et al. (2023); Williams et al. (2009).

Key operator families include: (i) dense GEMMs for QKV projections and MLP blocks, (ii) attention score computation + softmax + weighted value aggregation, and (iii) normalization and elementwise chains (layer norm, activations). Many of these operators are individually optimized, but end-to-end performance depends on how well the stack *reduces intermediate tensor materialization* and *keeps data resident* in fast memory across the block Chen et al. (2018b); Tillet et al. (2019).

During inference serving, KV-cache growth makes memory a first-class resource: bandwidth per generated token and KV-cache capacity often set the performance envelope, especially at long contexts and high concurrency Kwon et al. (2023); Williams et al. (2009). IO-aware attention implementations tile computation to reduce intermediate materialization and improve locality (e.g., FlashAttention-style tiling), while runtime

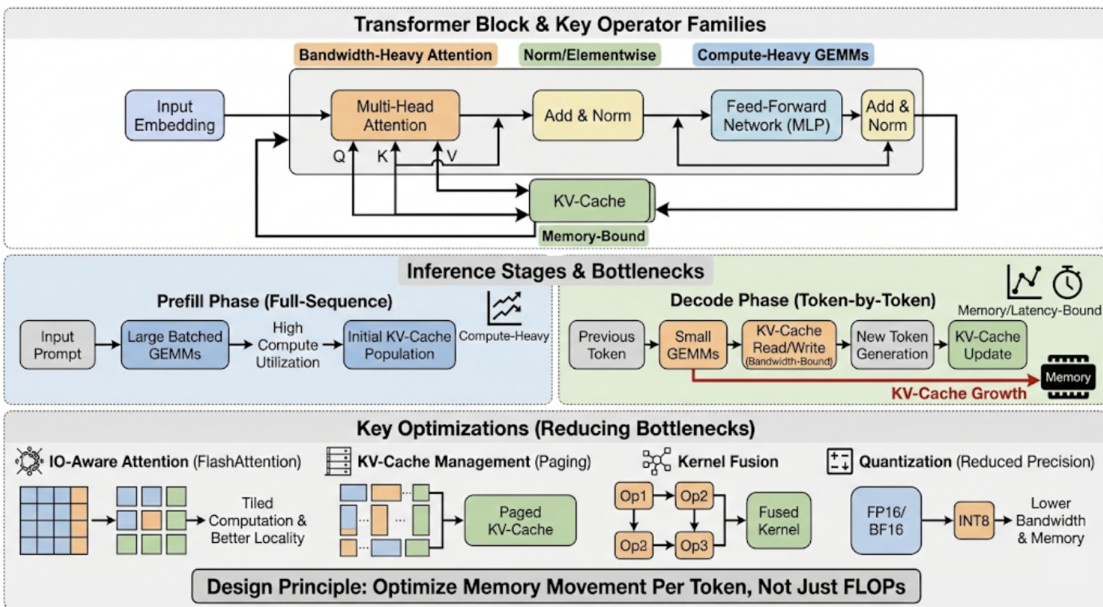

Figure 23: Critical kernels and system bottlenecks in LLM acceleration, detailing the interplay between dense GEMM projections, bandwidth-intensive attention mechanisms, and the impact of sequence length on memory capacity.

techniques such as paging KV-cache are critical to scaling concurrency under fixed device memory Dao et al. (2022); Kwon et al. (2023). These techniques illustrate a key design principle for LLM accelerators: optimize *memory movement per token*, not just FLOPs.

Fusion and compilation further matter because Transformer blocks contain many short operator chains around the GEMMs. Fused kernels reduce launch overhead and global-memory traffic across the attention–MLP pipeline, and layout/epilogue selection often determines whether the workload is compute-bound or bandwidth-bound Tillet et al. (2019); Chen et al. (2018b); Williams et al. (2009). Precision choices matter throughout: mixed precision is standard for training, while quantized inference reduces bandwidth and improves throughput-per-watt when the stack supports it end-to-end Micikevicius et al. (2018); Jacob et al. (2018). Sparsity can reduce arithmetic, but it introduces irregular access patterns and scheduling challenges unless sparsity structure is supported by kernels and data formats Gale et al. (2019).

Recent acceleration strategies for Transformer/LLM inference increasingly target *decode efficiency* and *memory traffic per token*. Examples include faster IO-aware attention implementations beyond the original FlashAttention design Dao et al. (2022); Dao (2024), attention variants that reduce KV-cache bandwidth (e.g., grouped-query attention, which reduces the number of KV heads relative to Q heads) Ainslie et al. (2023), and serving-time techniques such as speculative decoding that trade extra parallel work for lower end-to-end latency Leviathan et al. (2023). Post-training quantization has also evolved into practical, high-accuracy pipelines for LLMs—including SmoothQuant Xiao et al. (2023), GPTQ Frantar & Alistarh (2023), and activation-aware weight quantization (AWQ) Lin et al. (2024)—which reduce bandwidth and enable lower-precision kernels when supported by hardware and libraries Jacob et al. (2018); Gong et al. (2024). More radically, the "1-bit LLM" paradigm (e.g., BitNet b1.58) proposes ternary weights $\{-1, 0, 1\}$ to drastically reduce memory footprint and replace expensive FP16 multiplications with integer addition, promising significant energy savings if hardware support matures Ma et al. (2024). For fine-tuning efficiency, recent work demonstrates that LoRA adaptations can operate effectively at extremely low precision (sub-2-bit), enabling parameter-efficient training on resource-constrained platforms Zhou et al. (2025).

### 5.4.3 Transformers/LLMs on GPUs

On GPUs, the dominant approach is to maximize tensor-core utilization for GEMMs while using specialized attention kernels and aggressive fusion to reduce memory traffic Tillet et al. (2019); Dao et al. (2022). For training and offline inference, large batched GEMMs in projections and MLPs can reach high utilization; for interactive LLM serving, performance hinges on decode efficiency where GEMMs are smaller and the runtime must orchestrate many concurrent requests Kwon et al. (2023); Williams et al. (2009).

Practical LLM serving on GPUs emphasizes (i) efficient decode for small GEMMs (kernel selection and fusion), (ii) KV-cache layout and paging to scale concurrency under limited device memory, and (iii) dynamic batching/scheduling to balance throughput and tail latency Kwon et al. (2023). Attention kernels are particularly important: IO-aware tiling reduces intermediate tensor size and improves bandwidth efficiency, which can directly improve tokens/s when the workload is KV-cache and bandwidth dominated Dao et al. (2022). Graph compilers and kernel DSLs are used to fuse epilogues and pointwise chains around GEMMs to reduce launch overhead and global memory traffic Chen et al. (2018b); Tillet et al. (2019).

Mixed precision (FP16/BF16) is widely used for training; for inference, quantization can be effective but requires kernel/library support and careful handling of conversions to avoid eroding gains Micikevicius et al. (2018); Jacob et al. (2018). In addition, memory hierarchy effects matter: even with fast GEMMs, repeated reads of KV-cache and activations can saturate bandwidth and stall compute, making layout and caching policy central to performance Williams et al. (2009); Kwon et al. (2023).

When MoE is used, GPU efficiency depends on routing implementation, expert parallelism, and load balance; otherwise, conditional execution can reduce utilization and increase tail latency even when compute capacity is high Shazeer et al. (2017); Gale et al. (2019). Overall, GPUs remain the default platform because the software ecosystem (libraries, compilers, serving runtimes) evolves rapidly with model variants and can incorporate new attention kernels and scheduling policies without changing hardware Chen et al. (2018b); Tillet et al. (2019). Continued architectural scaling is evident in recent generations such as NVIDIA's Blackwell platform, which integrates narrower precision support (FP4/FP6) and massive chip-to-chip bandwidth to support trillion-parameter training and inference NVIDIA Corporation (2024a). Similarly, AMD's Instinct MI300X series targets the same high-bandwidth, high-capacity regime by integrating CPU and GPU cores with large shared HBM pools to reduce host-device transfer overheads AMD (2024).

### 5.4.4 Transformers/LLMs on TPUs/NPUs

On TPUs/NPUs, dense GEMMs map naturally to systolic arrays, and compilation plans tiling/layout to keep the array utilized while managing on-chip SRAM reuse Jouppi et al. (2017); Kung (1982). A central accelerator approach is therefore *compile-time dataflow planning*: choose blocking that maximizes operand reuse, schedule transfers to and from SRAM, and select layouts that reduce reshaping between layers Jouppi et al. (2017); Williams et al. (2009).

For large-scale training, TPU-class systems demonstrate that system co-design (HBM bandwidth, on-chip SRAM, and high-bandwidth interconnect) matters as much as per-chip compute. Scaling LLM training requires sharding and collective communication that interact with compilation decisions, and overall efficiency depends on overlap between compute and communication as well as on memory footprint (activations, optimizer state) Jouppi et al. (2021); Micikevicius et al. (2018). Recent iterations, such as Google's TPU v5p, further optimize this balance by increasing inter-chip interconnect bandwidth and HBM capacity specifically to handle the communication-intensive training of large generative models Google Cloud (2024). Reduced precision (BF16/FP16) is widely used to increase throughput and manage memory traffic Micikevicius et al. (2018).

The main challenge for inference and serving is that attention and KV-cache can become bandwidth- and capacity-limited at long contexts and high concurrency, reducing utilization even when peak compute is high Kwon et al. (2023); Williams et al. (2009). Similar to GPUs, IO-aware attention and careful KV-cache management are required to avoid turning the execution into a sequence of bandwidth-bound memory operations Dao et al. (2022); Kwon et al. (2023).

In edge NPUs, feasibility depends on operator coverage (attention, softmax, layer norm) and on keeping the graph inside the accelerator without CPU fallbacks, because cross-device fallbacks introduce extra copies and latency Mazumder et al. (2021). Quantized inference is common for power/thermal reasons, but end-to-end gains depend on consistent quantized execution and minimizing conversions Jacob et al. (2018).

### 5.4.5 Transformers/LLMs on ASICs and LLM-serving accelerators

ASIC approaches aim to optimize the stable core (dense GEMM + attention primitives) with efficient dataflows and large on-chip buffers, but end-to-end wins depend on provisioning memory bandwidth/capacity for KV-cache and intermediate tensors Chen et al. (2016); Kwon et al. (2023). From a design standpoint, the challenge is that Transformer inference mixes a matmul-centric datapath with bandwidth-heavy attention and cache access; thus, an ASIC must balance MAC throughput with memory-system throughput, rather than maximizing one in isolation Williams et al. (2009).

A common approach is to implement a high-throughput matrix engine (MAC array or systolic array) and pair it with multi-level buffering and an on-chip interconnect that sustains a chosen dataflow. Large on-chip SRAM helps reduce DRAM traffic for weights/activations that can be tiled, but KV-cache for long-context serving often exceeds on-chip capacity, forcing reliance on high external bandwidth and careful cache layout Kwon et al. (2023). Attention and softmax-like kernels may require dedicated support or efficient microcode because they can become performance-critical in decode regimes Vaswani et al. (2017); Dao et al. (2022).

Dedicated LLM-serving accelerators (e.g., LPU-style designs) emphasize predictable low-overhead execution for token generation by reducing dispatch complexity and targeting deterministic pipelines Groq (2024). However, they remain constrained by the fundamental bandwidth-per-token cost of reading/writing KV-cache, which sets a system-level lower bound on achievable latency and energy per token at high concurrency Kwon et al. (2023); Williams et al. (2009). As a result, practical designs focus on holistic system efficiency: memory provisioning, cache management, and minimizing control overheads that add to tail latency Kwon et al. (2023).

For MoE-like workloads, system support for routing and load balance is essential; otherwise, conditional execution shifts the bottleneck to scheduling and communication Shazeer et al. (2017); Gale et al. (2019). Compared with GPUs, ASICs can offer strong efficiency for stable operator sets, but they risk underperforming when models evolve rapidly (new attention variants, new normalization/activation patterns) unless sufficient programmability is provided Hennessy & Patterson (2018).

### 5.4.6 Transformers/LLMs on FPGAs

FPGA acceleration (Figure 24) for Transformers focuses on building streaming pipelines for projections/MLP and (when feasible) attention/softmax, using custom precision and on-chip buffering to reduce off-chip traffic Zhang et al. (2024a); Nurvitadhi et al. (2017); Boutros et al. (2024). The FPGA approach mirrors classical CNN FPGA acceleration: exploit spatial pipelining and determinism, tailor datapath precision, and fuse adjacent stages so that intermediate tensors do not spill to DRAM Venieris & Bouganis (2016); Williams et al. (2009).

In practice, attention and KV-cache pressure external bandwidth, and limited on-chip BRAM can force frequent off-chip accesses, reducing throughput Zhang et al. (2024a); Kwon et al. (2023); Williams et al. (2009). This is especially challenging for decode, where the computation per token is small but the KV-cache traffic is unavoidable. As a result, FPGA designs often explore structured sparsity or block-sparse attention patterns, and they rely on careful buffering and tiling to keep memory access regular Zhang et al. (2024a).

System integration is a major determinant of end-to-end benefit. Host I/O (e.g., PCIe), data marshaling, and batching policies can dominate latency unless the serving pipeline is engineered so that the FPGA remains compute-fed Nurvitadhi et al. (2017). Consequently, FPGA designs often target fixed model shapes and low-latency regimes where deterministic pipelines and tight I/O integration offset lower peak compute, while toolchain support and memory-system design remain central to achieving robust speedups Venieris & Bouganis (2016); Nurvitadhi et al. (2017).

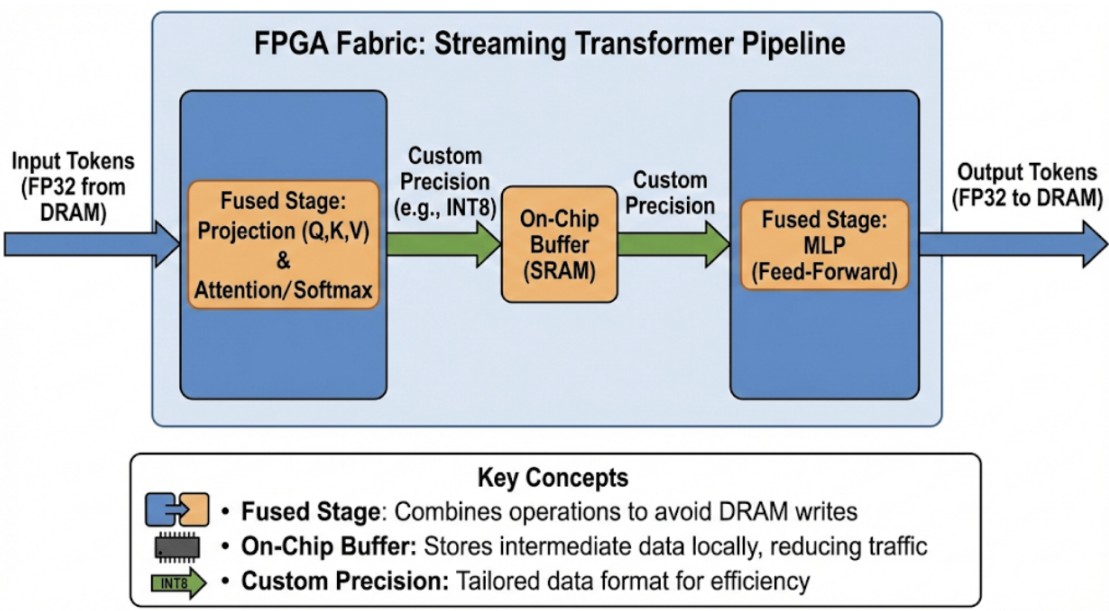

Figure 24: FPGA acceleration pipeline for Transformers, demonstrating how streaming dataflows, custom precision, and fused operators are used to handle attention mechanisms and reduce latency.

More broadly, recent Transformer/LLM platforms reflect a converging set of system requirements: (i) very high memory bandwidth and capacity (to sustain KV-cache and large activations), (ii) hardware support for low-precision tensor math (to increase throughput and reduce bandwidth pressure), and (iii) high-bandwidth interconnect to scale training and to avoid host bottlenecks in serving Williams et al. (2009); Kwon et al. (2023). In datacenters, this has driven larger, LLM-focused accelerator complexes: GPU systems emphasizing large-scale inference and deployment NVIDIA (2023), TPU clusters designed for large-scale training/inference in pods (e.g., recent TPU generations reported as Trillium and Ironwood) Wikipedia contributors (2025d), and alternative form factors such as wafer-scale systems (e.g., CS-3/WSE-3) that reduce distributed communication by concentrating more compute and memory on a single device Wikipedia contributors (2024). In parallel, specialized serving accelerators that prioritize deterministic low-latency token generation remain an active direction, reflecting that decode is often dominated by memory movement and dispatch overhead rather than peak GEMM throughput Groq (2024); Kwon et al. (2023). For practitioners, the implication is consistent with the mapping principles above: match the model's execution regime (training vs. serving; prefill vs. decode; long-context vs. short-context) to a platform whose *memory system* (HBM capacity/bandwidth and interconnect) and *software stack* (attention kernels, KV-cache management, quantization support) can sustain end-to-end throughput and tail latency Dao et al. (2022); Kwon et al. (2023).

Beyond mainstream GPU/TPU deployments, the 2023–2025 landscape also includes a growing set of alternative datacenter accelerators, such as AMD Instinct MI300-class systems Wikipedia contributors (2025a), Intel Gaudi-class accelerators Wikipedia contributors (2025c), and AWS Trainium-class instances Wikipedia contributors (2025b); see Silvano et al. (2025) for a broader overview. On the research side, there is active work on transformer-specific accelerators and system co-design: dynamic sparsity/activation pruning in transformer blocks (e.g., AccelTran) Wang et al. (2023), 3D heterogeneous manycore organizations for end-to-end transformer execution (e.g., HeTraX) Zhang et al. (2024b), FPGA toolflow-driven LLM inference (e.g., HLS-based transformer implementations) ?, and heterogeneous processing-in-memory (PIM) approaches that couple SRAM/HBM compute to reduce bandwidth bottlenecks ?. These efforts reinforce the same overarching lesson: as models scale and contexts grow, end-to-end speedups come from jointly optimizing kernels, memory systems, and runtimes—not from compute throughput alone Williams et al. (2009); Kwon et al. (2023).

## 5.5 Compiler and runtime systems for neural acceleration

As the preceding sections illustrate, hardware capability alone does not determine end-to-end performance; the *compiler and runtime layer* bridges model semantics to the physical datapath, and its quality often dominates realized efficiency. This subsection provides a deeper look at three critical aspects of this layer: graph-level compilation, code generation for tiled/systolic architectures, and serving-system innovations for LLM inference.

### 5.5.1 Graph-level compilation and optimization

Modern deep-learning compilers operate on a *computation graph* (or a traced/captured program representation) and apply a cascade of optimizations before lowering to device kernels.

**PyTorch `torch.compile`.** PyTorch 2's compilation pipeline combines *TorchDynamo*, which captures Python-level computation graphs via bytecode analysis, with *TorchInductor*, a code-generation backend that emits optimized Triton or C++ kernels Ansel et al. (2024). Key graph-level passes include operator fusion (merging pointwise chains with preceding GEMMs to avoid intermediate global-memory writes), layout propagation (selecting channel-last or blocked formats that improve cache behavior), and automatic mixed-precision insertion. Because Dynamo captures at the Python level, it can handle dynamic control flow by graph-breaking and recompiling, trading compilation overhead for generality. In practice, `torch.compile` closes a significant fraction of the gap between eager-mode and hand-tuned kernel performance, particularly for fusion-heavy inference workloads Ansel et al. (2024); Tillet et al. (2019).

**TensorRT.** NVIDIA's TensorRT targets inference optimization by ingesting an Open Neural Network Exchange (ONNX) or TorchScript graph and applying layer fusion, precision calibration (FP32→FP16/INT8 with per-tensor or per-channel scaling), kernel auto-tuning (selecting the fastest kernel variant for each fused subgraph on the target GPU), and memory planning (minimizing peak activation footprint via in-place reuse) NVIDIA Corporation (2024b). TensorRT's strength lies in its deep integration with NVIDIA's kernel library: it can select from hundreds of hand-tuned GEMM, convolution, and attention kernels and schedule them with minimal launch overhead. However, it requires a mostly static graph and fixed input shapes (or a small set of optimization profiles), which limits applicability to highly dynamic models.

**TVM and its ecosystem.** Apache TVM takes a more hardware-agnostic approach. The Relay intermediate representation (IR) captures high-level operator semantics and enables target-independent passes (constant folding, dead-code elimination, layout transformation), while the TensorIR (TIR) layer provides a low-level loop-nest abstraction for per-operator scheduling and tiling Chen et al. (2018b); Feng et al. (2023). Automated search strategies such as Ansor explore the space of loop tile sizes, vectorization widths, and memory scopes to find high-performance schedules for each target device Zheng et al. (2020). This approach enables deployment on diverse backends—GPUs, CPUs, FPGAs, and custom accelerators—but the quality of generated code depends on the maturity of target-specific scheduling templates and auto-tuning budgets.

### 5.5.2 Code generation for tiled systolic architectures

Systolic arrays (and tensor-core-like units) impose rigid constraints on operand shapes, data layouts, and scheduling. The compiler must tile each GEMM or convolution into blocks that match the physical array dimensions, stage operands through the on-chip memory hierarchy with cycle-level precision, and handle edge cases where the problem size does not evenly divide the tile size (requiring padding or masking).

Google's XLA (Accelerated Linear Algebra) compiler exemplifies this for TPUs: it takes a high-level HLO (High-Level Operations) graph, performs algebraic simplification and fusion, then lowers to TPU-specific instructions that orchestrate data movement between HBM, vector memory, and the systolic array Sabne (2020); Jouppi et al. (2017). A central challenge is the *performance cliff*: if the compiler's tiling choice causes even a small fraction of intermediate data to spill from on-chip SRAM to HBM, throughput can drop sharply. As a result, XLA's memory planner must reason globally about liveness and buffer sharing across the entire computation graph, not just individual operators.

For GPUs, analogous challenges arise in mapping to tensor-core warp-level matrix operations (e.g., `mma` instructions): the compiler or kernel author must select warp tile shapes, shared-memory swizzling patterns, and software pipelining depths. Kernel DSLs such as Triton abstract some of this complexity by exposing a tile-based programming model where the user specifies block sizes and the compiler handles register allocation, memory coalescing, and barrier insertion Tillet et al. (2019). Nevertheless, achieving peak utilization on the latest architectures (e.g., Blackwell's FP4 tensor cores) typically still requires kernel-level tuning that accounts for hardware-specific latencies and throughput ratios.

### 5.5.3 Serving runtimes and KV-cache management

For LLM serving, the runtime system is as critical as the kernel library. The dominant innovation in this space is *PagedAttention*, introduced by vLLM, which manages KV-cache memory analogously to operating-system virtual memory Kwon et al. (2023). Rather than pre-allocating a contiguous memory block per sequence (which wastes capacity due to variable and unpredictable output lengths), PagedAttention divides KV-cache into fixed-size pages that are allocated on demand and can be stored non-contiguously in GPU memory. This reduces internal fragmentation, enabling the system to serve significantly more concurrent sequences within the same memory budget.

Beyond paging, modern serving runtimes incorporate several co-designed optimizations:

- **Continuous batching**: rather than waiting for an entire batch to complete before scheduling new requests, the runtime inserts new requests into the batch as soon as slots become available (at each decode step), improving GPU utilization and reducing queuing delay Kwon et al. (2023).

- **Prefix caching**: when multiple requests share a common system prompt, the runtime caches and reuses the KV-cache of the shared prefix, avoiding redundant prefill computation.

- **Speculative decoding integration**: the runtime coordinates a small draft model with the target model, verifying multiple candidate tokens in parallel to reduce the number of sequential decode steps while preserving output quality Leviathan et al. (2023).

These runtime-level innovations interact tightly with kernel design: for example, PagedAttention requires an attention kernel that can gather KV blocks from non-contiguous memory locations, which changes the memory access pattern compared to standard FlashAttention Dao et al. (2022). The overall lesson is that compiler and runtime co-design—spanning graph optimization, kernel generation, and serving-system policies—is now a first-class component of the acceleration stack, often determining whether hardware capabilities translate into real-world throughput and latency improvements.

## 6  Evaluation

Because accelerator results are highly sensitive to *model regime* (training vs. inference; prefill vs. decode), *software stack* (kernels, compiler, runtime scheduling), and *hardware* (compute, memory hierarchy, and interconnect), fair comparison requires reporting both **what was measured** and **how it was measured**. We emphasize end-to-end, user-facing metrics—including TTFT/ITL latency, throughput vs. goodput under concurrency, energy efficiency, roofline limits, KV-cache footprint, and scaling efficiency—to make results comparable across platforms (CPU, GPU, TPU/NPU, FPGA, ASIC, and emerging PIM-style systems).

### 6.1  Workload decomposition: prefill vs. decode

For decoder-only LLM serving, it is often necessary to evaluate *prefill* and *decode* separately because they stress different resources. Let $L_{\mathrm{in}}$ be prompt length (tokens) and $L_{\mathrm{out}}$ be generated length. Define measured wall-clock times $T_{\mathrm{prefill}}$ and $T_{\mathrm{decode}}$. A common decomposition is:

$$T_{\mathrm{total}} = T_{\mathrm{prefill}} + T_{\mathrm{decode}}. \tag{1}$$

Prefill tends to be GEMM-heavy (benefits from batching), while decode is commonly bandwidth/latency dominated due to KV-cache traffic Kwon et al. (2023); Williams et al. (2009).

### 6.2 Core performance metrics

**Latency (per request).** Report mean and tail latency (e.g., $p50/p95/p99$) for interactive settings, and state whether measurements include warmup, compilation, and cache effects. For $N$ requests with end-to-end times $\{T_i\}_{i=1}^N$, define percentile latency:

$$T_p = \text{percentile}_p\left(\{T_i\}_{i=1}^N\right), \quad p \in \{50, 95, 99\}. \tag{2}$$

For LLM serving, also report *time-to-first-token* (TTFT) and *inter-token latency* (ITL) because users perceive responsiveness primarily through TTFT, while sustained throughput is largely determined by ITL. One simple decomposition is:

$$T_{\text{TTFT}} \approx T_{\text{queue}} + T_{\text{prefill}} + T_{\text{first-decode}}. \tag{3}$$

For a request producing $L_{\text{out}}$ tokens, the average ITL is:

$$\overline{T}_{\text{ITL}} = \frac{T_{\text{decode}}}{L_{\text{out}}}, \qquad \text{TP}_{\text{decode}} = \frac{1}{\overline{T}_{\text{ITL}}}. \tag{4}$$

When reporting tail behavior, it is often useful to provide $p99$ for TTFT and ITL separately, since queueing and scheduling can dominate tails under high concurrency Kwon et al. (2023).

**Throughput.** Throughput is tokens processed per second. For prefill and decode:

$$\text{TP}_{\text{prefill}} = \frac{L_{\text{in}}}{T_{\text{prefill}}}, \qquad \text{TP}_{\text{decode}} = \frac{L_{\text{out}}}{T_{\text{decode}}}. \tag{5}$$

In serving, also report *system throughput* under concurrency $C$: $\text{TP}_{\text{sys}} = \frac{\sum_{i=1}^C L_{\text{out},i}}{T_{\text{window}}}$ over a fixed measurement window $T_{\text{window}}$, since scheduler and batching policies strongly affect realized tokens/s Kwon et al. (2023). To connect throughput to service quality, define *goodput* as the throughput of tokens (or requests) that satisfy a latency SLO (e.g., $T_{\text{TTFT}} \leq \tau_{\text{TTFT}}$ and/or $T_{p99} \leq \tau$):

$$\text{Goodput} = \frac{\sum_{i=1}^C L_{\text{out},i} \cdot \mathbf{1}[\text{SLO}(i)]}{T_{\text{window}}}. \tag{6}$$

Goodput can decrease even as raw tokens/s increases when batching and queueing inflate tail latency; reporting both clarifies this trade-off.

**Batching efficiency and scheduling overhead.** Because many accelerators reach higher utilization at larger effective batch sizes, it is useful to quantify how efficiently throughput scales with batch size $B$. Let $\text{TP}(B)$ be throughput at batch size $B$; a simple batching efficiency is:

$$\eta_{\text{batch}}(B) = \frac{\text{TP}(B)}{B \cdot \text{TP}(1)}. \tag{7}$$

For online serving, also report the fraction of time spent outside accelerator kernels (e.g., runtime dispatch, padding, memory copies). If $T_{\text{kernel}}$ is time executing device kernels and $T_{\text{total}}$ is end-to-end time, an overhead ratio is:

$$\rho_{\text{overhead}} = 1 - \frac{T_{\text{kernel}}}{T_{\text{total}}}. \tag{8}$$

**Energy and power (efficiency).** Measure average power $P$ (watts) and energy $E$ (joules) over the measurement interval $T$:

$$E = \int_0^T P(t)\, dt \approx P_{\text{avg}} \cdot T. \tag{9}$$

Energy per token (useful across platforms) is:

$$\text{EPT} = \frac{E}{L_{\text{in}} + L_{\text{out}}}. \tag{10}$$

A widely used combined metric is energy–delay product (EDP):

$$\text{EDP} = E \cdot T, \qquad \text{ED}^2\text{P} = E \cdot T^2. \tag{11}$$

For easier cross-platform comparison, also report normalized efficiency such as tokens per joule and tokens per watt:

$$\text{TPJ} = \frac{L_{\text{in}} + L_{\text{out}}}{E}, \qquad \text{TPW} = \frac{L_{\text{in}} + L_{\text{out}}}{P_{\text{avg}} \cdot T}. \tag{12}$$

**Utilization and achieved performance.** Report achieved throughput relative to theoretical limits. If a run performs $F$ floating-point operations in time $T$, achieved FLOP/s is:

$$\text{FLOP/s}_{\text{ach}} = \frac{F}{T}, \qquad \text{Utilization} = \frac{\text{FLOP/s}_{\text{ach}}}{\text{FLOP/s}_{\text{peak}}}. \tag{13}$$

For bandwidth-limited regimes, similarly report achieved memory bandwidth $B_{\text{ach}}$ and $\frac{B_{\text{ach}}}{B_{\text{peak}}}$.

**Roofline analysis (compute vs. memory bound).** Roofline connects operational intensity $I$ (FLOPs per byte moved from main memory) to achievable performance $\mathcal{P}$:

$$I = \frac{F}{Q}, \qquad \mathcal{P} \leq \min\left(\mathcal{P}_{\text{peak}}, \ I \cdot B_{\text{peak}}\right), \tag{14}$$

where $Q$ is bytes transferred and $B_{\text{peak}}$ is peak memory bandwidth Williams et al. (2009). This is particularly informative for decode, where KV-cache reads can push $I$ low, making bandwidth the dominant limiter.

**Memory footprint and KV-cache capacity.** For long-context or high-concurrency LLM serving, peak memory often limits batch size or concurrency before compute saturates. A practical quantity to report is the KV-cache footprint as a function of context length. For a decoder-only transformer with $N_\ell$ layers, KV heads $H_{\text{kv}}$, head dimension $d$, context length $L$, and element size $s$ bytes (e.g., $s = 2$ for FP16/BF16), a rough KV-cache size is:

$$M_{\text{KV}} \approx N_\ell \cdot 2 \cdot L \cdot H_{\text{kv}} \cdot d \cdot s, \tag{15}$$

and scales linearly with the number of concurrent sequences held in memory. Reporting $M_{\text{KV}}$ (and whether paging/compression is used) helps explain why two platforms with similar peak FLOP/s can behave very differently at the same concurrency Kwon et al. (2023).

**Scaling efficiency (multi-device).** For multi-accelerator setups, report speedup and parallel efficiency as a function of device count $n$:

$$S(n) = \frac{T(1)}{T(n)}, \qquad E(n) = \frac{S(n)}{n}. \tag{16}$$

Because communication patterns differ across data-, tensor-, and pipeline-parallel schemes, it is important to state the parallelization strategy and whether communication is overlapped with compute; otherwise speedup numbers are not comparable across platforms or software stacks Mattson et al. (2020).

**Area footprint and power density.** For ASIC and PIM accelerators, die area is a primary cost driver: it determines manufacturing yield, packaging options, and ultimately dollar-per-TOPS. Two normalized metrics are useful for cross-family comparison:

- **TOPS/mm$^2$** (area efficiency): peak throughput divided by die area. This metric highlights how effectively silicon is used for compute versus control, buffers, and I/O. For example, dense systolic ASICs can exceed $10\,\text{TOPS/mm}^2$ in advanced nodes, while analog crossbar designs trade area for memory density Chen et al. (2016); Shafiee et al. (2016).
- **W/mm$^2$** (power density): TDP divided by die area. Power density determines thermal feasibility; designs that exceed $\sim 1\,\text{W/mm}^2$ require advanced cooling. Datacenter GPUs (e.g., B200 at $\sim 814\,\text{mm}^2$, $1000\,\text{W}$) operate near this limit, while edge ASICs and neuromorphic chips stay well below it NVIDIA Corporation (2024a); Davies et al. (2018).

Reporting area alongside power and throughput enables Pareto analysis across the full efficiency surface (TOPS/W vs. TOPS/mm$^2$) and exposes designs that achieve high throughput-per-watt only by consuming disproportionate die area Hennessy & Patterson (2018); Parashar et al. (2019).

# 7 Conclusion

This survey has reviewed how neural-network acceleration has progressed from simply increasing peak compute to optimizing end-to-end execution under memory, communication, energy, and software constraints. Across model families and deployment settings, the dominant costs increasingly come from moving and managing data (activations, weights, and state) rather than from arithmetic alone. As a result, the most successful accelerator platforms are those that treat compute, memory hierarchy, interconnect, and the software stack as a single co-designed system.

We organized the hardware landscape into several major architectural families, each reflecting a different point in the design space. GPUs provide broad programmability and remain central for training and general-purpose inference, while TPU/NPU-style designs emphasize dense tensor throughput and efficiency via structured datapaths. FPGAs occupy a niche where determinism, custom precision, and streaming pipelines are valuable, and ASIC inference engines pursue maximum performance-per-watt at the cost of flexibility. Emerging architectures—including LPU-like designs for predictable token generation, in-/near-memory and analog approaches aimed at reducing data movement, and neuromorphic/event-driven processors for sparse workloads—highlight continued experimentation as models and serving requirements evolve.

Several optimization levers recur throughout the survey: precision reduction and quantization to improve effective bandwidth and energy efficiency; sparsity and pruning to reduce compute and memory footprint (when supported end-to-end by kernels and scheduling); and compilation, fusion, and layout transformations that reduce intermediate materialization and kernel overheads. For LLM serving in particular, system performance is often governed by memory capacity and bandwidth, KV-cache management, and request scheduling, making runtime policies as important as kernel-level efficiency.

Looking ahead, progress will likely be driven by three system-level needs. First, scalable LLM serving must sustain high concurrency and long contexts without becoming bandwidth- or capacity-limited. Second, dynamic and conditional computation (e.g., routed experts, tool-augmented pipelines, and heterogeneous operators) requires robust support for irregular control flow and load balance. Third, evaluation practice must mature: meaningful comparisons should report end-to-end metrics and configurations that reflect real deployment constraints, rather than relying on peak throughput alone.

In conclusion, the next generation of neural acceleration will be defined less by raw compute density and more by efficient data orchestration and hardware-software co-design. As workloads diversify and deployment expands from data centers to edge devices, architectures that combine adaptable software stacks with well-balanced memory and communication subsystems will remain the most impactful.

## Acknowledgements

This work was partially funded by DARPA (AMP, N6600120C4020; FIRE, P000050426), the NSF (FDT-Biotech, 2436801), and the Helmsley Charitable Trust (2-SRA-2017-503-M-B).

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

# A    Appendix

Table 3: List of Acronyms.

| Acronym | Definition |
|---------|------------|
| ADC | Analog-to-Digital Converter |
| ANN | Artificial Neural Network |
| ASIC | Application-Specific Integrated Circuit |
| BERT | Bidirectional Encoder Representations from Transformers |
| BF16 | Brain Floating Point (16-bit) |
| BNN | Binarized Neural Network |
| BRAM | Block RAM |
| CMOS | Complementary Metal-Oxide-Semiconductor |
| CNN | Convolutional Neural Network |
| DAC | Digital-to-Analog Converter |
| DRAM | Dynamic Random Access Memory |
| DSL | Domain-Specific Language |
| DSP | Digital Signal Processor |
| ECC | Error Correction Code |
| EDP | Energy-Delay Product |
| FLOPs | Floating Point Operations |
| FP16 | 16-bit Floating Point |
| FP32 | 32-bit Floating Point |
| FPGA | Field-Programmable Gate Array |
| GEMM | General Matrix Multiplication |
| GNN | Graph Neural Network |
| GPT | Generative Pre-trained Transformer |
| GPU | Graphics Processing Unit |
| GRU | Gated Recurrent Unit |
| HBM | High-Bandwidth Memory |
| HLO | High-Level Operations |
| HLS | High-Level Synthesis |
| INT4 | 4-bit Integer |
| INT8 | 8-bit Integer |
| ICI | Inter-Chip Interconnect |
| IO | Input/Output |
| IR | Intermediate Representation |
| ISA | Instruction Set Architecture |
| ITL | Inter-Token Latency |
| JIT | Just-In-Time |
| KV-cache | Key-Value cache |
| LLM | Large Language Model |
| LPU | Language Processing Unit |

| Acronym | Definition |
|---|---|
| LSTM | Long Short-Term Memory |
| LUT | Look-Up Table |
| MAC | Multiply-Accumulate |
| MIG | Multi-Instance GPU |
| MLP | Multilayer Perceptron |
| MoE | Mixture-of-Experts |
| NIC | Network Interface Card |
| NoC | Network-on-Chip |
| NPU | Neural Processing Unit |
| ONNX | Open Neural Network Exchange |
| OS | Operating System |
| PCM | Phase-Change Memory |
| PHY | Physical Layer |
| QKV | Query-Key-Value |
| ReRAM | Resistive RAM |
| RNN | Recurrent Neural Network |
| RTL | Register Transfer Level |
| SIMD | Single Instruction, Multiple Data |
| SIMT | Single Instruction, Multiple Threads |
| SLO | Service Level Objective |
| SM | Streaming Multiprocessor |
| SoC | System-on-Chip |
| SRAM | Static Random Access Memory |
| SSD | Solid State Drive |
| STDP | Spike-Timing-Dependent Plasticity |
| TIR | TensorIR (low-level loop IR in TVM) |
| TOPS | Tera Operations Per Second |
| TPU | Tensor Processing Unit |
| TTFT | Time-To-First-Token |
| XLA | Accelerated Linear Algebra |

