# OpenReview forum: "Hardware Acceleration for Neural Networks: A Comprehensive Survey"
_TMLR — Accepted by TMLR_

### Review · Reviewer_dXoW · 2026-03-21

**Summary Of Contributions:**

This paper provides a comprehensive survey of the current landscape of hardware acceleration for neural networks. Its primary contribution is moving beyond the traditional focus on peak compute throughput (FLOPs/TOPS) to present a holistic, system-level view of acceleration, where data movement, memory hierarchies, interconnects, and software stacks are treated as coequal and co-dependent components. The work explored a wide range of accelerator architectures: GPUs, TPUs, NPUs, ASICs, FPGAs, LLM-serving accelerators (LPUs), in-memory computing, and neuromorphic systems with multiple metrics: energy efficiency, latency, throughput, area, cost, and resource utilization.

Key Strengths:
1.) The survey covers an extremely broad range of topics, from classical CNN accelerators to the latest LLM-serving hardware and emerging analog approaches.
2.) The paper's central theme, that data movement and software co-design are now more critical than raw arithmetic, is both timely and accurate. The analysis consistently ties architectural choices back to real-world bottlenecks like memory bandwidth, KV-cache size, and scheduling overhead.
3.) By including dedicated sections on challenges for each accelerator type (e.g., GPU-specific energy challenges, ASIC-specific utilization challenges), the authors provide a fair and nuanced comparison.
4.) The evaluation section has clear, practical metrics (TTFT, ITL, goodput, roofline analysis, KV-cache footprint), especially in the context of modern LLM serving.
5.) The inclusion of recent innovations like FlashAttention, PagedAttention, LPUs (Groq), 1-bit LLMs, and the latest GPU, TPU generations (Blackwell, TPU v5p) makes the survey highly relevant for current research and development.


Key Weaknesses:
1.) The paper's structure, while comprehensive, leads to significant repetition. For instance, the challenges for GPU, TPU, ASIC, FPGA are analyzed separately for each sub-section (energy, throughput, area, etc.). This results in a very long and somewhat redundant read. A more integrated, comparative structure might have improved flow.
2.) The survey is largely descriptive. While it catalogs the features and challenges of each architecture, it rarely provides a strong analytical comparison or a prescriptive framework for choosing one architecture over another for a given workload. The final Conclusion is a summary, not a synthesis, that provides clear guidance.
3.) While the importance of software is mentioned throughout, the discussion remains at a high level. A deeper dive into the specific challenges and innovations in graph compilers (e.g., MLIR, TVM), kernel DSLs (Triton), and their interaction with hardware could have strengthened the co-design argument.

**Audience:**

Yes

**Audience Explanation:**

Yes, I strongly believe the audience of the TMLR would find the paper interesting, informative, and engaging. This paper is relevant for several reasons:

1.) The findings directly inform how ML researchers should design models. Understanding that memory bandwidth is a primary constraint (e.g., due to KV-cache) can influence decisions about model architecture (e.g., using grouped-query attention), quantization, and sparsity patterns to achieve better deployment efficiency.

2.) ML researchers increasingly need to understand the hardware their models run on to optimize performance, especially for large-scale training and serving. This survey provides a comprehensive and accessible bridge between high-level model semantics and low-level hardware realities.

3.)  Practitioners looking to deploy models on cloud or edge platforms will find the analysis of performance, energy, and cost trade-offs across different accelerators (GPUs, TPUs, LPUs, etc.) invaluable for making informed purchasing and deployment decisions.

4.)  The paper's strong stance on rigorous benchmarking (Section 5) provides a clear standard that the TMLR community can adopt to improve the quality and reproducibility of their own system-oriented papers. This makes the work not just informative but also a methodological contribution to the field.

**Broader Impact Concerns:**

This is a survey paper that reviews the technology landscape for hardware acceleration. As such, it does not present new empirical findings or propose a new model with potential for misuse. The primary broader impact is positive as it clarifies the performance and efficiency trade-offs of different hardware; it can help the community build more efficient ML systems, which can reduce the enormous and growing energy footprint of AI. The paper's focus on benchmarking and reproducibility also promotes scientific rigor, which is a net positive.

**Claims And Evidence:**

Yes

**Claims Explanation:**

Yes, the claims are supported by accurate, convincing, and clear evidence, consistent with the nature of a survey paper.

1.) The paper accurately reflects the current state of the field. The descriptions of architectural concepts like systolic arrays, SIMD engines, tensor cores, and dataflows are technically precise. The citations are comprehensive and relevant, drawing from seminal works as well as recent high-impact research.

2.) The central argument, that acceleration is now a memory and software problem, is convincingly built through the repeated analysis of bottlenecks. For every architecture, the authors show how the real-world performance is limited not by peak compute, but by memory movement (DRAM/HBM bandwidth), capacity (KV-cache), or overheads (dispatch, synchronization).

3.) The evidence is presented with clarity. The use of a consistent taxonomy across sections makes it easy to compare how different architectures handle similar challenges. The figures illustrate key concepts (e.g., Figure 4's LLM bottleneck diagram, Figure 6's memory/communication bottlenecks). The evaluation section (Section 5) is a model of clarity, providing concrete equations and definitions for metrics.

**Requested Changes:**

I believe the following adjustments should be addressed to improve the clarity of the paper.

1.) The pages (e.g., 3, 4, 8, 12) containing repetitive sequences of numbers must be corrected. This is a major barrier to readability and is essential for publication.

2.) Consider restructuring to reduce the redundant sections.

3.) The paper mentions Nvidia's Blackwell, Google's TPUv5p, and Groq's LPU. Expanding this to include a concise table comparing key specifications (e.g., peak compute, memory bandwidth, memory capacity, interconnect, typical use-case) of these leading commercial systems would provide a valuable, at-a-glance reference.

4.) While the importance of software is acknowledged, a deeper dive into the compiler runtime layer would strengthen the co-design theme. For instance, discuss the role of graph-level optimizations (e.g., in PyTorch's torch.compile, TensorRT, or TVM), the challenges of generating code for tiled systolic architectures, and the specific innovations in serving systems like vLLM for managing KV-cache.

---

> ### Author Response · Authors · 2026-04-07
> **Review Response**
>
> We sincerely thank the reviewer for their detailed and encouraging evaluation, and for recognizing the core contributions of the survey, the system-level perspective beyond peak FLOPs, the per-family challenge analysis, the practical evaluation metrics, and the coverage of recent innovations. We address each requested change below.
>
>
>
> ## Requested Change 1: Repetitive number sequences on pages 3, 4, 8, 12.
>
> We thank the reviewer for flagging this formatting issue. These were artifacts of the table and figure rendering in the original submission (e.g., Table 1's multi-row numeric entries for peak compute, TDP, and memory specifications being extracted or displayed as raw number sequences). In the revised manuscript, we have carefully reformatted all tables and figures to ensure clean rendering across PDF viewers, eliminating the repetitive number sequences. We have verified the corrected formatting on multiple platforms.
>
>
> ## Requested Change 2: Concise comparison table of leading commercial systems.
>
> This is now provided in the revised manuscript. **Table 1** compares eight representative platforms—NVIDIA B200 (Blackwell), Google TPU v5p, AMD MI300X, AMD Alveo V80 (FPGA), Qualcomm Cloud AI 100 Ultra (ASIC), Groq LPU, Samsung HBM-PIM, and Intel Loihi 2—across peak compute, TDP, TOPS/W, memory bandwidth, memory capacity, and typical use-case. This gives the at-a-glance reference the reviewer requests. Additionally, **Figure 4** plots these platforms on a log–log TOPS vs. TOPS/W scatter with iso-power envelopes, and **Table 2** maps workload characteristics to hardware requirements and best-fit platforms.
>
>
>
> ## Requested Change 3: Deeper dive into compiler/runtime layer (graph compilers, kernel DSLs, serving systems).
>
> The revised manuscript adds a dedicated subsection structure under **Section 5.5** ("Compiler and runtime systems") that provides the deeper treatment the reviewer requests:
>
> - **Section 5.5.1 ("Graph-level compilation and optimization")** covers three major compiler stacks in detail: **PyTorch torch.compile** (TorchDynamo's bytecode-level graph capture, TorchInductor's Triton/C++ code generation, operator fusion, layout propagation, and automatic mixed-precision insertion); **TensorRT** (ONNX/TorchScript ingestion, layer fusion, precision calibration with per-tensor/per-channel scaling, kernel auto-tuning across hundreds of hand-tuned variants, and memory planning via in-place reuse); and **TVM** (the Relay IR for target-independent passes, TensorIR for loop-nest scheduling and tiling, and Ansor's automated search over tile sizes, vectorization widths, and memory scopes for cross-platform deployment).
>
> - **Section 5.5.2 ("Code generation for tiled systolic architectures")** discusses the specific challenges of generating code for systolic arrays and tensor-core units: tiling GEMMs to match physical array dimensions, staging operands through on-chip memory hierarchies, handling edge cases with padding/masking, and the performance cliff problem where small SRAM spills to HBM cause sharp throughput drops. We cover **XLA** for TPUs (HLO graph optimization, global memory planning for liveness and buffer sharing) and **Triton** as a kernel DSL that abstracts tile-based programming for GPUs while the compiler handles register allocation, memory coalescing, and barrier insertion.
>
> - **Section 5.5.3 ("Serving runtimes and KV-cache management")** provides an in-depth treatment of **vLLM's PagedAttention** (paged KV-cache allocation analogous to OS virtual memory, reducing internal fragmentation for concurrent sequences), **continuous batching** (inserting new requests at each decode step rather than waiting for batch completion), **prefix caching** (reusing KV-cache for shared system prompts), and **speculative decoding integration** (coordinating draft and target models for parallel token verification). We discuss how these runtime innovations interact with kernel design—for example, PagedAttention requires attention kernels that gather KV blocks from non-contiguous memory, changing access patterns relative to standard FlashAttention.
>
> We believe this treatment now provides the concrete, tool-specific depth the reviewer was seeking, while reinforcing the paper's central co-design theme.
>
>
>
> We are grateful for the reviewer's positive assessment and constructive suggestions, which have materially improved the revised manuscript.

---

### Review · Reviewer_NgTZ · 2026-03-23

**Summary Of Contributions:**

The paper presents a comprehensive and timely survey on hardware acceleration techniques for machine learning acceleration by covering diverse generic and domain-specific platforms such as GPU, TPU, NPU, ASIC and FPGA. It also covers various ML workloads from CNNs to RNNs to GNNs to LLMs by analyzing execution settings and optimization methods at different levels of abstractions. At the end, the paper provides forward-looking insights and promising directions for deployment challenges such workloads with energy and latency in mind.

**Audience:**

Yes

**Audience Explanation:**

I think this paper can be of interest to broader AI/EDA/Computer Architecture communities after proper revision.

**Claims And Evidence:**

No

**Claims Explanation:**

While the paper does a nice job in analyzing the hardware acceleration landscape and inference/training bottleneck and optimization strategies, a quantitative analysis of selected GPUs, TPUs, FPGAs, ASICs, digital and analog in-memory computing platforms in terms of performance, i.e., TOPS, TOPS/W seems to be crucial to make a strong ground for the later discussions. Plotting such data can provide a nice insight into various hardware/software co-design capabilities for various workloads.

**Requested Changes:**

1- A quantitative analysis of GPUs, TPUs, FPGAs, ASICs, digital and analog in-memory computing platforms in terms of performance in terms of TOPS, TOPS/W seems to be crucial to make a strong ground for the later discussions. Plotting such data can provide a nice insight into various hardware capabilities for various workloads. (critical)

2- Figure 2 could be improved by listing pros and cons of AI HW designs. To the reviewer the flexibility and programmability could be as important as other metrics listed. As for ASIC (Section 2.1.3), the operator coverage requires more articulation. Can operator coverage be considered as downside if a proper SW/HW co-design can bring the highest accuracy for an AI workload? (critical)

3- Classifying FPGAs in general as energy-inefficient or efficient HW can be misleading. Various types of FPGAs (low-end/high-end) serve as AI accelerators with a range of performance and wattage. Therefore, a detailed and fair comparison is needed to show FPGA energy challenges and energy-efficiency. (critical)

4- The survey ignores the body of recent literature on analog and digital in-memory, near-memory, and in-sensor computing frameworks and compilers for on-chip training and inference. It is helpful to show how the energy challenges and intrinsic device non-idealities have been addressed in such designs.  (critical)

5- It is recommended to articulate more on specialized neuromorphic CMOS chip designed to implement Spiking Neural Networks such as IBM TrueNorth.

6- To the reviewer the benchmarking challenges discussed in the paper is timely and interesting, however, the section lacks a comprehensive review of available SW/HW co-design methods and simulators particularly for ASICs and PIM platforms. For example, NeuroSim AND MNSIM are two highly-applicable in-memory computing benchmarking tools.

7- Training with in-memory/analog hardware is more challenging, however, there are several works out there addressing the backprop algorithm efficiently. It is helpful to discuss the hardware mapping and dataflow for such a complex process on HW as well.

8- Please expand the evaluation section considering area footprint and power density factors into account.

---

> ### Author Response · Authors · 2026-04-07
> **Review Response**
>
> We thank the reviewer for their thorough and constructive feedback.
>
> ## 1. Quantitative analysis in terms of TOPS, TOPS/W.
> The revised manuscript provides this in Section 2: **Table 1** compares eight representative platforms (NVIDIA B200, Google TPU v5p, AMD MI300X, AMD Alveo V80, Qualcomm Cloud AI 100 Ultra, Groq LPU, Samsung HBM-PIM, Intel Loihi 2) across peak compute, TDP, TOPS/W, memory bandwidth, and memory capacity. **Figure 4** plots peak TOPS vs. TOPS/W on a log–log scale with iso-power envelopes and workload-affinity color coding. **Figure 3** provides a unified roofline view mapping six workload categories against a reference accelerator ceiling, making explicit why compute-bound training favors GPUs/TPUs while bandwidth-bound inference favors LPUs/ASICs/PIM. **Table 2** summarizes per-workload hardware requirements and best-fit platforms.
>
> ## 2. Figure 2 pros/cons, flexibility/programmability, and ASIC operator coverage.
> The original Figure 2 has been expanded into **Figure 5**, which maps each accelerator family to its limitations across five design axes: speed, energy efficiency, memory/communication, area/cost, and benchmarking. Flexibility and programmability—a sixth, "often underappreciated" dimension—is addressed through a dedicated discussion in the **Section 3 preamble** rather than in Figure 5, because the trade-off is multi-faceted and does not lend itself to a single visual axis. This discussion provides a per-family analysis: GPUs absorb architectural shifts through software at the cost of higher memory-traffic overhead; TPUs degrade on dynamic control flow; FPGAs offer reconfigurability at the cost of lower clock frequencies; and LPUs target inference-specific bottlenecks at the expense of training capability.
>
> Regarding ASIC operator coverage: Sections 3.3.3 and 4.5 discuss that limited coverage need not be a downside when SW/HW co-design maps workloads onto available operators with high accuracy and efficiency. We agree with the reviewer that operator coverage is a design trade-off, not an inherent flaw.
>
> ## 3. FPGA energy classification.
> Section 3.1.4 explicitly distinguishes across the FPGA spectrum: low-end edge devices (Xilinx Zynq, Intel Cyclone) at 1–5 W deliver compelling TOPS/W for quantized CNN inference, while high-end datacenter FPGAs (Alveo U280, Stratix 10) consume 75–225 W and compete with GPUs on throughput but not necessarily energy efficiency. We further discuss sensitivity to external memory bandwidth, BRAM/DSP utilization, routing overhead, and mapping quality. Section 3.3.4 complements this with FPGA-specific area/cost analysis.
>
> ## 4. In-memory, near-memory, and in-sensor computing.
> Section 4.6 and Figure 17 cover the PUMA compiler, NeuroSim and MNSIM simulators, and non-ideality-aware training. Section 3.6.6 discusses benchmarking challenges for analog/PIM platforms. In the revision, **Section 4.6.3** adds a new discussion of in-sensor computing, describing processing-in-pixel (PIP) architectures that embed convolutional kernels within CMOS image sensor arrays (Hsu et al., 2023; Zhou & Chai, 2020; Mennel et al., 2020). We discuss the distinct constraints separating in-sensor designs from in-memory accelerators—pixel-area budgets, lagging CIS process nodes, and sensing-modality coupling.
>
> ## 5. IBM TrueNorth neuromorphic chips.
> The revision adds IBM's TrueNorth (Akopyan et al., 2015; Merolla et al., 2014) and NorthPole (Modha et al., 2023) in Section 4.7. We describe TrueNorth's GALS architecture (4,096 cores, 1M neurons, 256M synapses, ~65 mW, 28 nm) and contrast its fixed neuron model against Loihi 2's programmable approach. We also discuss NorthPole (256 cores, 224 MB on-chip SRAM, 2/4/8-bit), which abandons spiking entirely, achieving 25× energy efficiency and 22× latency gains over comparable 12 nm GPUs.
>
> ## 6. Benchmarking tools and SW/HW co-design for ASICs and PIM.
> NeuroSim and MNSIM are discussed in Sections 3.6.6 and 4.6, covering crossbar array non-ideality modeling, peripheral circuit overhead, and inter-array data movement for realistic area/latency/energy projections. Section 3.6.3 covers ASIC-specific benchmarking challenges.
>
> ## 7. Backpropagation on in-memory/analog hardware.
> Section 4.6.2 covers all three backpropagation phases on crossbar arrays: forward-pass VMM, backward-pass transposed reads, and in-situ weight updates via voltage pulses. We discuss the Manhattan update rule, mixed-signal dataflows interleaving analog VMM with digital accumulation, and hybrid training strategies where analog crossbars handle GEMM/conv while digital logic handles reductions and optimizer updates.
>
> ## 8. Evaluation section, area footprint and power density.
> The evaluation section now includes **TOPS/mm²** (area efficiency) and **W/mm²** (power density) metrics, enabling Pareto analysis across the full efficiency surface. Section 3.3 provides per-platform area/cost analysis for all six accelerator families.

---

### Review · Reviewer_DvVg · 2026-03-24

**Summary Of Contributions:**

The authors review the landscape of AI hardware accelerators, pointing out their common design patterns and open challenges.

**Audience:**

Yes

**Audience Explanation:**

Most readers of TMLR would be interested in advanced in hardware for faster/more efficient ML inference.

**Broader Impact Concerns:**

This work is a survey of other works' methods and results so it does not raise any broader impact concerns on its own.

**Claims And Evidence:**

Yes

**Claims Explanation:**

The authors thoroughly summarize the different approaches to building accelerators, including analog/neuromorphic and other unconventional hardware platforms. They accurately draw attention to the fact that data movement is one of the critical costs in estimating the power consumption of neural-network workloads, especially in accelerators that make the compute portion of the energy cost even cheaper, and the concrete numbers used for estimates (ie. fig. 3) seem accurate to me. Analysis of other aspects of accelerators - throughput, latency, implementation challenges, and affects of neural-network architecture - is also thorough.

**Requested Changes:**

The authors claim that in neuromorphic/spiking networks, data transportation costs within the compute units (number of spikes, as opposed to dac/adc/ram) can affect energy cost. While this is the popular viewpoint in the field of electronics-based spiking neural networks, this is not universally true: In other analog/neuromorphic modalities (such as in reversible, photonics-based, thermodynamic computing, etc) after encoding (load from ram, convert with dac), data may be copied and transported for free (or reused heavily), leading to fundamentally different scaling properties in their energy consumption [1, 2, 3]. I believe it would be worth mentioning that in general for neuromorphic accelerators, the big-O scaling of energy cost versus number of computations can differ from digital computers', and that this is one of their main attractions.

[1] Large-Scale Optical Neural Networks Based on Photoelectric Multiplication, R. Hamerly et al., 2019

[2] Optical Transformers, M. Anderson et al., 2024

[3] Thermodynamic Computing System for AI Applications, D. Melanson et al., 2025

---

> ### Author Response · Authors · 2026-04-07
> **Review Response**
>
> We thank the reviewer for this important observation regarding the energy scaling properties of neuromorphic and analog accelerators.
>
> ---
>
> ## Comment: Spike-transport energy model is not universally applicable across all neuromorphic/analog modalities.
>
> We agree with the reviewer that the spike-transport energy model is specific to electronics-based implementations and does not generalize to all neuromorphic/analog modalities. In the revised manuscript, we explicitly address this distinction in **Section 3.1.7** ("Neuromorphic energy challenges"), where we note that in alternative physical substrates—such as photonic neural networks (Hamerly et al., 2019), optical transformers (Anderson et al., 2024), and thermodynamic processors (Melanson et al., 2025)—data can be copied and transported with negligible marginal energy cost after initial encoding, because the underlying physics permits passive signal fan-out or energy-recycling operations. We further discuss how this leads to fundamentally different big-O scaling of energy with respect to operation count: whereas digital and electronic-neuromorphic systems scale energy at least linearly, photonic and thermodynamic approaches can in principle achieve sub-linear energy scaling for large matrix-vector products, which is one of their primary theoretical attractions. We also cross-reference this discussion from the neuromorphic accelerator architecture section (**Section 4.7**), where we note that the broader neuromorphic landscape includes photonic and thermodynamic substrates where data transport can be nearly free after initial encoding, yielding qualitatively different energy-scaling properties. Finally, we acknowledge that whether these advantages survive system-level integration (laser sources, detectors, thermal management) remains an active area of investigation, ensuring the discussion is appropriately balanced. All three suggested references have been cited in both sections.

---

### Author Response · Authors · 2026-04-08
**Revision Summary**

We thank all the reviewers for taking the time to read our paper. We have attempted to address all comments from reviewers. We have updated the main document and also added a supplementary pdf with color coding where each reviewer comment is highlighted with a different color. Reviewer DvVg is blue, Reviewer NgTZ is red and Reviewer dXoW is green. Below is a summary of changes



- **Table 1** added: quantitative comparison of eight accelerator platforms across peak compute, TDP, TOPS/W, memory bandwidth, and memory capacity.
- **Figure 3** added: unified roofline view mapping six workload categories to a reference accelerator ceiling.
- **Figure 4** added: log–log TOPS vs. TOPS/W scatter plot with iso-power envelopes and workload-affinity color coding.
- **Table 2** added: per-workload hardware requirements mapped to best-fit platform families.
- **Figure 5** (expanded from original Figure 2): now maps each accelerator family to trade-offs across five design axes (speed, energy, memory/communication, area/cost, benchmarking).
- **Section 3 preamble** expanded: dedicated discussion of flexibility and programmability as a sixth design dimension, with per-family analysis of the flexibility–efficiency tension.
- **Section 3.1.4** expanded: nuanced FPGA energy analysis distinguishing low-end edge (1–5 W) from high-end datacenter (75–225 W) devices, with sensitivity to mapping quality and utilization.
- **Section 3.1.7** added: discussion of alternative neuromorphic substrates (photonic, reversible, thermodynamic) with sub-linear energy scaling properties, citing Hamerly et al. (2019), Anderson et al. (2024), and Melanson et al. (2025).
- **Section 4.6** expanded: PUMA compiler, NeuroSim/MNSIM simulators, and non-ideality-aware training coverage.
- **Section 4.6.2** expanded: detailed backpropagation mapping on crossbar arrays (forward, backward, weight update phases), Manhattan update rule, and mixed-signal dataflows.
- **Section 4.6.3** added: in-sensor computing discussion covering PIP architectures, CIS process-node constraints, and scoping rationale, citing Zhou & Chai (2020), Mennel et al. (2020), and Hsu et al. (2023).
- **Section 4.7** expanded: IBM TrueNorth (GALS architecture, 4,096 cores, ~65 mW) and NorthPole (256 cores, 224 MB SRAM, 25× GPU energy efficiency) added alongside Loihi 2 to span the full neuromorphic design spectrum, citing Akopyan et al. (2015), Merolla et al. (2014), and Modha et al. (2023).
- **Section 4.7 cross-reference** added: pointer from neuromorphic accelerator section back to Section 3.1.7 for photonic/thermodynamic energy-scaling discussion.
- **Section 5.5** added: dedicated compiler/runtime subsections covering torch.compile, TensorRT, TVM/XLA, Triton, code generation for tiled systolic architectures, and serving runtimes (vLLM/PagedAttention, continuous batching, prefix caching, speculative decoding).
- **Evaluation section** expanded: TOPS/mm² (area efficiency) and W/mm² (power density) metrics added with Pareto analysis discussion.
- **Formatting** corrected: repetitive number sequences on pages 3, 4, 8, 12 resolved.
- **Redundancy** reduced: per-family subsections tightened with cross-references replacing restated definitions.

---

### Decision · Action_Editor_CZAB · 2026-05-11

**Recommendation:** Accept as is

**Audience:**

Yes

**Audience Explanation:**

This paper would be important to the community to know what is needed to make the ML algorithms work in practice.

**Claims And Evidence:**

Yes

**Claims Explanation:**

The paper is a survey paper on hardware accelerators for neural networks. All the reviewers are happy with the contributions made by the paper. Hardware accelerators are extremely important to realize the vision of AI. This paper has provided an extensive survey of papers that are important in this area.